# A visual pathway for skylight polarization processing in *Drosophila*

**Ben J Hardcastle[1]\*, Jaison J Omoto[2‡], Pratyush Kandimalla[2‡], Bao-Chau M Nguyen[2], Mehmet F Keleş[1§], Natalie K Boyd[2], Volker Hartenstein[2†], Mark A Frye[1†]**

[1]Department of Integrative Biology and Physiology, University of California, Los Angeles, Los Angeles, United States; [2]Department of Molecular, Cell and Developmental Biology, University of California, Los Angeles, Los Angeles, United States

**Abstract** Many insects use patterns of polarized light in the sky to orient and navigate. Here, we functionally characterize neural circuitry in the fruit fly, *Drosophila melanogaster*, that conveys polarized light signals from the eye to the central complex, a brain region essential for the fly's sense of direction. Neurons tuned to the angle of polarization of ultraviolet light are found throughout the anterior visual pathway, connecting the optic lobes with the central complex via the anterior optic tubercle and bulb, in a homologous organization to the 'sky compass' pathways described in other insects. We detail how a consistent, map-like organization of neural tunings in the peripheral visual system is transformed into a reduced representation suited to flexible processing in the central brain. This study identifies computational motifs of the transformation, enabling mechanistic comparisons of multisensory integration and central processing for navigation in the brains of insects.

**\*For correspondence:**
hardcastle@ucla.edu

[†]These authors contributed equally to this work

**Present address:** [‡]Division of Biology and Biological Engineering, California Institute of Technology, Pasadena, United States; [§]Department of Neurology, Johns Hopkins University, Baltimore, United States

**Competing interests:** The authors declare that no competing interests exist.

## Introduction

A critical challenge of active locomotion is knowing the right way to go. Sensorimotor reflexes can influence momentary changes in direction to hold a course or to avoid looming threats, but goal-directed behaviors, such as returning to a previous location from unfamiliar surroundings, require additional information and processing (*Braitenberg, 1986*; *Gomez-Marin et al., 2010*). External sensory cues must be transformed into an internal representation of position and orientation within the environment, which can also be modified by past experience (*Collett and Collett, 2002*). In Dipteran flies, as in other invertebrates, a collection of neuropils known as the central complex (CX) is believed to coordinate such behaviors and plays a role in spatial memory, object memory, and action selection (*Giraldo et al., 2018*; *Neuser et al., 2008*; *Ofstad et al., 2011*; *Strausfeld and Hirth, 2013*), in addition to homeostatic processes including hunger and sleep (*Donlea et al., 2014*; *Dus et al., 2013*; *Liu et al., 2016*).

Recent studies in *Drosophila* have revealed that activity in a network of CX neurons encodes and maintains a representation of the animal's angular heading relative to its environment (*Kim et al., 2017*; *Seelig and Jayaraman, 2015*), with similarity to head-direction cells in vertebrates (*Taube et al., 1990*). This neural representation of heading can be updated by internal, proprioceptive estimates of self-motion during locomotion, and by external cues, such as moving visual patterns and directional airflow (*Fisher et al., 2019*; *Green et al., 2017*; *Kim et al., 2019*; *Okubo et al., 2020*; *Shiozaki et al., 2020*). In other insects, including locusts, crickets, bees, butterflies, and beetles, the functional organization of the CX has frequently been studied in the context of navigation via celestial cues, particularly polarized light (*Heinze, 2014*). The nearly ever-present pattern of polarization in the sky, formed by scattering of light in the atmosphere, offers an indicator of

orientation to organisms able to detect and interpret it, and may be more stable than terrestrial landmarks (*Cronin and Marshall, 2011*; *Dacke et al., 2003*; *Horváth and Varju, 2004*; *Mappes and Homberg, 2004*; *Frisch, 1949*; *Wehner and Müller, 2006*). In these non-Dipteran insects, a multi-modal neural circuit transmits polarization signals from the eyes to the central complex (*el Jundi et al., 2015*; *el Jundi et al., 2014*; *Heinze, 2013*; *Heinze and Reppert, 2011*; *Homberg et al., 2011*; *Pfeiffer et al., 2005*). This circuit is known as the 'sky compass' pathway for its proposed role in processing skylight polarization patterns and information about the position of the sun to bestow an animal with a sense of direction. In *Drosophila*, the anterior visual pathway (AVP), which comprises neurons connecting the medulla, anterior optic tubercle, bulb, and ellipsoid body, has been postulated to represent the homologue of the sky compass pathway (*Omoto et al., 2017*; *Timaeus et al., 2020*; *Warren et al., 2019*). Visual processing in the AVP appears to be segregated into three topographically organized, parallel streams, of which two have been shown to encode distinct small-field, unpolarized stimuli (*Omoto et al., 2017*; *Seelig and Jayaraman, 2013*; *Shiozaki and Kazama, 2017*; *Sun et al., 2017*). The neurons involved in polarization processing in *Drosophila* have not been identified beyond peripheral circuits of the dorsal rim area, a specialized region of the eye for detecting skylight polarization (*Fortini and Rubin, 1991*; *Wada, 1974*; *Weir et al., 2016*; *Weir and Dickinson, 2015*; *Wernet et al., 2012*; *Wolf et al., 1980*).

A detailed mapping of the relevant polarization-sensitive neurons would allow the exquisite genetic tools and connectomic studies available in *Drosophila* (*Scheffer et al., 2020*) to be leveraged to understand the workings of the CX and its integration of multiple sensory modalities. Behavioral experiments have demonstrated that *Drosophila* orient relative to polarization patterns while walking and in tethered-flight (*Mathejczyk and Wernet, 2019*; *Stephens et al., 1953*; *Warren et al., 2018*; *Weir and Dickinson, 2012*; *Wernet et al., 2012*; *Wolf et al., 1980*). A comparative approach would therefore provide insight into the processing strategies employed across taxa as well as species-specific adaptations (*Honkanen et al., 2019*). Furthermore, it may be possible to reconcile the existing evidence of a common, fixed representation of polarization patterns in the CX of non-Dipteran insects (*Heinze and Homberg, 2007*; *Heinze and Reppert, 2011*) with the emerging model of a flexible representation of both visual information and heading direction in the *Drosophila* CX (*Fisher et al., 2019*; *Kim et al., 2019*; *Kim et al., 2017*; *Seelig and Jayaraman, 2015*; *Turner-Evans et al., 2020*). Alternatively, fundamental differences in the organization and processing of polarized light signals between species may reflect specialized navigational requirements.

Here, we set out to test the hypothesis that the anterior visual pathway conveys polarized light signals from the eye to the central complex in *Drosophila*. We used neurogenetic tracing techniques and in vivo calcium imaging to characterize the organization of the neurons at each stage and their coding and transformation of visual features. We show that parallel circuitry in the medulla conducts polarization signals from photoreceptors in the dorsal rim area to a stereotyped domain of the anterior optic tubercle. From there, a postsynaptic population of neurons projecting to the anterior bulb relays polarization signals to ring neurons of the ellipsoid body, and in turn, the 'compass neurons' of the central complex. The superior bulb multiplexes polarized and unpolarized light signals, while the inferior bulb does not appear to be involved in polarization processing. Finally, we examine population responses in the central complex and find hallmarks of a flexible encoding of a single angle of polarization which could be used to direct motor output for navigation behavior.

## Results

### Polarization processing in the medulla dorsal rim area

In flies, the pair of inner photoreceptors in each ommatidium, R7/8, are involved in the detection of color and linear polarization of light (*Hardie, 1984*). Within a narrow strip of skyward-facing ommatidia in each eye, known as the dorsal rim area (DRA), each R7/8 pair is sensitive to a different angle of polarization (AoP, also referred to as the e-vector orientation), organized in a 'polarotopic' fashion (*Figure 1A*). This specialized array of polarization detectors covers the complete 180° range of orientations and, with a peak spectral sensitivity to UV light, is well-suited to sensing the patterns of polarized light in the sky (*Feiler et al., 1992*; *Salcedo et al., 1999*; *Sharkey et al., 2020*; *Weir et al., 2016*). A previous characterization of DRA R7/8 in *Drosophila* established their visual

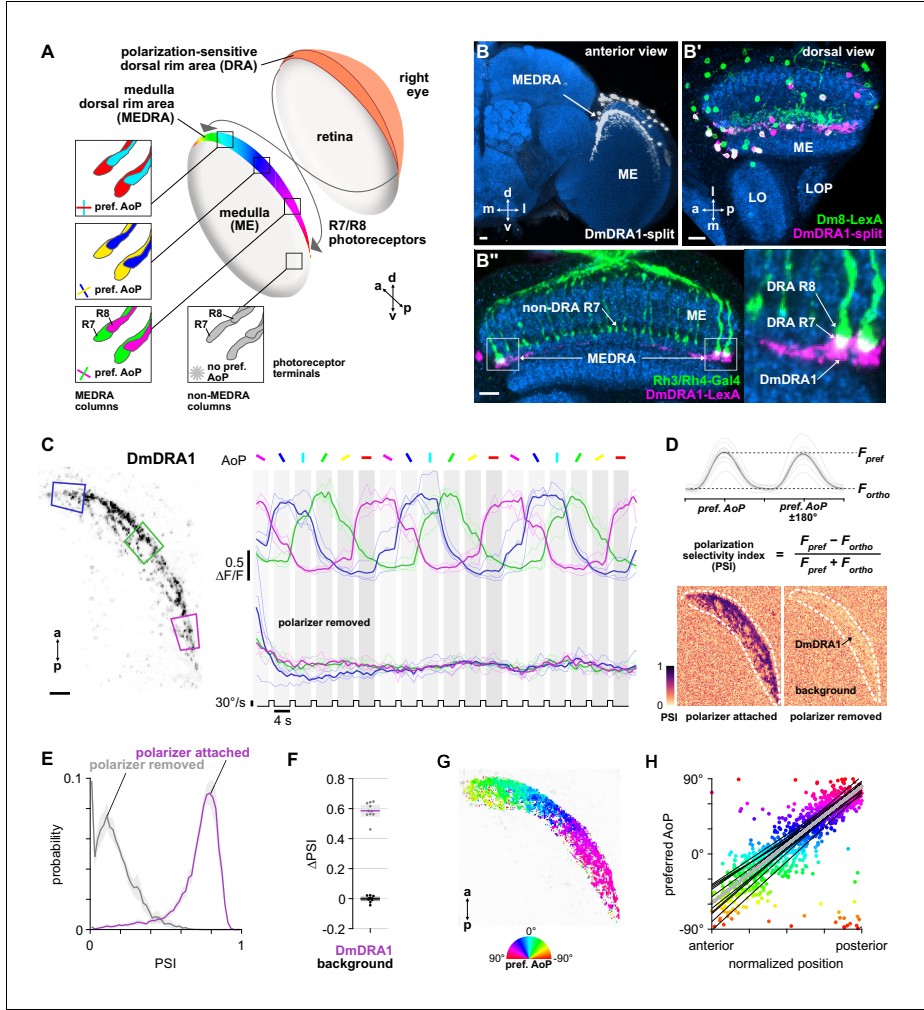

**Figure 1.** Polarization processing in the medulla dorsal rim area. (A) Schematic of the dorsal rim area (DRA) of the right eye and the projection of DRA R7/R8 photoreceptors to corresponding columns in the medulla dorsal rim area (MEDRA) across the first optic chiasm. Insets: R7 and R8 terminals in each individual column are tuned to orthogonal angles of polarization (AoP), and their tunings change linearly across the MEDRA. R7/R8 outside the MEDRA are equally sensitive to all AoPs. (B) Confocal projection (anterior view) of DmDRA1 expression pattern in the MEDRA (DmDRA1-split>GFP). (B') Dual-labeling of Dm8 and DmDRA1 neurons (dorsal view) (R24F06-LexA>GFP, green; DmDRA1-split>RFP, magenta) (mean cell bodies per brain hemisphere, DmDRA1: 23.13, SEM 1.16; Dm8∩DmDRA1: 21.25, SEM 0.49, N = 8 animals). Overlap in expression (visible as white cell bodies) indicates neurons in split driver are DmDRA1 neurons, and not DmDRA2 (*Sancer et al., 2019*). (B'') Dorsal view of the medulla showing DRA R7/R8 photoreceptors (Rh3/Rh4-Gal4, green) and their proximity to DmDRA1 neurons (R13E04-LexA, magenta), indicating contact between R7 and DmDRA1. Right: Enlargement of medulla dorsal rim area (MEDRA). Scale bars denote 10 μm. (C) Left: Example time-averaged maximum-intensity projection showing dorso-posterior two-photon imaging view of GCaMP activity in DmDRA1 neurons (DmDRA1-split>sytGCaMP6s). Three ROIs were manually drawn in anterior (blue), dorsal (green), and posterior (magenta) MEDRA in each recording. Scale bar denotes 10 μm. Right: Time-series of normalized mean intensity values for ROIs in equivalent positions in three animals (thin traces) and their mean (thick trace), with the polarizing filter (polarizer) attached (top) and removed (bottom). Shaded patches denote periods that the polarizer remained at a fixed orientation. (D) Definition of polarization-selectivity index (PSI), as obtained from polarization tuning curves (top, see *Appendix 1—figure 1*), and example spatial maps of PSI for the example recordings in (C), with the polarizer attached (bottom, left) and removed (bottom, right). (E) Probability distributions of PSI values in DmDRA1 neurons with the polarizer attached (average PSI DmDRA: 0.74, CI95 [0.70 0.78], N = 10 animals) and removed (average PSI DmDRA1 control: 0.16, CI95 [0.18 0.21], N = 7 animals). Mean ± SEM. (F) Effect of polarizer on median PSI values versus controls with polarizer removed, within DmDRA1 neurons (light dots) and background regions (dark dots) in individual animals (DmDRA1, pink line: mean ΔPSI = 0.59, CI95 [0.55 0.63], N = 10, p < 10⁻⁶ t-test; background,

*Figure 1 continued on next page*

*Figure 1 continued*

black line: mean ΔPSI = −0.002, CI95 [−0.02 0.01], N = 10, p=0.82, t-test). (**G**) Example polarization tuning map for DmDRA1. Preferred angles of polarization are shown for each pixel with an above-threshold PSI value using the color map shown. Pixels with a below-threshold PSI value, or falling outside an ROI drawn around the DmDRA1 population, show average intensity in grayscale. Data shown are from maximum-intensity projections through the MEDRA. (**H**) Scatter plot showing the common polarotopic organization of DmDRA1 neurons. Individual points represent pixels recorded from DmDRA1 neurons, showing their normalized horizontal position in the MEDRA and their preferred angle of polarization (AoP). Thin lines show linear-circular fits for data from individual animals with significant correlations (mean individual ρ = 0.89, CI95 [0.85 0.91], N = 10 animals), thick line shows fit for all pooled data (pooled ρ = 0.85, CI95 [0.79 0.88], N = 10 recordings, $p < 10^{-4}$ permutation test).

The online version of this article includes the following figure supplement(s) for figure 1:

**Figure supplement 1.** Polarizer stimulus characterization and R7/R8 photoreceptor stimulation.

**Figure supplement 2.** Polarization-opponent flash responses in DmDRA1.

---

response properties and the spatial organization of their preferred angles of polarization (*Weir et al., 2016*). Here, we followed the pathway for skylight polarization signals from the eye and investigated direct downstream targets of DRA R7/R8s at their axon terminals in the second optic neuropil, the medulla (ME).

## DmDRA neurons

First, we concentrated on distinct morphological forms of distal medulla (Dm) interneurons which are localized to the medulla dorsal rim area (MEDRA). Two types of these interneurons have been anatomically characterized, DmDRA1 and DmDRA2. Individual DmDRA1 neurons span approximately 10 MEDRA columns and receive input exclusively from DRA R7 photoreceptors while avoiding input from non-DRA columns (*Sancer et al., 2019*). DmDRA2 receives exclusive input from DRA R8 photoreceptors. Due to their contact with polarization-sensitive photoreceptors, both DmDRA subtypes are thought likely to respond to polarized light (*Sancer et al., 2019*). To test this, we generated a split-Gal4 driver (R13E04-p65.AD, VT059781-Gal4.DBD) for a population of DmDRA neurons (*Figure 1B*; *Courgeon and Desplan, 2019*; *Jenett et al., 2012*). To identify which subtype expressed this driver, we co-labeled it with an established Dm8 driver (R24F06-LexA) which is known to be expressed in DmDRA1 and not DmDRA2 (*Sancer et al., 2019*). We found highly overlapping expression between these drivers (*Figure 1B'*), indicating that the split-Gal4 is predominantly expressed in DmDRA1. We confirmed that DmDRA neurons in the split-Gal4 were also in close proximity to photoreceptor terminals in the MEDRA, and found clear overlap with the proximal tip of each DRA R7/R8 pair, providing further evidence of exclusive contact with DRA R7 (*Figure 1B''*). Hereafter, we refer to this driver as the DmDRA1-split.

## DmDRA1 responses to polarized light

After validating a polarized light stimulus by confirming the previously characterized response properties of DRA R7/R8 (*Weir et al., 2016*; *Figure 1—figure supplement 1A-F*, see Materials and methods), we recorded presynaptic calcium signals in the DmDRA1-split using GCaMP6s localized to synapses (*Cohn et al., 2015*) while presenting different angles of polarization (AoP) to the dorsal rim (*Figure 1C*, *Figure 1—figure supplement 1A-F*). We found that the activity of DmDRA1 neurons varied with the AoP presented and followed an approximately sinusoidal response profile typical of polarization-sensitive neurons (*Heinze, 2013*).

To quantify the extent to which the neurons were modulated by the AoP, we calculated a polarization-selectivity index (PSI) by comparing the peak response with the response at orthogonal angles (*Figure 1D*). We used absolute fluorescence intensity values to calculate PSI, rather than normalized ΔF/F values (see *Appendix 1–figure 1* for polarization tuning curves using activity normalized to baseline), and found the difference between preferred and anti-preferred responses divided by their sum. This facilitated comparison across neurons with different baseline activity levels and differs from the polarization-sensitivity (PS) metric used in previous studies, which could result in negative values or, in the case of low baseline activity, very large values. PSI values had a minimum possible value of 0, indicating equal responses to all angles presented, and a maximum of possible value of 1, indicating maximum response to two diametrically opposite angles with zero activity at their two

respective orthogonal angles. Among DmDRA1 neurons, we found high PSI values throughout the population with an average of 0.74, while background regions in each recording contained an average PSI of 0.20 (*Figure 1D,E*, *Appendix 1–table 1*).

To verify that the neurons were modulated by the changing AoP and not any other unintended consequence of rotating the stimulus device, we repeated the experiment without the linear polarizer attached to the device. With the polarizer removed, all neurons were suppressed at the initial onset of unpolarized UV light and were no longer modulated by the rotation of the device (*Figure 1C*). We then calculated the difference in PSI values between experiments performed with and without the polarizer, which we refer to as ΔPSI (*Figure 1F*). With the polarizer removed, PSI values in DmDRA1 neurons fell by approximately 80% (*Figure 1E,F*), reflecting their lack of modulation and resulting in an average ΔPSI of 0.59, whereas PSI values in the background showed no change (*Figure 1F*), resulting in an average ΔPSI of 0.

## DmDRA1 polarotopy in the MEDRA

Within the population of DmDRA1 neurons, we observed preferential responses to different angles of polarized light depending on their position in the MEDRA (*Figure 1C,G*). The preferred AoP showed a linear relationship with position, which we refer to as polarotopy (*Figure 1H*). Moving anterior to posterior in the right optic lobe, the preferred AoP shifted counter-clockwise (*Figure 1G, H*). This polarotopy was mirrored in the left optic lobe, with a similar range of preferred AoPs represented in the opposite posterior-anterior direction (*Figure 1—figure supplement 1I*). Throughout the MEDRA, the preferred AoPs of DmDRA1 neurons closely matched those of R8 photoreceptors at similar positions (*Figure 1H*, *Figure 1—figure supplement 1E*). Since R7/R8 are likely inhibitory (*Davis et al., 2020*; *Gao et al., 2008*), we expected that the preferred AoP of a neuron postsynaptic to either R7 or R8 would be shifted by 90°. We therefore posit that it is R7 signals that are responsible for the predominant response characteristics of DmDRA1 neurons, consistent with the anatomical overlap between DmDRA1 and R7 terminals (*Figure 1B''*) and the established connectivity of the DmDRA1 subtype (*Sancer et al., 2019*).

## Polarization-opponency in DmDRA1 neurons

We then asked whether DmDRA1 responses demonstrate polarization opponency, with activity increases at preferred angles and inhibition at anti-preferred angles. This would likely require antagonistic processing of local, orthogonally tuned R7 and R8 signals in the MEDRA. Although DmDRA1 does not contact R8, inhibitory interactions between R7/R8 in each column suggest that direct input from R8 may not be necessary to influence activity in DmDRA1 neurons (*Schnaitmann et al., 2018*; *Weir et al., 2016*). To explore opponent DmDRA1 responses, we employed brief flashes of polarized light at orthogonal angles. We first identified anterior regions in the MEDRA where the preferred AoP of DmDRA1 was found to be around 0° in the previous tuning experiment (*Figure 1G*) and generated ROIs around pixels with similar preferred AoPs (*Figure 1—figure supplement 2*). We then measured the responses of each ROI to flashes of UV light with 0° and 90° AoP (*Figure 1—figure supplement 2*). The preferred AoP of 0° caused an increase in activity relative to baseline, while flashes at 90° caused a decrease of greater magnitude, followed by a slight rebound above baseline after the offset of the flash (*Figure 1—figure supplement 2*). For light flashes with the polarizer removed we observed inhibition of DmDRA1 at all regions, regardless of position in the MEDRA (*Figure 1—figure supplement 2*).

Further to this, the normalized tuning curves of DmDRA1 responses demonstrated inhibition over most of the range of angles presented and above-baseline activity for a narrower range (*Appendix 1–figure 1A*). Taken together, these results support a model of polarization-opponent processing, whereby DmDRA1 neurons are inhibited by anti-preferred angles of polarization and by unpolarized light, and 'excited' by preferred angles of polarization. This excitation is likely to be a release from R7-mediated inhibition, with R7 and DmDRA1 tuned to orthogonal angles of polarization.

## Medulla projection neurons conveying polarization signals from the MEDRA to the AOTU

In other insect species, polarization-sensitive photoreceptors in the dorsal rim are thought to provide input to transmedulla neurons (also referred to as line-tangential neurons), which project from the optic lobe to the anterior optic tubercle (AOTU) (*el Jundi et al., 2011*; *Homberg et al., 2003*; *Immonen et al., 2017*; *Pfeiffer and Kinoshita, 2012*; *Zeller et al., 2015*). In all species investigated, it is the small subunit of the AOTU (often called the lower-unit, LU) which is involved in processing polarized light signals (*Heinze, 2013*), although to our knowledge these signals have not been explored in transmedulla neurons themselves. In *Drosophila*, corresponding medullo-tubercular (MeTu) neurons have been described (*Figure 2A*), some of which have been shown to play a role in color vision-dependent behaviors (*Omoto et al., 2017*; *Otsuna et al., 2014*). The dendrites of individual MeTu neurons typically innervate 10–15 columns of the medulla in layers M6–7 (*Omoto et al., 2017*; *Figure 2—figure supplement 1*) and, as an ensemble, tile larger areas of the medulla (*Figure 2A*). We predicted that MeTu neurons with dendrites in the MEDRA would be postsynaptic to DmDRA1 neurons and/or DRA R7/R8, and would therefore similarly respond to polarized light.

### Parallel inputs to MeTu neurons from DmDRA1 and DRA R7 photoreceptors in the MEDRA

We used the anterograde circuit tracing technique, *trans*-Tango (*Talay et al., 2017*), to identify putative postsynaptic partners of the DmDRA1 neurons (*Figure 2B*). We found that DmDRA1-split driving *trans*-Tango labeled a population of neurons in the dorsal medulla that project to the small, lateral subunit of the AOTU via a fiber bundle in the anterior optic tract (AOT) (*Figure 2B*), and matched the anatomy of MeTu neurons (*Figure 2A*). We then performed a similar experiment to label putative postsynaptic partners of DRA R7 photoreceptors and explore whether they also provide input to MeTu neurons. We used a Gal4 driver which targets neurons expressing the UV-sensitive rhodopsins Rh3 and Rh4 (pan-R7-Gal4, which we refer to as Rh3/Rh4-Gal4), which includes DRA R7/R8. We again found *trans*-Tango labeling of the small subunit of the AOTU (*Figure 2C*), indicating connectivity between UV-sensitive photoreceptors and MeTu neurons.

However, since the Rh3/Rh4 driver is not exclusively expressed by DRA R7 photoreceptors (*Figure 2C*), the labeling of MeTu neurons we observed could have been due to other synaptic input, for example from DRA R8 or photoreceptors outside of the MEDRA. We were unable to evaluate this from the *trans*-Tango experiment alone, due to the high density of labeled neurons in the medulla (*Figure 2C*). We therefore co-labeled a population of MeTu neurons (R56F07-Gal4) along with all R7/R8 photoreceptors (antibody 24B10, *Fujita et al., 1982*; *Figure 2—figure supplement 1A*) and closely examined DRA R7 terminals and their possible contact with MeTu neurons. Throughout layer M6 in the dorsal medulla, MeTu dendrites were in close proximity to non-DRA R7 terminals and at the dorsal-posterior margin we found clear overlap with DRA R7 terminals, distinguishable by their larger terminals and deeper projections (*Figure 2—figure supplement 1A*). In short, at least some MeTu neurons appear to receive input from DRA R7 photoreceptors (in addition to non-DRA photoreceptors and possibly DRA R8) and the putative connections revealed here suggest a parallel pathway for polarization signals in the MEDRA: DRA R7→DmDRA1, DmDRA1→MeTu, DRA R7→MeTu (*Figure 2D*).

### AOTU innervation by populations of MeTu neurons

Several discrete populations of MeTu neurons have been characterized based on the distinct domains of the small subunit of the AOTU that their terminals occupy: intermediate-medial (im), intermediate-lateral (il), and lateral (l), which is further divided into anterior (la), central (lc), and posterior (lp) domains (*Figure 2A',A'',D*). The larger subunit comprising the medial domain (m) is not innervated by MeTu neurons and corresponds to the polarization-insensitive upper-unit (UU) of other species (*Omoto et al., 2017*; *Timaeus et al., 2020*). We examined the domains of the AOTU targeted by the putatively polarization-sensitive MeTu neurons which were labeled by *trans*-Tango (*Figure 2B'–C'*). Both the DmDRA1 and Rh3/Rh4 *trans*-Tango experiments predominantly labeled the intermediate-lateral domain (AOTUil), with encroachment on the lateral domain (AOTUl) (*Figure 2B''–C''*). We found no detectable intermediate-medial (AOTUim) or medial (AOTUm) labeling in either experiment (*Figure 2B'–C'*).

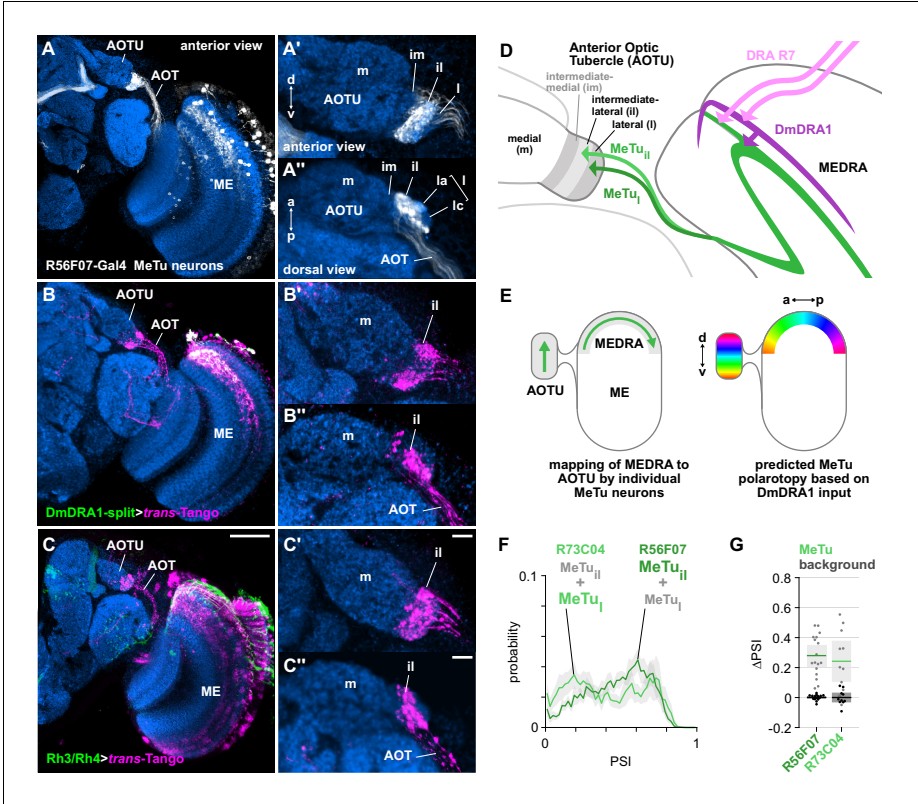

**Figure 2.** Medulla projection neurons receive polarization signals from DmDRA1 and DRA photoreceptors. (**A**) Confocal projection (anterior view) of R56F07-Gal4 driving a population of MeTu neurons with dendrites in the dorsal medulla (ME) and projections to the anterior optic tubercle (AOTU) via the anterior optic tract (AOT). High-magnification anterior (**A'**) and dorsal (**A''**) views showing AOTU domains innervated by MeTu neurons in the driver line. (**B**) Confocal projection (anterior view) of *trans*-Tango signal (magenta) labeling putative postsynaptic partners from DmDRA1-split (green, visible as white due to overlap). High-magnification anterior (**B'**) and dorsal (**B''**) views showing innervation of AOTU by postsynaptic neurons, similar to that of MeTu neurons in (**A**). (**C**) Confocal projection (anterior view) of *trans*-Tango signal (magenta) labeling putative postsynaptic partners from Rh3/Rh4-Gal4 (green), which labels DRA R7/R8 + non-DRA R7. Scale bar denotes 50 μm. High-magnification anterior (**C'**) and dorsal (**C''**) views showing innervation of AOTU by postsynaptic neurons, similar to that of MeTu neurons in (**A**). Scale bars denote 10 μm. (**D**) Schematic of proposed parallel connectivity in the medulla dorsal rim area (MEDRA) and regions of the AOTU targeted by polarization-sensitive MeTu neurons. (**E**) Schematic of proposed one-dimensional mapping of MEDRA position to AOTU position by MeTu neurons (left) based on single-cell clones (see *Figure 2—figure supplement 1B-D*), and predicted polarotopy in the AOTU (right) assuming excitatory input from DmDRA1 neurons in the MEDRA. (**F**) Probability distributions of PSI values in MeTu neurons (average PSI R56F07: 0.48, CI95 [0.41 0.55], N = 17 animals; R73C04: 0.42, CI95 [0.29 0.56], N = 11 animals). Mean ± SEM. (**G**) Effect of polarizer on median PSI values versus controls with polarizer removed, within MeTu neurons (light dots) and background regions (dark dots) in individual animals (R56F07 MeTu, green line: mean ΔPSI = 0.28, CI95 [0.21 0.35], N = 17, p < 10⁻⁶ t-test; R56F07 background, black line: mean ΔPSI = 0.001, CI95 [−0.01 0.01], N = 17, p=0.84, t-test; R73C04 MeTu, green line: mean ΔPSI = 0.242, CI95 [0.11 0.38], N = 11, p=0.03 t-test; R73C04 background, black line: mean ΔPSI = 0.000, CI95 [−0.03 0.03], N = 11, p=0.98, t-test). The online version of this article includes the following figure supplement(s) for figure 2:

**Figure supplement 1.** Retinotopic mapping of medulla dorsal rim area to AOTU by MeTu neurons.

We next identified two Gal4 drivers for populations of MeTu neurons arborizing in the AOTUl and AOTUil: one with dendrites predominantly tiling the dorsal medulla (R56F07-Gal4) (*Figure 2A*) and one with dendrites throughout the medulla (R73C04-Gal4) (Figure 5A; *Omoto et al., 2017*). Both populations contain a mixture of MeTu neurons innervating different domains of the AOTU: R56F07 is predominantly expressed in MeTu neurons innervating the intermediate-lateral (il) domain, with a smaller proportion innervating the anterior portion of the lateral (l) domain (MeTu$_{il}$ and MeTu$_{l}$,

respectively); R73C04 is predominantly expressed in MeTu neurons innervating the three lateral (l) domains, with a smaller proportion innervating the intermediate-lateral (il) domain (MeTu$_l$ and MeTu$_{il}$, respectively) (Figure 5A'). From confocal images of single-cell MCFO (MultiColor FlpOut) clones (*Nern et al., 2015*), we determined a consistent relationship between the anterior→posterior position of MeTu dendrites in the MEDRA and the ventral→dorsal position of MeTu axon terminals in the AOTU (*Figure 2E*, *Figure 2—figure supplement 1B-D*). For MeTu neurons with dendrites outside of the MEDRA, we found no clear relationship between ventrodorsal position in the medulla and mediolateral position in the AOTU, confirming a previous study (*Timaeus et al., 2020*).

## MeTu responses and polarotopy in the AOTU

We recorded presynaptic calcium signals in the AOTU for the two MeTu drivers in response to rotations of the polarizer, as in *Figure 1*. In both MeTu populations, we found broad distributions of PSI values (*Figure 2F*) with generally lower values than in DmDRA1 neurons recorded in the MEDRA (*Figure 1E*, *Appendix 1–figure 1B*), implying more variable and less strong modulation of MeTu activity by the polarizer. Nonetheless, compared to control experiments with the polarizer removed, the polarizer caused a statistically significant increase in average PSI values among both MeTu populations (*Figure 2G*). We observed that the highest PSI values were spatially restricted within the AOTU, appearing as vertical columns or bands within the 2-D projections (*Figure 3A,B*). In R56F07, the most responsive MeTu terminals were found within the most lateral regions of the population (*Figure 3A*, *Figure 3—figure supplement 1A*). In R73C04, the most responsive terminals tended to be clustered in a narrow medial band of the population (*Figure 3B*, *Figure 3—figure supplement*

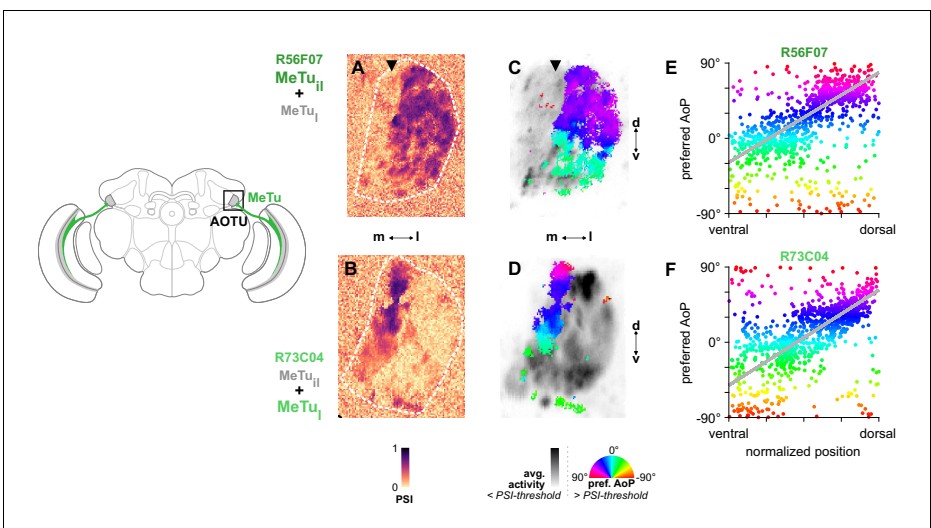

**Figure 3.** MeTu neurons convey polarization signals to the AOTU in a polarotopic fashion. (A) Example spatial map of polarization-selectivity index (PSI) in MeTu terminals in the AOTU (R56F07-Gal4>sytGCaMP6s; predominantly MeTu$_{il}$ neurons innervating intermediate-lateral (il) domain, with smaller proportion of MeTu$_{la}$ innervating lateral-anterior (la) domain, see *Figure 2*). Arrowhead indicates vertical band in medial region of population with low PSI values cf. average activity in (C). See also *Figure 3—figure supplement 1A*. (B) Example spatial map of PSI in MeTu terminals in the AOTU for an alternative driver (R73C04-Gal4>sytGCaMP6s; predominantly MeTu$_l$ neurons innervating lateral (l) domains, with smaller proportion of MeTu$_{il}$ innervating intermediate-lateral (il) domain, see *Figure 5A'*). (C) Example polarization tuning map for above-threshold pixels in R56F07 MeTu neurons from the example recording in (A). (D) As in (C), for R73C04 MeTu neurons from the example recording in (B). (E) Scatter plot showing the predominant polarotopic organization of R56F07 MeTu neurons. Individual points represent pixels recorded in MeTu neurons, showing their normalized vertical position in the MEDRA and their preferred angle of polarization (AoP). Line shows fit for all pooled data (pooled ρ = 0.71, CI95 [0.54 0.80], N = 8 recordings, p < 10–4 permutation test). (F) As in (E), for R73C04 MeTu neurons (pooled ρ = 0.56, CI95 [0.42 0.69], N = 10 recordings, p < 10$^{-4}$ permutation test).

The online version of this article includes the following figure supplement(s) for figure 3:

**Figure supplement 1.** Organization of polarization-selective responses in MeTu neurons in the AOTU.

*1B*), likely corresponding to the anterior region of AOTUil and possibly AOTUla. These spatially localized regions of polarized light modulated activity indicated that both MeTu populations contained subpopulations of strongly polarization-sensitive neurons, which occupied a common region in the AOTU, while adjacent regions contained terminals of MeTu neurons with lower polarization-sensitivity. We surmise that these regions of differing polarization-sensitivity result from each population containing a combination of MeTu neurons with dendrites contacting the MEDRA, which constitutes only around 5% of medulla columns (*Weir et al., 2016*), and neurons with dendrites outside the MEDRA. We also note that the proportion of low PSI values (<0.5) was slightly higher in the population containing neurons with dendrites in the ventral and dorsal medulla (R73C04) compared to the dorsal-only population (R56F07), which likely contained a higher proportion of MEDRA-contacting neurons (*Figure 2F*, *Figure 3A,B*).

Based on the polarotopic organization of R7/R8 and DmDRA1 in the MEDRA and the anatomical mapping of the MEDRA to the AOTU by MeTu neurons (*Figure 2E*), we were able to predict that polarization-sensitive MeTu neurons would exhibit a counter-clockwise shift in their preferred AoP from ventral to dorsal in the right AOTU. To assess this, we examined pixels with above-threshold PSI values (greater than the mean background value + 1 SD, see Materials and methods), which limited the analysis to polarization-sensitive MeTu terminals (*Figure 3C,D*). Across animals, both populations showed a predominant polarotopic organization which matched our prediction: from ventral to dorsal in the right AOTU, the preferred AoP shifted counter-clockwise (*Figure 3E,F*). This polarotopy is consistent with MeTu neurons receiving polarized light responses from either DmDRA1 or DRA R7 in the MEDRA and conveying them to the AOTU with the positional mapping we identified (*Figure 2D,E*). Consistent with this mapping, we observed no clear relationship between preferred AoP and horizontal position (*Figure 3—figure supplement 1A,B*). However, we observed vertical organizations of responses which deviated from the norm in approximately 20% of experiments across both drivers. The most common of these resembled an inverted form of the predominant polarotopy (from ventral to dorsal in the right AOTU, the preferred AoP rotated clockwise) and also typically contained tunings to a different range of AoPs than the predominant organization (*Figure 3—figure supplement 1E,E'*). Although we could not determine why one organization was observed over another, this finding suggests that a further transformation of MeTu responses may take place. However, a reversed mapping of responses could be achieved by combining signals originating from the contralateral eye (*Figure 1—figure supplement 1G,H*), which we explore in the following section.

## Multiple visual features encoded in the AOTU

We wondered whether functional divisions of MeTu responses exist within the AOTU which could contain, for example, spatially segregated responses to unpolarized visual features, such as narrow vertical bars shown to stimulate downstream neurons. We recorded MeTu responses to unpolarized, small-field vertical bar stimuli presented at different azimuth positions in the visual field. We then explored the spatial distribution of polarization- and bar-sensitive regions within the AOTU by constructing combined activity maps (*Figure 4A*, right). Distinct regions could be observed which were modulated by either the polarized or unpolarized light stimuli (green or blue, respectively, *Figure 4A*), along with areas that were equally active during both experiments (white, *Figure 4A*, right). In the previous polarization tuning experiment, MeTu terminals in the R73C04 population exhibited high PSI values within a vertical band on the medial side (*Figure 3B,D*, *Figure 3—figure supplement 1B*) and below-threshold values elsewhere. Here, it could be seen that the lateral side of the population also contained regions which responded more strongly to the polarized stimulus than the unpolarized bars, despite their low PSI values (*Figure 3B,D*). We explored whether a polarotopic organization of responses also existed within this region, and whether responses to the unpolarized bars exhibited a corresponding retinotopic organization.

Within ROIs drawn on lateral MeTu terminals in R73C04 (likely occupying the ventral AOTUlc domain, green ROIs, *Figure 4A*, left), we found moderate modulation of activity during the rotation of the polarizer (*Figure 4B*) which was sufficient to identify the preferred AoP. Like the terminals with above-threshold PSI values (*Figure 3D*), we observed a vertical polarotopic organization consistent with the anatomical mapping of MeTu neurons (*Figure 2—figure supplement 1B–D*): from ventral to dorsal, the AoP rotated counter-clockwise in the right AOTU and clockwise in the left AOTU

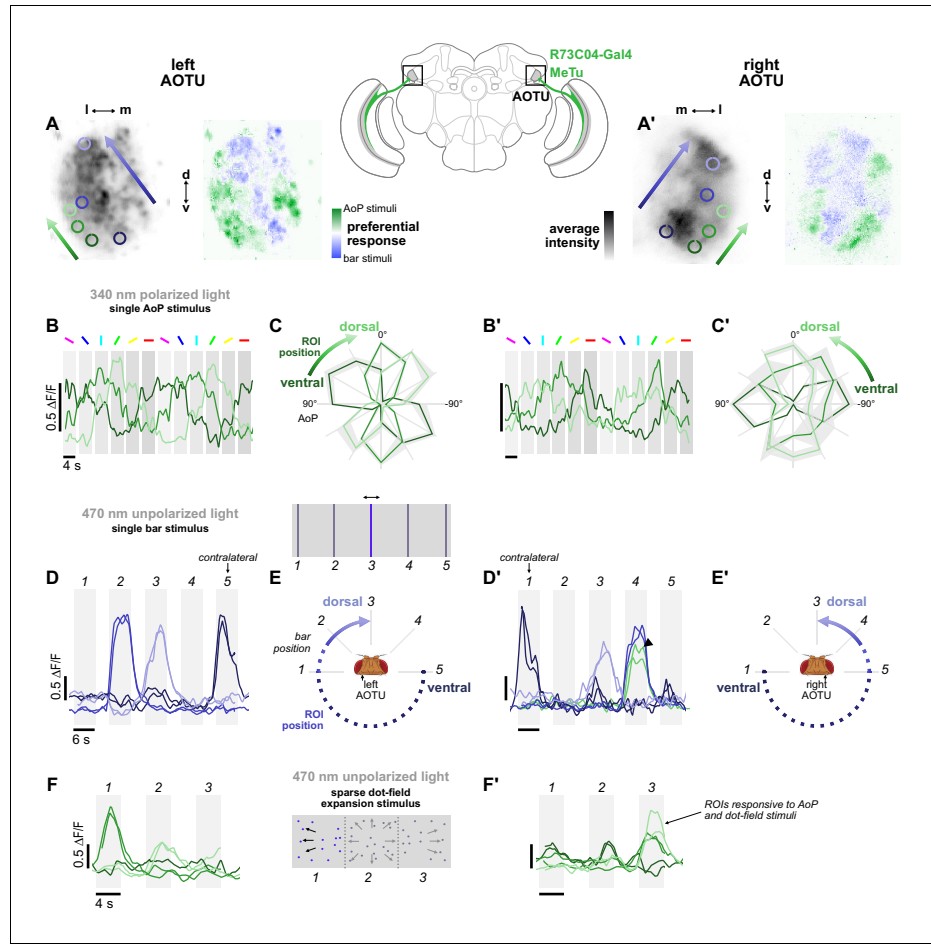

**Figure 4.** Retinotopic encoding of multiple visual features in the AOTU. (**A**) Left: Example time-averaged maximum-intensity projection showing GCaMP activity in R73C04 MeTu neuron terminals in the left AOTU and examples of lateral ROIs (green) and medial ROIs (blue) (R73C04-Gal4>sytGCaMP6s). Right: corresponding spatial map of AoP-selective and bar-selective regions, based on responses to stimuli shown below. (**A'**) As in (**A**), for MeTu neurons recorded in the right AOTU in the same animal. (**B**) GCaMP activity in lateral MeTu neurons showing responses to different angles of polarization. Each trace shows the mean of ROIs at equivalent positions in three different animals (one ROI per animal). (**B'**) As in (**B**), for MeTu neurons recorded in the right AOTU in the same animals. (**C**) Normalized tuning curves for responses shown in (**B**). Mean ± SEM. (**C'**) As in (**C**), for MeTu neurons recorded in the right AOTU in the same animals. (**D**) Responses of MeTu neurons in medial positions to an unpolarized blue bar oscillating in five positions in the frontal visual field. Traces of the same color show the mean of three ROIs in equivalent positions in the AOTU in three different animals (one ROI per animal). Bar positions 1 and 5 correspond to ±90° azimuth in the contralateral visual field for recordings in the right (**D'**) and left (**D**) AOTU, respectively. (**D'**) As in (**D**), for MeTu neurons recorded in the right AOTU in the same animals. Arrowhead indicates the response of an ROI in a lateral position (green) with similar responses to the bar stimulus at position 4. (**E**) Proposed mapping of azimuthal position in visual field to vertical position in AOTU, based on (**D**). (**E'**) As in (**E**), for the right AOTU. (**F**) Responses of MeTu neurons in lateral positions to a sparse dot-field expansion pattern presented in three regions of the frontal visual field. Traces of the same color show the mean of three ROIs in equivalent positions in the AOTU in three different animals (one ROI per animal). (**F'**) As in (**F**), for MeTu neurons recorded in the right AOTU in the same animals.

(*Figure 4C*). Within an intermediate band of MeTu terminals (likely corresponding to AOTUla, blue ROIs, *Figure 4A*), left, we observed clear responses to bars in ipsilateral-frontal and frontal positions, with the more frontal position represented dorsally in the AOTU on both sides of the brain (*Figure 4D*). In the ventral AOTU, we found responses to bars presented in the contralateral-lateral visual field (±90° azimuth), outside the field of view of the ipsilateral eye (*Figure 4D,E*). Together, these results suggest that retinotopic representations of both visual space and angles of polarization

are found within different regions of the lateral and intermediate-lateral domains of the AOTU (*Figure 4C,E*).

Furthermore, these regions do not appear to be mutually exclusive (white regions, *Figure 4A*, right), and we occasionally observed responses to both polarized and unpolarized stimuli at the same location (green trace, *Figure 4D'*). In addition, we presented a sparse dot-field pattern simulating forward thrust optic-flow to test whether MeTu neurons were stimulated by other unpolarized visual features, which may also be processed downstream of the AOTU in central neurons. Optic-flow sensitive neurons have been found in the central complex of bees, for example, and similar stimuli with only a single pixel of light are sufficient to drive central complex-mediated navigation behavior in *Drosophila* (*Giraldo et al., 2018*; *Stone et al., 2017*). Here, we found that MeTu neurons were indeed stimulated by a sparse dot-field pattern presented at different locations (*Figure 4F*). Responses were found in regions which were also modulated by the polarizer (green ROIs, *Figure 4A*, left), further highlighting the range of visual features represented within a given region of the AOTU.

## Inter-tubercle neurons conveying polarization signals to the contralateral AOTU

Evidence from other insects suggested that we might find bilateral, inter-tubercle neurons which, if in contact with MeTu neurons, could be conveying the responses we observed in the AOTU to contralateral stimuli (*Heinze et al., 2013*; *Pfeiffer et al., 2005*; *Pfeiffer and Kinoshita, 2012*). We used the MeTu driver R73C04-Gal4 to drive *trans*-Tango and reveal putative postsynaptic neurons in the AOTU (*Figure 5A*). We found clear labeling of a population of neurons projecting to the bulb which resembled the tubercular-bulbar (TuBu) neurons (*Omoto et al., 2017*; *Figure 5A'*), in addition to labeling of the inter-tubercle tract (ITT) (*Strausfeld, 1976*; *Figure 5A''*), suggesting inter-hemispheric signaling postsynaptic to MeTu neurons in the AOTU. We then identified a Gal4 driver (R17F12-Gal4) that is expressed by a population of two tubercular-tubercle (TuTu) neurons per brain hemisphere, with axonal projections to the contralateral AOTU via the ITT (*Figure 5B*). Within the AOTU, these TuTu neurons predominantly innervate the intermediate-lateral domain (AOTUil) (*Figure 5B'*). TuTu neurons with similar morphology were reconstructed in a recent connectome analysis of the *Drosophila* central brain (TuTuB) (*Scheffer et al., 2020*). As with the genetic driver line used here (*Figure 5B*), two instances of TuTuB neurons were found per brain hemisphere. The TuTuB neurons are postsynaptic to partially-reconstructed MeTu neurons in the AOTU (MC61), corroborating our *trans*-Tango results (*Figure 5A''*), and we suggest that they likely correspond to the TuTu neurons investigated here (see *Appendix 1—table 1* for correspondence between neurons in genetic drivers used and neurons in the hemibrain connectome).

### TuTu neuron responses

We recorded presynaptic calcium activity in the terminals of contralateral TuTu neurons in the AOTU (Figure 5C). Unexpectedly, we did not find responses to the unpolarized bar stimuli at any of the positions tested (*Figure 5C*), indicating that these TuTu neurons likely do not mediate the contralateral responses we observed in the MeTu neurons (*Figure 4D*). Rather, we found that the TuTu neurons were polarization-sensitive with PSI values similar to those of the MeTu neurons (*Figure 5E*), and tunings to a limited range of polarization angles (~30˚) centered around a near-horizontal orientation (*Figure 5F*). Although the tunings we recorded in TuTu neurons did not match any of the polarotopic mappings we found in MeTu neurons, and are insufficient alone to explain the inverse polarotopy occasionally observed (*Figure 3—figure supplement 1E,E'*), their connectivity nevertheless indicates a role in shaping MeTu responses: TuTuB neurons found in the hemibrain connectome appear to be pre-synaptic to MeTu neurons (*Scheffer et al., 2020*).

Hence, the anatomy, polarization-sensitivity, and number of the population of TuTu neurons investigated here suggests that they may correspond to the TuTu1 neurons described in locusts, although their preferred AoPs differ (*Pfeiffer et al., 2005*). TuTu1 neurons in the locust have also been shown to respond to unpolarized visual stimuli, whereas those investigated here did not, however TuTu1 responses were also selective for both spatial position and color, and the unpolarized stimuli presented here are not directly comparable (*Pfeiffer and Homberg, 2007*). The specificity of TuTu1 responses is thought to reflect their role in time-compensated processing of polarized light

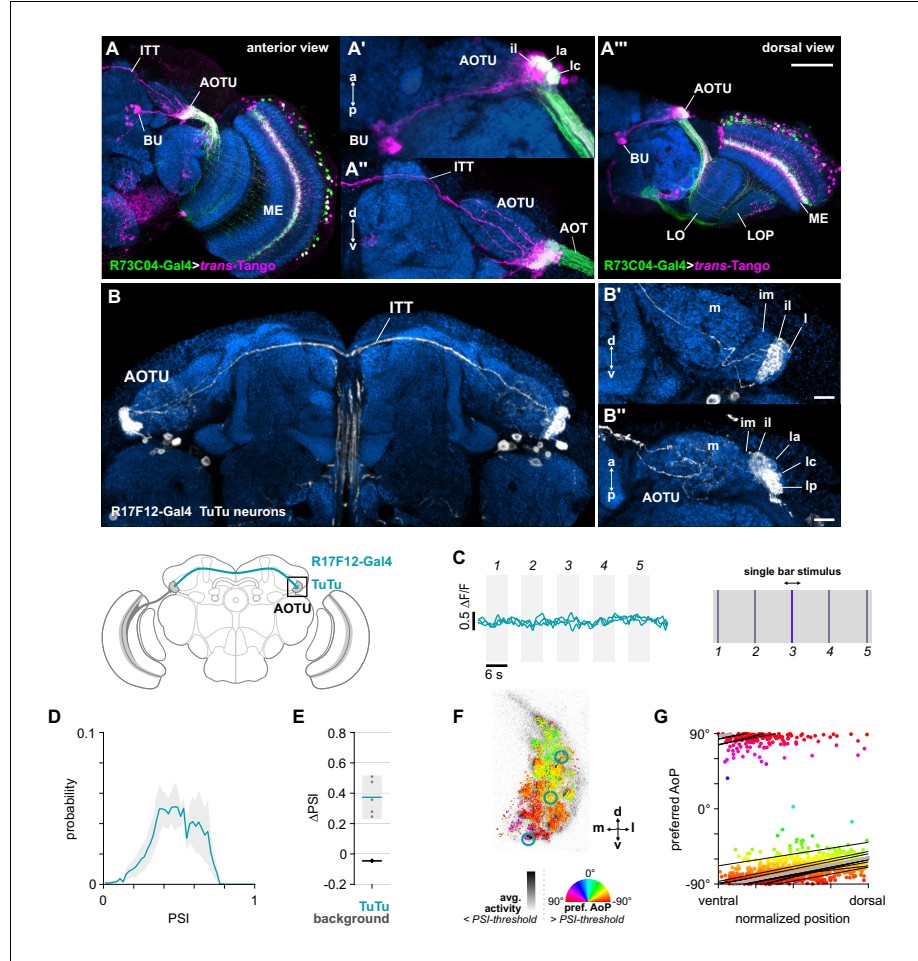

**Figure 5.** Bilateral TuTu neurons convey polarization signals to the contralateral AOTU. (**A**) Confocal projection (anterior view) of *trans*-Tango signal (magenta) labeling putative postsynaptic partners of R73C04-Gal4 MeTu neurons (green). (**A'**) High-magnification dorsal view highlighting TuBu neurons projecting from AOTU to bulb (BU). (**A''**) High-magnification anterior view highlighting projections to contralateral AOTU. (**A'''**) Dorsal view. Scale bar denotes 50 μm. (**B**) Confocal projection (anterior view) of TuTu neuron expression pattern (R17F12-Gal4>GFP). High-magnification anterior (**B'**) and dorsal (**B''**) views. Scale bars denote 10 μm. (**C**) Responses of presynaptic TuTu neurons to an unpolarized blue bar oscillating in five positions in the frontal visual field. Traces show the mean of three ROIs in equivalent positions in the contralateral AOTU (see **F**) in three different animals. (**D**) Probability distribution of PSI values in TuTu neurons (average PSI TuTu: 0.48, CI95 [0.34 0.63], N = 5 animals). Mean ± SEM. (**E**) Effect of polarizer on median PSI values versus controls with polarizer removed, within TuTu neurons (light dots) and background regions (dark dots) in individual animals (TuTu, blue line: mean ΔPSI = 0.37, CI95 [0.23 0.52], N = 5, p=0.02 t-test; background, black line: mean ΔPSI = −0.045, CI95 [−0.05 –0.04], N = 5, p < $10^{-4}$ t-test). (**F**) Example polarization tuning map for above-threshold pixels in the terminals of R17F12 TuTu neurons in a single imaging plane (R17F12-Gal4>sytGCaMP6s). (**G**) Scatter plot showing the common polarotopic organization of R17F12 TuTu neurons. Thin lines show linear-circular fits for data from individual animals with significant correlations (mean individual ρ = 0.64, CI95 [0.54 0.72], N = 5 animals), thick line shows fit for all pooled data (pooled ρ = 0.55, CI95 [0.39 0.68], N = 5 recordings, p < $10^{-4}$ permutation test).

signals and the integration of information about the position of the sun and spectral content of the sky.

## Tubercle projection neurons conveying polarization signals from the AOTU to the bulb

Next, we focused on the TuBu neurons and asked whether they receive polarization signals in the lateral (l) and intermediate-lateral (il) domains of the anterior optic tubercle (AOTU), as suggested by

*trans*-Tango labeling from polarization-sensitive MeTu neurons (*Figure 5A*). We examined three populations of TuBu neurons, grouped according to the region of the bulb (BU) they project to: superior (TuBu$_s$), inferior (TuBu$_i$), and anterior (TuBu$_a$) (*Figure 6A*). The dendrites of TuBu neurons in each population have also been shown to predominantly innervate stereotypical domains of the AOTU (*Omoto et al., 2017*; *Figure 6A*). We recorded calcium activity using Gal4 drivers for each population, noting that the driver for superior bulb-projecting TuBu$_s$ neurons (R88A06-Gal4) is also expressed in TuBu$_a$ neurons, and these groups could not be easily separated in recordings of the AOTU. For each genetic driver, we also identified what we believe to be the corresponding groups of TuBu neurons in the hemibrain connectome (*Appendix 1—figure 1*).

## TuBu neurons receive polarization signals in the lateral and intermediate-lateral AOTU

Among the dendrites of TuBu neurons recorded in the AOTU, we found that the populations innervating the AOTUl and AOTUil domains (TuBu$_s$ and TuBu$_a$, respectively) contained high PSI values that indicated strong modulation by the polarizer (*Figure 6B*), with average values significantly higher than the background regions of recordings (*Figure 6C*). In contrast, dendrites innervating the AOTUim domain (TuBu$_i$) contained PSI values not greater than 0.5 (*Figure 6B*) and, on average, were indistinguishable from background regions (*Figure 6C*). We typically found very few pixels with above-threshold PSI values in recordings of TuBu$_i$ dendrites (*Figure 6D*) and across all recordings we did not find a common relationship between the preferred angle of polarization (AoP) of TuBu$_i$ neurons and their ventral-dorsal position within AOTUim (*Figure 6E*).

Within the joint population of TuBu$_s$ and TuBu$_a$ neurons (R88A06-Gal4), the lateral domain (AOTUl) containing TuBu$_s$ dendrites typically exhibited a mixture of below-threshold PSI values and a smaller proportion of above-threshold values (*Figure 6F*), whereas the more-medial AOTUil domain containing TuBu$_a$ dendrites consistently showed above-threshold PSI values (*Figure 6F*). Pooling data from both domains, the preferred AoP covered a range of angles from −90° to +90° and we found a common relationship between preferred AoP and ventral-dorsal position within the AOTU (*Figure 6G*). Correspondingly, dendritic regions specifically within the population of TuBu$_a$ neurons (R34H10-Gal4) contained entirely above-threshold PSI values (*Figure 6H*) and obeyed the same polarotopic organization (*Figure 6I*).

## Polarotopy of TuBu dendrites and MeTu terminals in the AOTU do not correspond

For the dendrites of TuBu$_a$ and TuBu$_s$ neurons, we found that the direction of polarotopy in the AOTU (a counter-clockwise rotation of preferred AoP from ventral to dorsal) matched the direction of polarotopy in the putatively presynaptic MeTu neurons. However, the relative positions of tunings along the ventrodorsal axis of the AOTU do not correspond directly. For example, in the dorsal half of the AOTU the preferred AoPs of MeTu terminals were in the range 0° to +90° (*Figure 3E,F*), whereas for TuBu$_a$ dendrites in the dorsal half of the AOTU preferred AoPs were in the range −90° to 0° (*Figure 6I*). If MeTu neurons are indeed presynaptic to TuBu neurons in the AOTU, this result suggests either inhibitory input from MeTu neurons, which would effectively shift the preferred AoP by 90°, or the integration of additional inputs from unidentified polarization-sensitive elements at the dendrites of TuBu neurons.

## TuBu responses in the three regions of the bulb

We next asked how responses of TuBu neurons are organized in the bulb (BU). As in other insects, the BU features giant synapses ('micro-glomeruli') formed by TuBu endings and their targets, the ring neurons. In *Drosophila*, the BU consists of three anatomical regions: superior (BUs), inferior (BUi), and anterior (BUa) (*Figure 6A*). We recorded presynaptic calcium activity in the micro-glomerular terminals of TuBu neuron populations that target each region. We first examined the prevalence of polarization-modulated activity, indicated by the polarization-selectivity index (PSI). Spatial maps of PSI values revealed that the majority of TuBu$_s$ neurons recorded in micro-glomeruli in the BUs contained low PSI values, and interspersed among them were micro-glomeruli with high PSI values (*Figure 7A*). The mixture of polarization-sensitive and insensitive micro-glomeruli is conveyed by the broad distribution, skewed towards zero, of PSI values found across all pixels recorded in the BUs

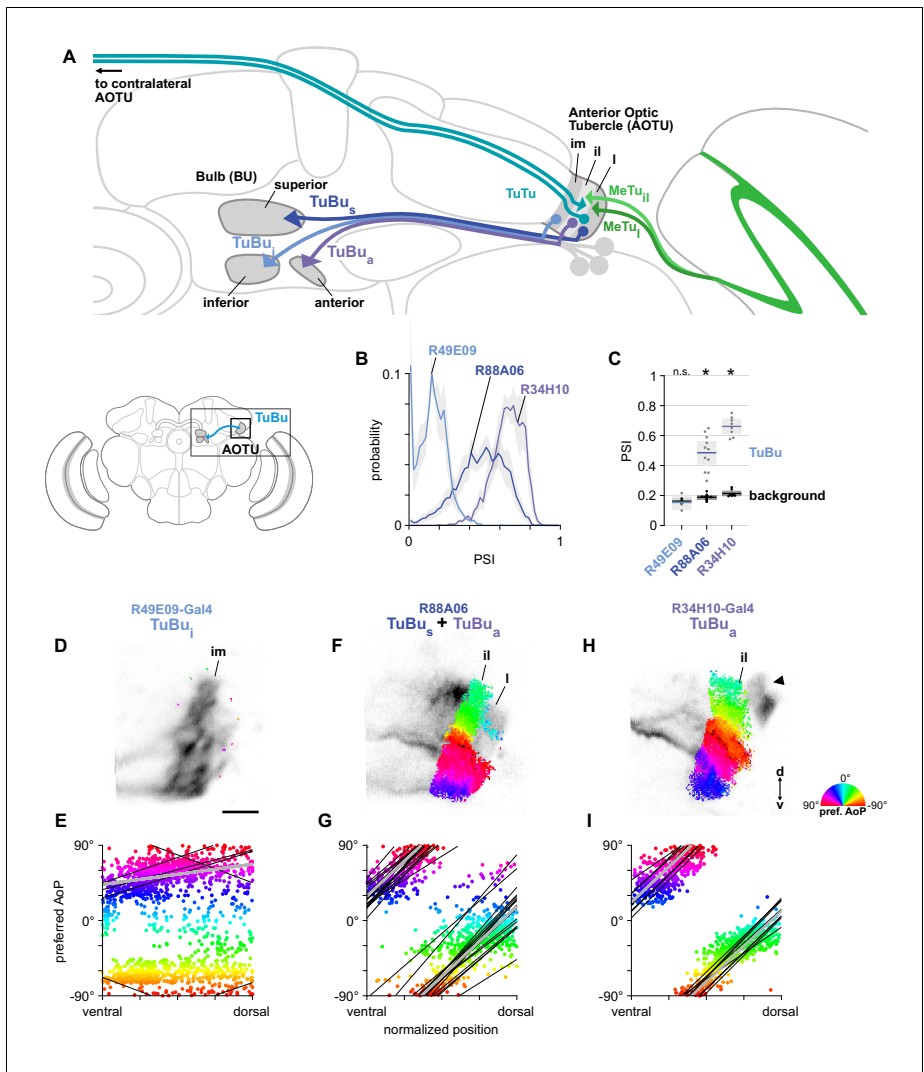

**Figure 6.** A population of TuBu neurons receives polarization signals in the AOTU. (**A**) Schematic of TuBu neuron types projecting to the bulb (BU) and connectivity in the AOTU. (**B**) Probability distribution of PSI values in TuBu neurons recorded in the AOTU. Mean ± SEM. Summarized in (**C**). (**C**) Average PSI values within TuBu neurons (light dots) and background regions (dark dots) in individual animals (R49E09 **TuBu$_i$** neurons: 0.15, CI95 [0.10 0.21], background: 0.16, CI95 [0.15 0.18], N = 5 animals, p=0.76 t-test; R88A06 **TuBu$_s$ + TuBu$_a$** neurons: 0.49, CI95 [0.41 0.56], background: 0.19, CI95 [0.18 0.20], N = 11 animals, p < $10^{-4}$ t-test; R34H10 **TuBu$_a$** neurons: 0.66, CI95 [0.61 0.71], background: 0.21, CI95 [0.20 0.23], N = 8 animals, p < $10^{-6}$ t-test). Shaded box denotes 95% confidence interval, asterisks denote statistically significant differences between neuron and background samples (p < 0.05, Bonferroni correction applied). (**D**) Example polarization tuning map for above-threshold pixels in the dendrites of TuBu$_i$ neurons in a single imaging plane (R49E09-Gal4>GCaMP6s). Below-threshold pixels display average intensity in grayscale. Scale bar denotes 5 μm. (**E**) Scatter plot showing the lack of polarotopic organization in TuBu$_i$ neurons. Individual points represent pixels recorded from TuBu$_i$ neurons, showing their normalized vertical position in the AOTU and their preferred angle of polarization (AoP). Thin lines show linear-circular fits for data from individual animals with significant correlations (mean individual ρ = 0.27, CI95 [−0.41 0.53], N = 4 animals; pooled data ρ = 0.16, CI95 [−0.32 0.53], N = 5 recordings, p < $10^{-4}$ permutation test). (**F**) As in (**D**), for a population containing TuBu$_s$ and TuBu$_a$ neurons (R88A06-Gal4>GCaMP6s). (**G**) As in (**E**), for the common polarotopic organization in TuBu$_s$ and TuBu$_a$ neurons (mean individual ρ = 0.74, CI95 [0.55 0.81], N = 11 animals; pooled data ρ = 0.12, CI95 [0 0.21], N = 11 recordings, p < $10^{-4}$ permutation test). (**H**) As in (**D**), for TuBu$_a$ neurons (R34H10-Gal4>GCaMP6s). Arrowhead indicates cell bodies excluded from analysis. (**I**) As in (**E**), for the common polarotopic organization in TuBu$_a$ neurons (mean individual ρ = 0.72, CI95 [0.46 0.89], N = 8 animals; pooled data ρ = 0.64, CI95 [0.03 0.89], N = 8 recordings, p < $10^{-4}$ permutation test).

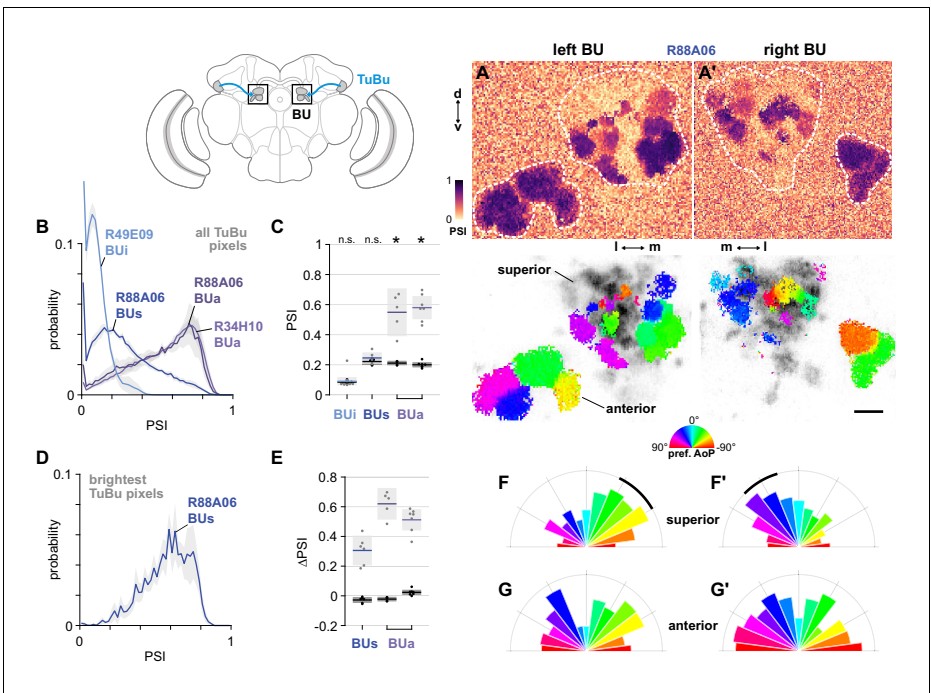

**Figure 7.** The anterior bulb is an entry point for polarization signals into the central complex. (A) Example spatial maps of polarization-selectivity index (PSI, top) and tuning (bottom) in TuBu neuron output micro-glomeruli in the superior and anterior regions of the left (A) and right (A′) bulbs in an individual fly (R88A06-Gal4>sytGCaMP6s). Scale bar denotes 5 μm. (B) Probability distribution of PSI values in all pixels recorded in TuBu neurons in the three regions of the bulb (BU). Mean ± SEM. Summarized in (C). (C) Average PSI values within TuBu neurons in the BU (light dots) and background regions (dark dots) in individual animals, including responsive and non-responsive (below PSI-threshold) pixels (**BUi** neurons: 0.09, CI95 [0.06 0.12]; background: 0.09, CI95 [0.08 0.09], N = 12 animals, p=0.68 t-test; **BUs** neurons: 0.25, CI95 [0.21 0.28], background: 0.22, CI95 [0.20 0.24], N = 6 animals, p=0.18 t-test; **BUa** (R88A06) neurons: 0.55, CI95 [0.39 0.71], background: 0.21, CI95 [0.20 0.23], N = 5 animals, p=0.0048 t-test; (BUa) (R34H10) neurons: 0.59, CI95 [0.50 0.66], background: 0.20, CI95 [0.18 0.22], N = 7 animals, $p < 10^{-4}$ t-test). Shaded box denotes 95% confidence interval, asterisks denote statistically significant differences between neuron and background samples (p < 0.05, Bonferroni correction applied). (D) Probability distribution of PSI values in 10% brightest pixels recorded in TuBu$_s$ neurons in BUs. Mean ± SEM. Summarized in (E). (E) Effect of polarizer on median PSI values versus controls with polarizer removed, within TuBu neurons (light dots) and background regions (dark dots) in individual animals (mean ΔPSI **TuBu$_s$** neurons: 0.31, CI95 [0.21 0.41], N = 6, p=0.0005 t-test, background: 0.03, CI95 [−0.05 −0.01], N = 6, p=0.02, t-test; **TuBu$_a$** neurons (R88A06, center column): 0.62, CI95 [0.51 0.73], N = 5, $p < 10^{-4}$ t-test, background: −0.02, CI95 [−0.04 −0.01], N = 5, p=0.18, t-test; **TuBu$_a$** neurons (R34H10, right column): 0.51, CI95 [0.44 0.58], N = 7, $p < 10^{-5}$ t-test, background: −0.02, CI95 [0.01 0.04], N = 7, p=0.19, t-test). Shaded box denotes 95% confidence interval. (F) Polar histogram of preferred angles of polarization in TuBu$_s$ neurons recorded in the left (F) and right (F′) superior bulb. Normalized probabilities in each bin are displayed as area of wedge; radial lengths of wedges not directly comparable. Arc denotes mean resultant angle ±95% confidence interval (**TuBu$_s$** left: 0.35, −43.0°, CI95 ±17.3°, N = 4, p=0.003 Rayleigh uniformity test; **TuBu$_s$** right: 0.35, 30.8°, CI95 ±14.9°, N = 5, p=0.0005 Rayleigh uniformity test). (G) As in (F), for TuBu$_a$ neurons recorded in the anterior bulb (R34H10) (**TuBu$_a$** left: 0.08, −60.2°, CI95 N/A, N = 6, p=0.61 Rayleigh uniformity test; **TuBu$_a$** right: 0.14, 66.12°, CI95 N/A, N = 6, p=0.22 Rayleigh uniformity test).

The online version of this article includes the following figure supplement(s) for figure 7:

**Figure supplement 1.** Unpolarized flash responses in TuBu neurons.

(*Figure 7B*). In contrast, the narrow distribution of PSI values close to zero in BUi micro-glomeruli demonstrates the absence of polarization-sensitive TuBu$_i$ neurons (*Figure 7B*). Finally, we found that all TuBu$_a$ neurons recorded exhibited high PSI values in the BUa (*Figure 7A,B*), in two Gal4 drivers. Average PSI values in the BUa were greater than 0.5 in both drivers (*Figure 7C*), while in the BUi and BUs, the average PSI values were not significantly different from the average in background regions of recordings, typically around 0.2 (*Figure 7C*).

We further explored the PSI values in the BUs by isolating the brightest pixels in TuBu$_s$ neurons in each recording, which were likely to represent active neurons (*Figure 7D*). We found that the distribution of PSI values among the brightest pixels was shifted towards one and was qualitatively different to the distribution across all pixels (*Figure 7B,D*). We then compared the average PSI value of the brightest pixels in the BUs with their average value in control experiments with the polarizer removed, and repeated this procedure with the brightest pixels in the BUa as a reference. Among active pixels in both the BUs and BUa we found a significant effect of the polarizer on PSI values versus controls, with the effect size larger in the latter (*Figure 7E*). In sum, we found polarized light responses in TuBu neuron output micro-glomeruli in both the superior and anterior bulb, and no appreciable responses to polarized light in TuBu neuron outputs in the inferior bulb. We interpret these findings as being consistent with the corresponding dendritic responses of TuBu neurons in the AOTU (*Figure 6B*).

## Encoding of polarized light and other visual features in the bulb

We then asked whether the information about polarized light available in the BUs and BUa differed in some way, for example by encoding different ranges of angles. We observed that a cluster of micro-glomeruli toward the medial edge of the superior bulb tended to show preferential responses to similar angles of polarization (AoP) (*Figure 7A*, bottom). When we examined the distribution of preferred AoPs in the BUs we found a non-uniform distribution with the highest frequency of preferred AoPs around −45° in the left bulb (*Figure 7F*) and +45° in the right bulb (*Figure 7F'*). Conversely, in the anterior bulb (BUa) on both sides we found an approximately uniform representation of preferred AoPs in TuBu$_a$ neurons (*Figure 7G,G'*). Preferential responses to certain AoPs therefore appear to be more common than others in the BUs. We expected that a uniform representation of the full range of polarization space would be necessary for decoding heading direction from skylight polarization patterns. The over-representation of certain AoPs in BUs micro-glomeruli resembles a detector for a particular feature, such as horizontally polarized reflections from the surface of water, rather than the main input to a system for polarized light-based navigation.

Without further investigation of the downstream circuitry in the central complex that processes these polarized light inputs from the bulb, our interpretations of possible functional roles are somewhat speculative. For the neuron populations examined here, our light microscopy images and the hemibrain connectome data (*Scheffer et al., 2020*) show that each TuBu neuron (BU input) connects exclusively with one ring neuron (BU output), suggesting that the distribution of preferred AoPs in the TuBu populations should be homogeneously sampled by populations of postsynaptic ring neurons. However, due to the physical shape of the bulb and the organization of micro-glomeruli within it, combined with the relatively low resolution of our functional imaging experiments, we could not ascertain whether all TuBu neurons were individually separable and their spatial locations accurately captured within the imaging volume, and these issues could have affected the uniformity of the distributions that we observed. Nevertheless, we found no clear linear organization of preferred AoPs in either the BUs or the BUa, a marked contrast to the consistent, linear organization in TuBu dendrites in the AOTU (*Figure 6H,I*). Circular, rather than linear, organizations of neurons in the BU have been proposed (*Timaeus et al., 2020*) and we explore these in the BUa in the next section (*Figure 8—figure supplement 1*).

TuBu neurons have previously been shown to respond to unpolarized visual stimuli presented to regions of the eye outside the DRA, including the narrow vertical bars which we found to stimulate MeTu neurons (*Omoto et al., 2017*; *Shiozaki and Kazama, 2017*; *Sun et al., 2017*; *Figure 4D*). We also recorded the responses of TuBu neurons to these stimuli (data not shown, included in dataset). As a simple, representative way to compare the responses of the three groups of TuBu neurons to unpolarized stimuli, we present the responses of each TuBu population to a wide-field flash of unpolarized blue light, recorded in both the AOTU and BU (*Figure 7—figure supplement 1A*). TuBu$_s$ and TuBu$_i$ neuron populations showed responses to the flash in the AOTU and, more strongly, in the BU, while TuBu$_a$ neurons recorded in either neuropil were inhibited by the unpolarized light stimulus (*Figure 7—figure supplement 1B*). We note that a previous study appeared to show excitation of BUa micro-glomeruli in response to unpolarized small-field stimuli presented in the contralateral visual field and inhibition in response to ipsilateral stimuli (*Shiozaki and Kazama, 2017*). These

results may reflect excitatory and inhibitory receptive fields of TuBu$_s$ neurons, while our recordings indicate that inhibition dominates the response of the population to wide-field visual stimuli.

## Polarization signals entering the central complex via the bulb

Taken together, our recordings of TuBu neurons indicate that polarized light signals are potentially delivered to the central complex via two parallel pathways: one through the superior bulb (BUs), containing a limited representation of polarization space in addition to other visual information, and a second channel through the anterior bulb (BUa). In the bulb, TuBu neuron presynaptic terminals innervate the globular dendrites of ring neurons (ER) in a largely one-to-one fashion, forming individual micro-glomeruli. ER ring neurons project medially to the ellipsoid body (EB) (*Figure 8A*), where their arborizations have a circular form and are both dendritic and axonal (*Figure 8B*; *Hanesch et al., 1989*; *Omoto et al., 2018*). We recorded calcium activity in the dendrites of two populations of ring neurons in the bulb, which we expected would include postsynaptic partners of polarization-sensitive TuBu neurons: one population innervating the medial two-thirds of the BUs, containing ER2 neurons (R19C08-Gal4), and one innervating the BUa, containing ER4m neurons (R34H10-Gal4) (*Figure 8A*). Both ER2 and ER4m ring neuron populations target the outer central domain of the EB, albeit following different trajectories (*Figure 8A,B*; *Omoto et al., 2018*; *Omoto et al., 2017*). These populations have previously been referred to as R2 and R4m. For each genetic driver, we identified what we believe to be the corresponding groups of neurons in the hemibrain connectome and have adopted the ER nomenclature for consistency (*Appendix 1—table 1*).

### Ring neurons receive polarization signals from TuBu neurons

As with TuBu$_s$ micro-glomerular outputs, we found that only a subset of ER2 neurons in the BUs were modulated by polarized light, with above-threshold PSI values typically in a medial cluster with a preferred angle of polarization (AoP) around 45° (*Figure 8C*). Low PSI values were common throughout the ER2 population and average values were not significantly different from average values in background regions (*Figure 8D*). We attribute this to a small proportion of polarization-sensitive ring neurons within a driver which expresses in a population of largely insensitive neurons. By contrast, in ER4m neurons in the BUa, average PSI values were greater than 0.5 (*Figure 8D*) and the overall distribution of values in the population was similar in shape to the distribution in TuBu$_a$ neurons (*Figure 6B*, *Figure 7B*). We found that the polarizer had a significant effect on PSI values of ER4m neurons versus controls with the polarizer removed (*Figure 8E*). Furthermore, we found that the dendrites of individual ER4m neurons exhibited distinct preferences for AoP in each recording (*Figure 8G*). Since ER4m neurons appear to receive monosynaptic input from TuBu neurons, we conclude that they almost certainly acquire their polarization-tuned responses from the presynaptic TuBu$_a$ neurons in the BUa (*Figure 8A,B,F*). We note that the average PSI value decreased from TuBu$_a$ neurons to ER4m neurons (*Figure 8G',H'*, *Appendix 1—figure 1B*) and we further explore the transformation of their signals in the next section. Although the BUs appears to contain polarization-sensitive elements, they are pervasive neither in the populations of BUs-projecting ring neurons we recorded, nor their putative presynaptic partners, the TuBu$_s$ neurons, and hereafter we focus on the BUa, where all micro-glomeruli seem to be involved in polarization processing.

### Polarotopy in the anterior bulb

In contrast to the linear polarotopic organization of tunings observed in the AOTU, which was consistent across animals (*Figure 6F,H*), the spatial organization of polarization tunings in the BUa was less clear (*Figure 8G,H*). We tested whether there was a common relationship between the horizontal (medial-lateral), vertical (ventral-dorsal), or angular position of micro-glomeruli within the BUa and their preferred AoP, for both TuBu$_a$ and ER4m neurons (*Figure 8—figure supplement 1*). We also considered whether there was a relationship within a population of neurons in an individual animal which was not common across animals. We found no indication of a relationship between position and preferred AoP except in recordings of TuBu$_a$ neurons in the left BUa, which showed a common vertically organized polarotopy (*Figure 8—figure supplement 1C*) and circularly organized polarotopies in individual animals (*Figure 8—figure supplement 1E*). However, we found no significant polarotopy in the corresponding TuBu$_a$ neurons in the right BUa, or in postsynaptic ER4m neurons.

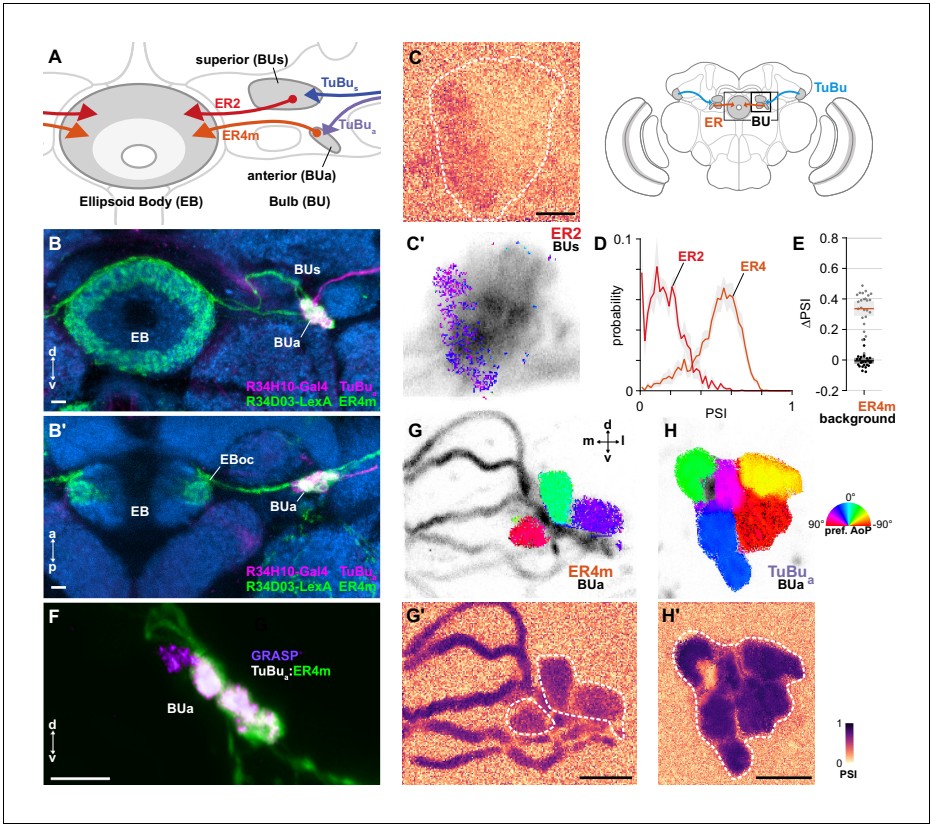

**Figure 8.** ER4m ring neurons receive polarization-tuned responses from TuBu neurons. (**A**) Schematic of TuBu and ring neuron (ER) connectivity in the bulb (BU). (**B**) Confocal projection (anterior view) of dual-labeled TuBu$_a$ neurons (R34H10-Gal4>RFP, magenta) and ER4m neurons (R34D03-Gal4>GFP, green), indicating overlap in anterior bulb (BUa). (**B'**) Dorsal view. Scale bars denote 5 µm. (**C**) Example spatial maps of polarization-selectivity index (PSI) and polarization tuning (**C'**) for ER2 dendrites recorded in the superior bulb (R19C08-Gal4>GCaMP6s). Scale bar denotes 5 µm. (**D**) Probability distributions of PSI values in ring neurons recorded in the bulb (average PSI **ER2** neurons: 0.17, CI95 [0.09 0.24], background: 0.20, CI95 [0.16 0.24], N = 4 animals, p=0.29 t-test; **ER4m** neurons: 0.51, CI95 [0.46 0.55], background: 0.22, CI95 [0.20 0.24], N = 25 animals, p < 10$^{-6}$ t-test). Mean ± SEM. (**E**) Effect of polarizer on median PSI values versus controls with polarizer removed, within ER4m neurons (light dots) and background regions (dark dots) in individual animals (mean ΔPSI **ER4m** neurons: 0.34, CI95 [0.29 0.38], N = 25, p < 10$^{-6}$ t-test, background: −0.05, CI95 [−0.02 0.01], N = 25, p=0.58, t-test). (**F**) Confocal projection (anterior view) of activity-dependent synaptic GRASP (GFP reconstitution across synaptic partners) signal between presynaptic TuBu$_a$ and postsynaptic ER4m neurons in anterior bulb (BUa). Scale bar denotes 5 µm. (**G**) Example polarization tuning map and corresponding PSI map (**G'**) in ER4m dendrites in BUa (R34D03-Gal4>GCaMP6s). Pixels falling outside an ROI drawn around the neurons of interest (shown in G'), show average intensity in grayscale. Clusters of similar colors correspond to individual polarization-tuned micro-glomeruli. Individual axons projecting medially to the EB are visible leaving the left side of the image. Scale bar denotes 5 µm. (**H**) As in (**G**), for TuBu$_a$ output micro-glomeruli at an approximately corresponding location in BUa (R34H10-Gal4>sytGCaMP6s). Clusters of similar colors correspond to individual polarization-tuned micro-glomeruli.

The online version of this article includes the following figure supplement(s) for figure 8:

**Figure supplement 1.** Unstructured organization of preferred angle of polarization in the anterior bulb.

Hence we cannot firmly conclude that either a vertical or circular organization of tunings exists in the anterior bulb. Furthermore, our assessment of circular organization is only valid for the dorso-posterior imaging plane used here, and we cannot exclude the possibility of a circular organization around a different axis of the bulb.

## Polarization processing in the ellipsoid body and protocerebral bridge

We next wanted to understand how polarized light signals are represented in the ellipsoid body (EB), where the tangential ring neurons supply visual information around its circular structure. Ring neurons interact bidirectionally with columnar E-PG neurons (*Omoto et al., 2018*), which have been shown to flexibly encode heading direction relative to visual landmarks (*Fisher et al., 2019*; *Kim et al., 2019*; *Seelig and Jayaraman, 2015*).

### ER4m responses in the EB

We recorded the synaptic terminals of the population of ER4m neurons in the EB (approximately ten neurons, five per brain hemisphere, R34H10-Gal4). As expected from recordings in the dendritic regions of ER4m in the anterior bulb (BUa), we observed modulation of their activity with rotations of the polarizer, indicated by their polarization-selectivity index (PSI) (*Figure 9A*). We noted that in some recordings, above-threshold PSI values were spatially localized to approximately one quadrant of the EB (*Figure 9A,B*, top, arrowhead). This observation, although not explored thoroughly, indicates localized calcium dynamics within the arborization trees of the ring neurons and a non-homogenous distribution of output signaling. ER4m neurons are both pre- and postsynaptic to E-PG neurons in the EB, and the plasticity of their connections is thought to underlie the workings of a flexible representation of heading direction by influencing inhibition and restricting activity to approximately one quadrant of the EB (*Fisher et al., 2019*; *Kim et al., 2019*). Coincident excitation of ring neurons and a localized subset of E-PG neurons is thought to lessen this inhibition and increase the likelihood of activity at that location. Increased activity within a section of the EB, as we observed here, may facilitate this process.

By recording GCaMP6s localized to synapses, we were able to investigate the responses of individual terminals. Individual ER4m terminals exhibited tunings to distinct angles of polarization (AoP), and a range of tunings could be found intermingled at any given position in the EB (*Figure 9B*). Additionally, we found that in many recordings the preferred angles of polarization (AoPs) of terminals were similar to each other and did not represent a full range of AoPs. The range of AoPs found varied across animals (*Figure 9B*), and as a result the frequency of preferred AoPs formed a unimodal distribution centered on a different angle in each recording (*Figure 9C*). We verified that the non-uniform distribution of AoPs was not an artifact of our image projection across multiple planes and that a predominant preferred AoP was also observed from a single imaging plane through a section of the EB (*Figure 9A–C*, bottom).

### Populations of ER4m neurons exhibit a preferred angle of polarization in the EB

As a result of the non-uniform distributions of preferred AoPs, it followed that the average activity of an entire ER4m population in the EB exhibited modulation induced by the polarizer and a single preferred AoP could effectively be identified for each population (*Figure 9D*). This was unexpected, given that the upstream population of TuBu$_a$ neurons appeared to uniformly represent a full range of AoPs (*Figure 6H,I*, *Figure 7G*). To compare the representation of preferred AoPs in ER4m populations across animals, and with the AoPs represented in TuBu$_a$ neuron populations, we calculated the mean resultant vector of the preferred AoPs of all pixels within a recording, weighted by their individual PSI values (*Figure 9E*). The length of the vector gives an indication of the distribution of polarization tunings in a single recording, with a value of 1 indicating an identical preferred AoP in all pixels and a value of 0 indicating a uniform distribution of preferred AoPs.

For ER4m terminals in the EB we found population tuning vectors with lengths exceeding 0.74 and an average length of 0.51 across animals (*Figure 9E*), while for ER4m dendrites recorded in either the left or right BUa individually we found an average length of 0.39 (*Figure 9F*). For TuBu$_a$ populations recorded in either bulb we found that the vector lengths did not exceed 0.3 and the average length was 0.18 across animals (*Figure 9G*). This increasing vector length, first from TuBu$_a$ to ER4m in the bulb, then from the dendrites of ER4m in the bulb to the terminals in the EB, indicates a reduction in the range of AoPs found at each stage. Since uneven sizes or quantities of neurons could affect these results, we repeated the analysis with ROIs drawn on individual microglomeruli in the bulb. We found a comparable number of micro-glomeruli in recordings of TuBu$_a$

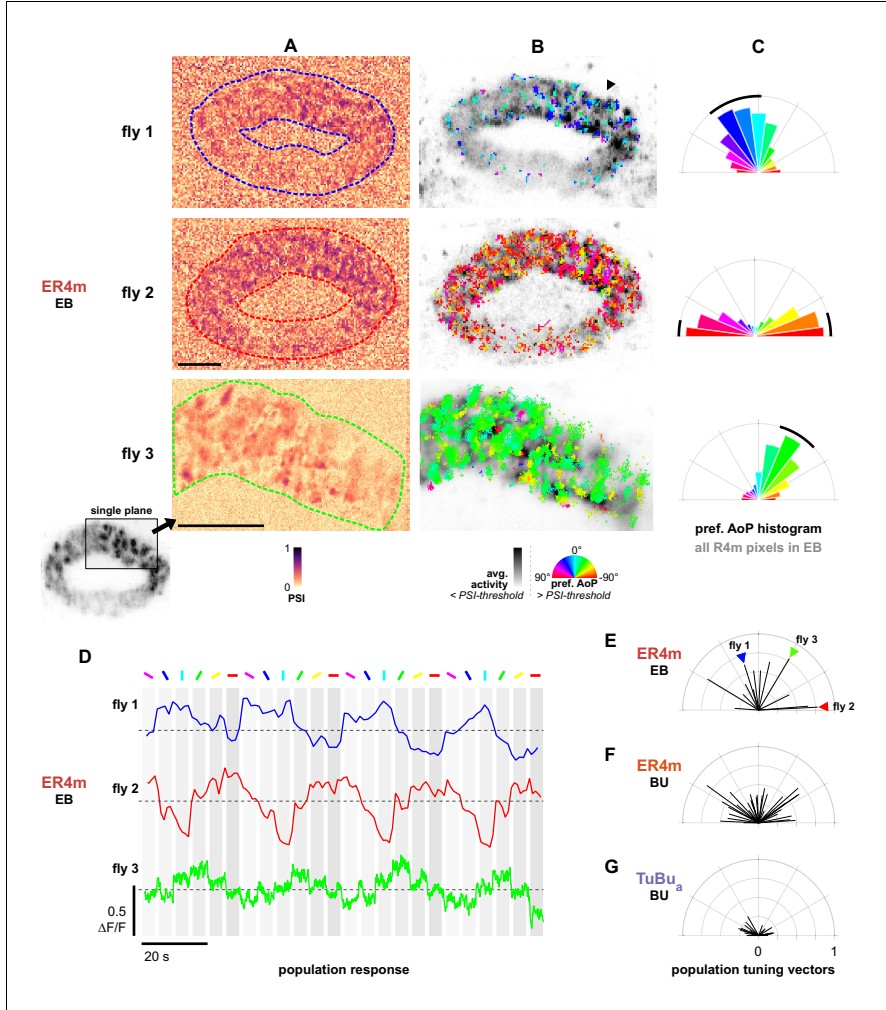

**Figure 9.** Populations of ER4m ring neurons exhibit a preferred angle of polarization. (A) Example spatial maps of polarization-selectivity index (PSI) in ER4m synapses recorded in the ellipsoid body (EB) (R34D03-Gal4>sytGCaMP6s). Data shown are from maximum-intensity projections through the EB (top, middle) or a single plane (bottom). (B) Example polarization tuning maps corresponding to recordings in (A). Pixels with a below-threshold PSI value, or falling outside an ROI drawn around the ER4m population, show average intensity in grayscale. Scale bars denote 10 μm. (C) Polar histograms of preferred angle of polarization in all pixels within the ROIs in (A). Normalized probabilities in each bin are displayed as area of wedge; radial lengths of wedges not directly comparable. Arc denotes mean resultant angle ±95% confidence interval (resultant length and angle **fly 1**: 0.57, 18.7°, CI95 ±16.6°, N = 4, p=0.002 Rayleigh uniformity test; **fly 2**: 0.72, −87.3°, CI95 ±15.0°, p=0.001 Rayleigh uniformity test; **fly 3**: 0.71, −31.6°, CI95 ±15.4°, p=0.001 Rayleigh uniformity test). (D) Average GCaMP activity in the ROIs in (A) in response to different angles of polarization. Dashed lines indicate 0 ΔF/F baseline. (E) Resultant tuning vectors for the population of all recorded ER4m synapses in the EB of individual animals (mean length, pixel-based: 0.51, SEM ±0.44, N = 7, p < 10⁻⁶ t-test). Arrowheads indicate data for examples in A–D. (F) Resultant tuning vectors for the population of all recorded ER4m neurons recorded in the left or right BU of individual animals (mean length, pixel-based: 0.39, SEM ±0.32, N = 25, p < 10⁻⁶ tailed t-test; ROI-based: 0.36, SEM ±0.46, N = 25, p=0.005 tailed t-test, 134 ROIs,>3 ROIs per BU). (G) Resultant tuning vectors for the population of all recorded TuBu_a neurons recorded in the left or right BU of individual animals (mean length, pixel-based: 0.18, SEM ±0.13, N = 7, p < 10⁻⁶ tailed t-test; ROI-based: 0.14, SEM ±0.15, N = 7, p=0.0002 tailed t-test, 101 ROIs,>3 ROIs per BU).

and ER4m neurons in the BUa, and the ROI- and pixel-based approaches both yielded a qualitatively similar result (*Figure 9F,G*).

These findings suggest that there is not an exact correlation between polarized light responses in the populations of presynaptic TuBu_a neurons and postsynaptic ER4m neurons in an individual

animal. In ER4m dendrites, the average strength of modulation is reduced compared to TuBu$_a$ neurons, illustrated by the decrease in PSI values (*Figure 8—figure supplement 1*, *Appendix 1—figure 1B*), and the distribution of tunings is less uniform (*Figure 9F,G*). In ER4m terminals in the EB, the distribution of tunings is less uniform still, hinting at subcellular processes which may impact ER4m signaling locally in the EB, a computational motif for which there is precedence both in the CX and in visual neurons generally (*Franconville et al., 2018*; *Turner-Evans et al., 2020*; *Yang et al., 2016*). As a consequence, it appears that the ensemble activity of ER4m synapses could convey a preferential response for a particular angle of polarization to columnar neurons at any location in the EB.

## Ring neuron and E-PG connectivity in the EB

We then asked whether columnar E-PG neurons (also referred to as 'compass' neurons) respond to polarized light cues. E-PG neurons are key elements in a network which maintains a neural representation of heading direction as a locus of activity, or 'bump', which changes position within the CX as the animal turns, like the needle of a compass (*Green et al., 2017*; *Seelig and Jayaraman, 2015*). In the previous literature, this activity bump has been observed in the ellipsoid body (EB), protocerebral bridge (PB), and fan-shaped body (FB), typically during walking or flight in restrained animals (*Giraldo et al., 2018*; *Shiozaki et al., 2020*). It has not been demonstrated in fully immobilized animals, hence we did not expect to see it here. Inputs to E-PG neurons from polarization-sensitive ER4m (or indeed ER2) neurons also represent only a fraction of the sensory input to the network (*Hulse et al., 2020*). Nevertheless, we hypothesized that E-PG activity could be modulated by a varying angle of polarized light since the same has been demonstrated in numerous columnar central complex neurons in other insects while immobilized (*Heinze and Homberg, 2007*; *Honkanen et al., 2019*). Moreover, the responses we observed in ER4m ring neurons (*Figure 9D*) suggested that the E-PG population should also exhibit tunings to a limited range of angles.

Ring neurons provide inhibitory input to E-PG neurons in the EB (*Figure 10A*), where interactions between ring and E-PG neurons are thought to be reciprocal (*Fisher et al., 2019*; *Kim et al., 2019*; *Omoto et al., 2018*). We confirmed connectivity between presynaptic E-PG neurons and postsynaptic ER4m neurons in the EB in the genetic drivers used (ER4m: R34D03-LexA, *Figure 8B*; E-PG: SS00096-Gal4, *Figure 10—figure supplement 1A*) by labeling synapses using activity-dependent GRASP (*Macpherson et al., 2015*; *Figure 10—figure supplement 1B*).

## E-PG organization and pairings in the PB

We opted to record calcium signals in the presynaptic terminals of E-PG neurons in the PB (*Figure 10A*) where they form 16 distinct glomeruli, each innervated by at least two E-PG neurons (*Wolff et al., 2015*). Due to their neighboring positions in the EB and connectivity with other neurons, the activity of E-PG neurons innervating the eight glomeruli in the left half of the PB is known to be coordinated with those in the eight glomeruli in the right half (*Figure 10—figure supplement 1D*), and on either side of the PB the ends are effectively wrapped (1L is continuous with 8L, 1R is continuous with 8R, as per the organization in the EB) (*Giraldo et al., 2018*; *Green et al., 2017*). We did not consider the lateral-most columns on either side of the PB (9L and 9R), which are skipped by the population of E-PG neurons: these columns are innervated by E-PGt neurons which receive significantly fewer inputs from ring neurons in the EB, where they entirely overlap with E-PG neurons projecting to PB columns 1L and 1R (*Hulse et al., 2020*; *Scheffer et al., 2020*). We therefore did not expect that their activity would represent a unique position within the PB for the purposes of assessing the organization of polarization responses.

To evaluate the most appropriate pairing of glomeruli from left and right columns in the PB, we cross-correlated the activity recorded from pairs of glomeruli using different schemes and found the normalized correlation coefficient as an indication of their similarity (*Figure 10—figure supplement 1E*). The pairing scheme following the logic 1L/1R, 8L/2R, 7L/3R, etc (*Figure 10—figure supplement 1D*) yielded the highest mean similarity across all pairs of glomeruli, which decreased with a sinusoidal profile as the distance between pairs increased (*Figure 10—figure supplement 1E*). This pairing confirms a scheme proposed based on anatomical connectivity (*Wolff et al., 2015*), but differs by one position from the proposed connectivity in the locust, where a pairing scheme corresponding to 1L/8R, 8L/1R, 7L/2R, etc. has previously been used to pool data (*Heinze and Homberg, 2009*). However, if the E-PGt neurons in 9L/9R are considered part of the same network and co-active with E-PG

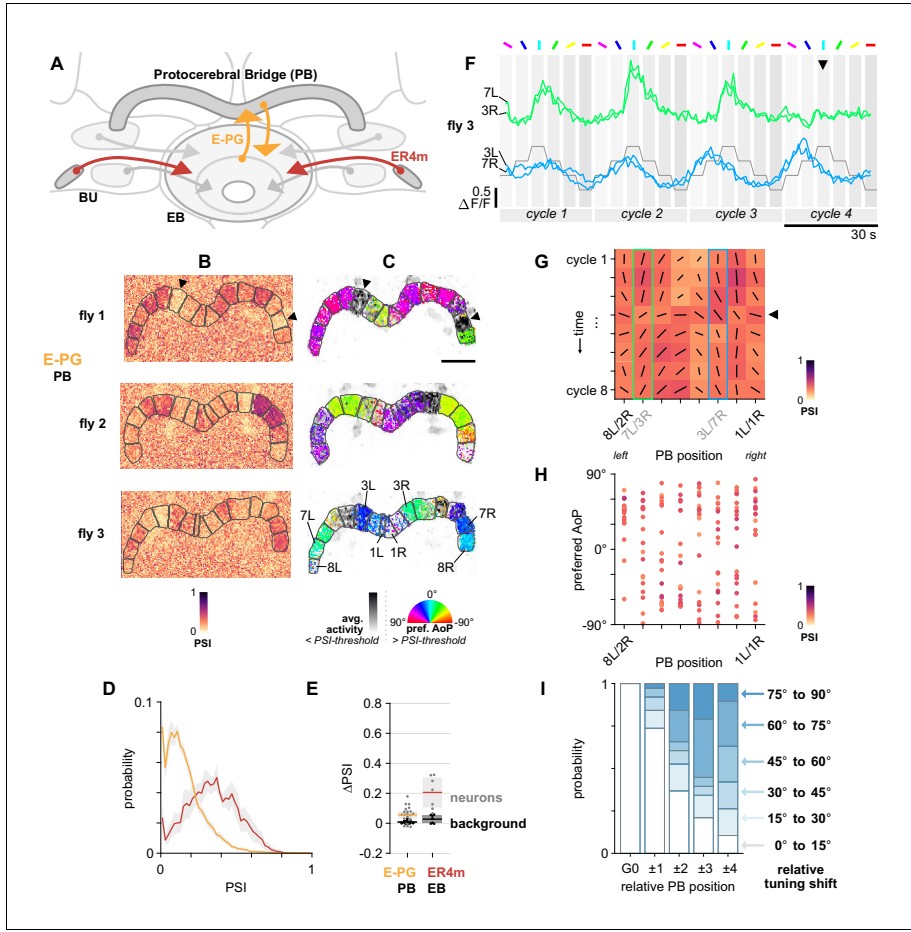

**Figure 10.** E-PG neurons respond to polarized light with flexible tuning and no fixed polarotopic map. (**A**) Schematic of E-PG columnar neuron projections and connectivity with tangential ring neurons in the ellipsoid body (EB). See also *Figure 10—figure supplement 1D*. (**B**) Example spatial maps of polarization-selectivity index (PSI) in E-PG synapses recorded in the protocerebral bridge (PB) (SS00096-Gal4>sytGCaMP6s). Data shown are from maximum-selectivity projections through the PB. ROIs (gray) demarcate glomeruli. (**C**) Example polarization tuning maps corresponding to recordings in (**A**). Pixels with a below-threshold PSI value, or falling outside an ROI drawn around the PB, show average intensity in grayscale. Scale bar denotes 25 μm. (**D**) Probability distributions of PSI values in E-PG neurons recorded in the PB and ER4m neurons recorded in the EB (average PSI E-PG neurons: 0.14, CI95 [0.12 0.17], background: 0.19, CI95 [0.19 0.20], N = 22 animals, p=0.0001 t-test; **ER4m** neurons: 0.34, CI95 [0.24 0.44], background: 0.21, CI95 [0.19 0.24], N = 7 animals, p=0.02 t-test). Mean ± SEM. (**E**) Effect of polarizer on median PSI values versus controls with polarizer removed, within E-PG and ER4m neurons (light dots) and background regions (dark dots) in individual animals (mean ΔPSI E-PG neurons: 0.06, CI95 [0.03 0.08], N = 22, p < $10^{-4}$ t-test, background: 0.01, CI95 [0.00 0.01], N = 22, p=0.0007, t-test; **ER4m** neurons: 0.21, CI95 [0.11 0.31], N = 7, p=0.002 t-test, background: 0.03, CI95 [0.00 0.05], N = 7, p=0.04, t-test). (**F**) Activity in two pairs of L/R ROIs in (**C**) (fly 3) in response to different angles of polarization. Arrowhead indicates position of expected peak. (**G**) Cycle-by-cycle characterization of E-PG responses across the PB in a single recording (**C,F**) (fly 3). Orientation of individual vectors represents preferred AoP, length represents PSI (grid spacing equals PSI of 1). Highlighted boxes indicate extended data for pairs shown in (**F**). Arrowhead indicates the same cycle as the arrowhead in (**F**). (**H**) Scatter plot showing position of paired E-PG glomeruli in the PB and preferred angle of polarization (AoP) (pooled data ρ = 0.23, N = 19 animals, p=0.006 permutation test, 152 ROIs, mean ROI PSI 0.34, SEM 0.06; five significant individual circular-circular correlations, mean ρ = 0.46, SEM 0.45). (**I**) Normalized probability of tuning shift magnitude with distance from the glomerulus with the highest PSI value (mean shift from position 2 to 3, p=0.21; position 3 to 4, p=0.65; position 2 to 4, p=0.08; all other pairs p < $10^{-3}$, N = 19 animals, 152 ROIs). See also *Figure 10—figure supplement 1F,G*.

The online version of this article includes the following figure supplement(s) for figure 10:

**Figure supplement 1.** E-PG neurons show inconsistent responses and variable tunings.

neurons in columns 1L/1R, this pairing scheme is essentially the same as in the locust (*Pisokas et al., 2020*).

## E-PG responses to polarized light

We found that E-PG activity in the PB was indeed modulated as the angle of polarization (AoP) rotated. We assigned PSI values to the pixels in each recording (*Figure 10B*) and calculated their preferred AoP (*Figure 10C*). As expected, the PSI values and preferred AoPs showed a bilateral coupling, with the right half of the PB (1R to 8R) resembling the left half (8L to 1L) (*Figure 10B,C*).

We observed that the distribution of PSI values was not homogenous across the PB, and high values typically clustered across a contiguous subset of 2–4 glomeruli, while low PSI values occurred throughout the remaining glomeruli (*Figure 10B*). In each animal, the preferred AoP was similar across the glomeruli in each cluster (*Figure 10C*). It should be noted that these clusters of high PSI values correspond to the regions of highest modulation over a period of minutes, not an instantaneous locus of intensity which moved across the PB (referred to as an activity 'bump') (*Giraldo et al., 2018*; *Green et al., 2017*). Indeed, glomeruli with high average intensities, signifying high levels of activity, often exhibited low PSI values (arrowhead, *Figure 10B,C*).

## Flexible polarization-tuning properties of E-PG neurons

Overall, we found substantially lower PSI values in E-PG neurons than in ER4m neurons (*Figure 10D*). We found a statistically significant effect of the polarizer on PSI values versus controls in both populations (*Figure 10E*), yet the effect size was small in E-PG neurons with an average ΔPSI of 0.06, and an average PSI value that was generally lower than in background regions of recordings (*Appendix 1–figure 1*). To explore this discrepancy, we examined the responses of individual glomeruli in the PB in response to cycles of the polarizer (*Figure 10F*). Here, in the PB, we observed characteristics which distinguished E-PG responses from those of all other polarization-sensitive elements that we recorded in the upstream pathway. First, the amplitude of responses was often found to be inconsistent over multiple rotation cycles of the polarizer (arrowhead, *Figure 10F*, top). Second, the peak response was often found to occur at different positions of the polarizer over multiple cycles (*Figure 10F*, bottom). For both of these response characteristics, variations were synchronized across the left and right PB glomerulus pair (*Figure 10F*). When we analyzed responses to individual cycles of the polarizer separately, these characteristics manifested as PSI values and preferred AoPs which varied over time (*Figure 10G*).

To quantify how variable these polarization-tuning properties were, we obtained a measure of synchronicity between E-PG activity and the polarizer stimulus. We examined the auto-correlation function of all individual glomerular responses and compared them with those of ER4m and TuBu$_a$ neurons recorded in the anterior bulb (BUa) (we could not perform the same test for ER4m responses in the EB since we could not distinguish individual neurons from each other). For E-PG neurons, we found that less than half of all glomeruli recorded exhibited a periodicity which matched the stimulus, while almost all ER4m and TuBu$_a$ neurons matched the stimulus (E-PG: 43.3%, ER4m: 98.4%, TuBu$_a$: 100%) (*Figure 10—figure supplement 1C*). Therefore, although E-PG activity was in some cases periodic and likely induced by the rotating angle of polarization, when observed over multiple cycles less than half of all E-PG glomeruli showed responses that were phase-locked with the stimulus. For comparison, the noisy fluctuations of activity recorded in E-PG neurons with the polarizer removed produced similar levels of synchronization for the majority of glomeruli (*Figure 10—figure supplement 1C*).

This finding is reminiscent of the observation of 'conditional' polarization-sensitivity in the locust central complex, whereby certain columnar neuron types (although notably not the homologue of E-PG neurons) were found to be sensitive to polarized light in some but not all animals (*Heinze and Homberg, 2009*), an effect attributed by the authors to the varying internal state of each animal. It should be noted that the activity we recorded and analyzed as a single glomerulus in the PB potentially represents multiple E-PG neurons which could have been differentially active. This is one alternative explanation for the variable tuning properties that we found, although it would still necessitate a degree of flexibility in the overall circuit and a mechanism for selective engagement of a single E-PG neuron from multiple candidates within the same glomerulus. The differences between ER4m and E-PG responses, in their periodicity and average PSI values, certainly imply that a

transformation or gating of signals occurred at their synapses, and the behavior of the system may be qualitatively different in animals during locomotion and navigation behaviors. Other recent work has proposed that plasticity at ring neuron to E-PG synapses in the EB is also likely required for the heading direction network to operate (*Fisher et al., 2019*; *Kim et al., 2019*). The variable responses to polarized light that we observed in E-PG neurons may therefore reflect their recent stimulation history, and even the laboratory conditions under which the animals were reared (*el Jundi et al., 2011*). The plasticity of these synapses, together with the reciprocal synapses from E-PG neurons back onto ER4m neurons, invites the question of whether the observed preference of ER4m populations for particular AoPs in the EB might also have a complex dependence on internal state or previous visual experience.

## E-PG neurons in neighboring columns exhibit similar responses, with no fixed polarotopic organization

Across different animals, glomeruli at corresponding positions in the PB exhibited different preferred angles of polarization (AoPs) (*Figure 10C*). However, since neither preferred AoPs nor the PSI values calculated for E-PG neurons were necessarily stable over the minutes taken to complete a single experiment (*Figure 10G*), we sought to address whether a dynamic organization of preferential responses might exist within the PB. We therefore limited our analysis to individual cycles of the stimulus, essentially capturing snapshots of tuning properties over a shorter timescale (30 s). For simplicity, we pooled the coordinated responses of glomeruli from the left and right sides of the PB. Across animals, we again found no common relationship between glomerulus position in the PB and the preferred AoPs of E-PG neurons (*Figure 10H*), matching the findings for the homologous CL1a neurons in locusts (*Heinze and Homberg, 2009*; *Pegel et al., 2019*).

We then asked whether, in an individual animal, there was any relationship between PB position and preferred AoP. In each recording, we picked at random a single response cycle in which the average PSI value across all glomerulus pairs exceeded a threshold (mean + 1 SD of PSI values in background regions of all E-PG recordings). We then identified the glomerulus pair with the maximum average PSI value, which we refer to as G0, and expressed all PB positions, preferred AoPs and PSI values relative to G0 (*Figure 10—figure supplement 1F*). Smooth transitions in preferred AoP across glomeruli were observed infrequently, and in 6 out of 19 animals this resulted in a weak relationship between PB position and preferred angle of polarization (asterisks, *Figure 10—figure supplement 1F*).

More generally, we found that glomeruli neighboring G0, at ±1 PB position, were likely to exhibit a similar preferred AoP to G0, to within 15˚ (*Figure 10I*, *Figure 10—figure supplement 1G*). At ±2–4 PB positions from G0, we found preferred AoPs generally shifted toward orthogonal angles (*Figure 10I*, *Figure 10—figure supplement 1F,G*) and among these positions there was again a similarity between neighboring glomeruli (*Figure 10—figure supplement 1H*). These data support our initial observation of clusters of glomeruli with similar PSI values and preferred AoPs (*Figure 10B,C*), contrasting with the polarotopic organization of tunings across the PB found for CPU1 neurons in locusts (likely homologous to P-F-R neurons in flies) (*Heinze and Homberg, 2007*; *Honkanen et al., 2019*; *Pegel et al., 2019*). Simplistically, this could be viewed as a limited representation of two approximately orthogonal angles of polarization among E-PG neurons. This would be congruent with a single predominant tuning being conveyed by the ER4m population (*Figure 9D*), since rectification of a sinusoidal tuning function could directly lead to two signals with peak responses at orthogonal angles.

## Discussion

In this study, we have demonstrated that each section of the *Drosophila* anterior visual pathway (AVP) contains polarization-tuned neurons. Together, they provide a circuit to convey polarized light signals from the specialized dorsal rim area of the eye to the compass neurons of the central complex, via the anterior optic tubercle and bulb. This pathway also conveys information about unpolarized visual features, as shown here and in previous studies. The encoding of multiple visual modalities, the similarities in the constituent neurons, and the organization of the neuropils which accommodate them (*Omoto et al., 2017*), support the view that the AVP in *Drosophila* is

homologous to the sky compass pathway described in locusts, bees, butterflies, and beetles, among other insects (*Honkanen et al., 2019*; *Warren et al., 2019*).

Our approach to investigating the neural processing of polarization vision offered a number of advantages over traditional intracellular electrophysiology. Firstly, it allowed us to simultaneously record from whole populations of neurons, which would otherwise be technically challenging. Here, we exploited this to investigate the spatial organization of polarization responses in an individual animal. This may be key in understanding the central complex, where dynamic responses reflect circuit plasticity and depend on numerous factors, such as proprioceptive inputs, internal states and goal-direction. Next, targeted expression of calcium indicators allowed us to isolate specific anatomical groups of neurons, such as specific TuBu or ring neuron populations, greatly increasing the repeatability of functional characterizations. Crucially, the identification of corresponding genetic drivers will enable silencing experiments, optogenetic stimulatio,n and multi-population recordings to probe circuit function in the future. Imaging of calcium indicators also facilitated the characterization of neurons whose axons are prohibitively thin for recording intracellularly. MeTu-like neurons, for example, have long been assumed to deliver polarization signals from the medulla to the anterior optic tubercle, and here we were able to confirm this by direct observation for the first time.

## Skylight polarization features extracted by the MEDRA

Since each detector for polarized light in the DRA essentially has a different field of view, the success of this approach depended on the ability to stimulate a sizable number of DRA ommatidia. Surprisingly, almost the full extent of the DRA was stimulated by polarized light originating from a single point in the visual field with a common angle of polarization. A wide range of polarization tunings was subsequently revealed in downstream neurons, supporting the idea that the *Drosophila* medulla dorsal rim area (MEDRA) analyzes the overall pattern of polarized light in the sky and extracts a predominant angle of polarization (AoP) (*Labhart, 2016*; *Rossel and Wehner, 1986*), rather than performing many local AoP estimates. During the morning and evening when *D. melanogaster* are most active, the pattern of polarization in the sky can be well approximated by a single, predominant AoP. DmDRA1 neurons appear to spatially integrate polarization signals from multiple columns of the MEDRA (*Figure 1*), and individual neurons heavily overlap each other (*Sancer et al., 2019*). This could provide an additional robustness to occlusions of the sky or of the DRA itself and average out inconsistencies in the available light (*Labhart et al., 2001*; *Rossel and Wehner, 1986*).

The parallel circuitry between DRA R7, DmDRA1, and MeTu neurons in MEDRA columns (*Figure 2D*), resembles the color-processing pathway found in non-DRA columns involving R7, Dm8, and Tm5c (*Gao et al., 2008*; *Karuppudurai et al., 2014*). MeTu neurons in the MEDRA may also integrate color signals, as their dendritic fields extend into the non-DRA medulla, indicating that color and polarization processing are compatible (*Figure 2—figure supplement 1*). Parallel circuits may support antagonistic processing of the color and polarization pathways downstream, potentially providing a means to selectively process polarization cues for navigation from the anti-solar hemisphere where they are strongest. We have not functionally described the responses of DmDRA2 neurons that contact R8 in this study (*Sancer et al., 2019*), and these neurons may be differently integrated with color processing. Both parallel functions will likely need to be incorporated to build a complete conceptual model of skylight polarization processing in the medulla.

## Sensory transformations through the AVP

In the anterior optic tubercle (AOTU), we found polarization-sensitive neuron populations entering and leaving the tubercle via the intermediate-lateral domain (*Figures 2–6*). We also observed polarization responses in the lateral domain, although it is unclear whether this is a result of separate polarization-sensitive MeTu types projecting from the MEDRA to different AOTU domains. Alternatively, since MeTu neurons are also postsynaptic in the AOTU (*Omoto et al., 2017*), signals from a single polarization input channel could be redistributed to different regions of the AOTU for integration with other visual modalities (*Figure 4*) or bilateral interactions (*Figure 5*). The AOTU in *Drosophila* is also likely to be a site for modulation of signals depending on time or internal states (*el Jundi et al., 2014*; *Guo et al., 2018*; *Lamaze et al., 2018*), and a capacity to modify responses may explain why we observed multiple polarotopic organizations in a MeTu neuron population in the

AOTU (*Figure 3—figure supplement 1*). However, there may also be multiple functional subtypes within the population that more tailored experiments may be able to distinguish.

Intriguingly, none of the polarotopies found in presynaptic MeTu neurons (*Figure 3E,F*) matched the polarotopy of postsynaptic TuBu dendrites in the AOTU (*Figure 6G,I*), which was extremely consistent across animals. Our findings suggest that TuBu neurons extract a processed form of the signals in the AOTU, encoding visual features within fewer neurons than the MeTu populations. TuBu neurons appear to divide signals into functional groups, and the anterior bulb-projecting TuBu$_a$ group in every fly contained a set of around six tunings covering −90° to +90° of polarization space in approximately 30° steps, tightly packed in a micro-glomerular structure with no apparent polarotopy (*Figure 7*, *Figure 8*). The question remains open as to whether a sun position system and skylight polarization system are independent in the bulb. Unlike the TuLAL neurons in locusts (homologous to TuBu), where there is convergence on the dendrites of postsynaptic neurons (*von Hadeln et al., 2020*; *Pegel et al., 2018*; *Pfeiffer et al., 2005*; *Träger et al., 2008*), TuBu neurons appear to form one-to-one contact with individual ring neurons (*Omoto et al., 2017*). Hence, we posit that the site of integration of celestial cues is not at the synapse between TuBu and ring neurons. Although we found evidence that angles of polarization are represented in the superior bulb (*Figure 7*, *Figure 8*), where unpolarized cues are also known to be represented, the populations we recorded contained a limited range of tunings and resembled a system for detecting visual features with a particular polarization signature (*Labhart, 2016*), such as horizontally polarized light reflected from surfaces like water, rather than a system for accurate estimation of orientation. Such responses would likely be mediated by more ventral regions of the eye than the DRA (*Velez et al., 2014*; *Wernet et al., 2012*). It should be noted that our polarized light stimulus broadly illuminated the eye from a dorsal position and, although we attempted to minimize reflections, we did not measure whether reflected polarized light fell on the ventral eye during our experiments.

## Stereotypic polarotopy in the periphery gives way to idiosyncratic plasticity in the CX

By recording the ensemble response of a population of ER4m ring neurons, both in the anterior bulb and ellipsoid body (EB), we determined that they do not simply relay the responses of presynaptic TuBu$_a$ neurons to the EB. Instead, they appear to deliver a subset of signals more prominently than others, bestowing the population with an ensemble response tuned to a specific angle of polarization (*Figure 9*). Furthermore, we found that this population tuning conveys a different angle of polarization in individual animals, and one exciting possibility is that this represents a flexible heading signal relative to polarized light cues, which could direct behavior (*Warren et al., 2018*). A question to address in future work is whether the preferred angle of polarization of an individual ring neuron is itself fixed, in which case we may have observed the result of a winner-take-all competition among the ER4m population in the EB, or if the whole population flexibly re-tunes to preferentially respond to a common AoP. Recordings from individual neurons will be required to resolve this.

It is clear that among ER4m and E-PG neurons, polarization tunings are not represented with a retinotopic map in the EB or PB which is common between individual animals (*Figure 9*, *Figure 10*). This is in contrast with the consistent polarotopic organizations found upstream in the MEDRA or AOTU (*Figures 1–6*), but in agreement with a previous study which showed that the azimuthal position of unpolarized visual stimuli is also not represented retinotopically in E-PG neurons (*Fisher et al., 2019*). The lack of organization in E-PG responses also matches previous findings in the corresponding CL1a neurons in locusts, but contrasts with the polarotopic organization found in other columnar neurons in the locust CX, such as CPU1, and the tangential TB1 neurons (*Heinze and Homberg, 2009*; *Heinze and Homberg, 2007*; *Pegel et al., 2019*). A potential explanation for the lack of consistent polarotopy in CL1a, or indeed E-PG neurons, was offered by *Heinze and Homberg, 2009*: at least two of each neuron type innervates an individual glomerulus in the PB. Could each of these have differential responses to polarized light to enable different configurations across the PB? Intriguingly, the TB1-like Δ7 neurons in the *Drosophila* PB appear to synapse onto only a subset of the E-PG neurons in a single glomerulus (*Turner-Evans et al., 2020*), perhaps indicating independent functional groups. We may therefore yet find a polarotopic organization of responses in the *Drosophila* CX. Alternatively, such an organization may reflect a common, genetically pre-programmed directional goal to facilitate migration, which flies may lack (*Honkanen et al., 2019*),

instead using polarization cues to follow a fixed course and disperse along idiosyncratic headings (*Dickinson, 2014*).

Our data suggest that in a given fly, E-PG neurons may respond to one of two approximately orthogonal angles of polarization, effectively dividing the population into two groups. Interestingly, when data from locust CPU1 neurons (likely homologues of P-F-R neurons in *Drosophila*) were pooled with tunings obtained from a number of other polarization-sensitive columnar CX neuron types, including CL1b (P-EG), CL2 (P-EN), CPU2, and CPU4 (P-FN), the organization of tunings in the locust PB could be interpreted as clustering around two orthogonal preferred angles (*Heinze and Homberg, 2009*). A binary system such as this would be well suited to influence downstream processes in a motor-centered coordinate frame (*Rayshubskiy et al., 2020*). For example, the eventual output of the compass network may be a command signal to activate one descending neuron of a bilateral pair to initiate a turn to either the left or right, and thus maintain a heading specified by polarization patterns in the sky.

An important next step will be to understand how polarized light influences the activity bump in columnar neurons and whether the activity of columnar neurons reciprocally influences the tunings of ER4m neurons. We did not observe an activity bump in E-PG neurons in the PB, likely due to the open-loop stimulus presentation and recordings performed in immobilized animals, although we could see evidence of flexible encoding of polarization information (*Figure 10*). According to our mappings of E-PG responses in the PB, the influence of a rotating polarized light stimulus might be to move the activity bump discontinuously between two positions, not dissimilar to observations in a recent investigation of the influence of airflow on the bump in E-PG neurons (*Okubo et al., 2020*). However, a limitation of the polarization stimulus used here is that the intensity gradient and position of the light source did not change as the angle of polarization rotated, as they would be seen to by an animal turning under a natural sky. If the ambiguity between 0/180° polarization cues is resolved by integrating light intensity information, then the stimulus we used here presented contradictory, unnatural changes (*Gkanias et al., 2019*).

Behavioral studies in ants (*Wehner and Müller, 2006*) and dung beetles (*el Jundi et al., 2015*) have demonstrated that skylight polarization cues can have a greater influence than other visual features in guidance and navigation behaviors. In *Drosophila*, intensity gradients have been shown to have a greater behavioral significance than polarized light (*Warren et al., 2018*), yet recent connectome analysis of the *Drosophila* CX highlights the polarization-sensitive ring neurons that we identified here as potentially being at the top of a hierarchy of sensory inputs (*Hulse et al., 2020*). Furthermore, the unique pattern of asymmetrical connectivity between the ER4m populations from each brain hemisphere and the E-PG network hints at an attractively simple system for obtaining 360° heading information from ambiguous 0/180° polarization cues, by using signals from one population or the other depending on which side of the animal the sun is on (*Hulse et al., 2020*). A key challenge for future studies will be to uncover such mechanisms for integrating and selecting from the multiple sensory modalities and visual qualities represented in the central brain in order to navigate complex environments.

## Materials and methods

**Key resources table**

| Reagent type (species) or resource | Designation | Source or reference | Identifiers | Additional information |
|---|---|---|---|---|
| Genetic reagent (*D. melanogaster*) | Rh3/Rh4-Gal4 | Bloomington *Drosophila* Stock Center | FLYB:FBti0058802;RRID: BDSC_8603 | Synonym: pan-R7-Gal4 *Wernet et al., 2006* |
| Genetic reagent (*D. melanogaster*) | DmDRA1 split (R13E04-p65.AD; VT059781-Gal4.DBD) | This paper | FLYB:FBti0187820; FLYB:FBti0194675 | Available upon request. Components: *Dionne et al., 2018*; *Tirian and Dickson, 2017* |
| Genetic reagent (*D. melanogaster*) | R13E04-LexA | Bloomington *Drosophila* Stock Center | FLYB:FBtp0087918; RRID:BDSC_52457 | *Pfeiffer et al., 2013* |
| Genetic reagent (*D. melanogaster*) | R24F06-LexA | Bloomington *Drosophila* Stock Center | FLYB:FBti0155501; RRID:BDSC_52695 | *Pfeiffer et al., 2013* |

*Continued on next page*

*Continued*

| Reagent type (species) or resource | Designation | Source or reference | Identifiers | Additional information |
|---|---|---|---|---|
| Genetic reagent (*D. melanogaster*) | R56F07-Gal4 | Bloomington *Drosophila* Stock Center | FLYB:FBti0136983; RRID:BDSC_39160 | *Jenett et al., 2012* |
| Genetic reagent (*D. melanogaster*) | R73C04-Gal4 | Bloomington *Drosophila* Stock Center | FLYB:FBti0138074; RRID:BDSC_39815 | *Jenett et al., 2012* |
| Genetic reagent (*D. melanogaster*) | R17F12-Gal4 | Bloomington *Drosophila* Stock Center | FLYB:FBti0133538; RRID:BDSC_48779 | *Jenett et al., 2012* |
| Genetic reagent (*D. melanogaster*) | R34H10-Gal4 | Bloomington *Drosophila* Stock Center | FLYB:FBti0135093; RRID:BDSC_49808 | *Jenett et al., 2012* |
| Genetic reagent (*D. melanogaster*) | R88A06-Gal4 | Bloomington *Drosophila* Stock Center | FLYB:FBti0139280; RRID:BDSC_46847 | *Jenett et al., 2012* |
| Genetic reagent (*D. melanogaster*) | R49E09-Gal4 | Bloomington *Drosophila* Stock Center | FLYB:FBti0136346; RRID:BDSC_38692 | *Jenett et al., 2012* |
| Genetic reagent (*D. melanogaster*) | R34D03-Gal4 | Bloomington *Drosophila* Stock Center | FLYB:FBti0135043; RRID:BDSC_49784 | *Jenett et al., 2012* |
| Genetic reagent (*D. melanogaster*) | R19C08-Gal4 | Bloomington *Drosophila* Stock Center | FLYB:FBti0133678; RRID:BDSC_48845 | *Jenett et al., 2012* |
| Genetic reagent (*D. melanogaster*) | SS00096 (R19G02-p65.AD; R70G12-Gal4.DBD) | other | FLYB:FBtp0117182; FLYB:FBtp0122173 | *Kim et al., 2017* Provided by V. Jayaraman and T. Wolff, available via https://www.janelia.org/split-GAL4 |
| Genetic reagent (*D. melanogaster*) | R60D05-LexA | Bloomington *Drosophila* Stock Center | FLYB:FBti0156166; RRID:BDSC_52867 | *Pfeiffer et al., 2013* |
| Genetic reagent (*D. melanogaster*) | R34D03-LexA | Bloomington *Drosophila* Stock Center | FLYB:FBti0135043; RRID:BDSC_49784 | *Pfeiffer et al., 2013* |
| Genetic reagent (*D. melanogaster*) | 20×UAS-IVS-Syn21-opGCaMP6s-p10 | other | FLYB:FBti0195669 | Provided by G. Rubin |
| Genetic reagent (*D. melanogaster*) | UAS-mCD4::tdTomato | Bloomington *Drosophila* Stock Center | FLYB:FBtp0068010; RRID:BDSC_35837 | *Han et al., 2011* |
| Genetic reagent (*D. melanogaster*) | UAS-sytGCaMP6s | Bloomington *Drosophila* Stock Center | FLYB:FBti0181795; RRID:BDSC_64415 | *Cohn et al., 2015* |
| Genetic reagent (*D. melanogaster*) | UAS-tdTomato | Bloomington *Drosophila* Stock Center | FLYB:FBti0145104; RRID:BDSC_36328 | *Schulte, 2011*; *Shaner et al., 2004* |
| Genetic reagent (*D. melanogaster*) | 10×UAS-mCD8::GFP | Bloomington *Drosophila* Stock Center | FLYB:FBti0131930; RRID:BDSC_32184 | *Pfeiffer, 2010* |
| Genetic reagent (*D. melanogaster*) | 26×LexAop-mCD8::GFP | Bloomington *Drosophila* Stock Center | FLYB:FBti0131946; RRID:BDSC_32207 | *Pfeiffer, 2010* |
| Genetic reagent (*D. melanogaster*) | 10×UAS-mCD8::RFP; 13×LexAop-mCD8::GFP | Bloomington *Drosophila* Stock Center | FLYB:FBst0032229; RRID:BDSC_32229 | *Pfeiffer, 2010* |
| Genetic reagent (*D. melanogaster*) | UAS-nSyb-spGFP1-10, LexAop-CD4-spGFP11 | Bloomington *Drosophila* Stock Center | FLYB:FBtp0111281; RRID:BDSC_64314 | Components: *Gallio, 2016*; *Macpherson et al., 2015* |
| Genetic reagent (*D. melanogaster*) | MCFO-4 | Bloomington *Drosophila* Stock Center | FLYB:FBst0064088; RRID:BDSC_64088 | *Nern et al., 2015* |
| Genetic reagent (*D. melanogaster*) | MCFO-5 | Bloomington *Drosophila* Stock Center | FLYB:FBst0064089; RRID:BDSC_64089 | *Nern et al., 2015* |
| Genetic reagent (*D. melanogaster*) | MCFO-6 | Bloomington *Drosophila* Stock Center | FLYB:FBst0064090; RRID:BDSC_64090 | *Nern et al., 2015* |
| Genetic reagent (*D. melanogaster*) | GRASP | Bloomington *Drosophila* Stock Center | FLYB:FBst0064314; RRID:BDSC_64314 | *Macpherson et al., 2015* |
| Genetic reagent (*D. melanogaster*) | *trans*-Tango | Bloomington *Drosophila* Stock Center | FLYB:FBst0077124; RRID:BDSC_77124 | *Talay et al., 2017* Provided by G. Barnea |
| Antibody | Anti-DN-cadherin (rat monoclonal) | Developmental Studies Hybridoma Bank | Cat#:DN-Ex #8; RRID:AB_528121 | (1:20) |

*Continued on next page*

*Continued*

| Reagent type (species) or resource | Designation | Source or reference | Identifiers | Additional information |
|---|---|---|---|---|
| Antibody | Anti-Neuroglian (mouse monoclonal) | Developmental Studies Hybridoma Bank | Cat#:BP 104; RRID:AB_528402 | (1:30) |
| Antibody | Anti-GFP (chicken polyclonal) | Abcam | Cat#:ab13970; RRID:AB_300798 | (1:1000) |
| Antibody | Anti-DsRed (rabbit polyclonal) | Takara Bio | Cat#:632496; RRID:AB_10013483 | (1:1000) |
| Antibody | Anti-HA (rabbit monoclonal) | Cell Signaling Technology | Cat#: 3724S; RRID:AB_1549585 | (1:300) |
| Antibody | Anti-V5 (mouse monoclonal) | Thermo Fisher Scientific | Cat#:37–7500; RRID:AB_2533339 | (1:1000) |
| Antibody | Anti-Mouse conjugated to Cy5 (goat polyclonal) | Jackson ImmunoResearch | Cat#:115-175-166; RRID:AB_2338714 | (1:300) |
| Antibody | Anti-Rat conjugated to Cy3 (goat polyclonal) | Jackson ImmunoResearch | Cat#:112-165-167; RRID:AB_2338251 | (1:300) |
| Antibody | Anti-Rat conjugated to Cy5 (goat polyclonal) | Jackson ImmunoResearch | Cat#:112-175-143; RRID:AB_2338263 | (1:300) |
| Antibody | Anti-Rabbit conjugated to Cy3 (goat polyclonal) | Jackson ImmunoResearch | Cat#:111-165-003; RRID:AB_2338000 | (1:300) |
| Antibody | Anti-GFP conjugated to Alexa488 (rabbit polyclonal) | Thermo Fisher Scientific | Cat#:A-21311; RRID:AB_221477 | (1:1000) |
| Antibody | Anti-GFP (mouse monoclonal) | Sigma-Aldrich | Cat#:G6539; RRID:AB_259941 | (1:1000) |
| Antibody | Anti-Chicken conjugated to Alexa488 (goat polyclonal) | Thermo Fisher Scientific | Cat#:A-11039; RRID:AB_2534096 | (1:1000) |
| Antibody | Anti-Mouse conjugated to Alexa488 (goat polyclonal) | Thermo Fisher Scientific | Cat#:A-10680; RRID:AB_2534062 | (1:1000) |
| Software, algorithm | Fiji | ImageJ | RRID:SCR_002285 | *Schindelin et al., 2012* |
| Software, algorithm | SlideBook 6 | Intelligent Imaging Innovations | RRID:SCR_014300 | |
| Software, algorithm | MATLAB R2017a | Mathworks | RRID:SCR_001622 | |

## In vivo calcium imaging

### Fly preparation

Flies were raised at 25°C on a standard cornmeal/molasses diet in 40 ml vials, under a 12:12 hr dark: light cycle. Imaging experiments were performed between ZT0–14, although time of day was not a factor in our experimental design or analysis. We imaged 1–7 day old female flies expressing either UAS-GCaMP6s (*Chen et al., 2013*) for dendritic regions or UAS-sytGCaMP6s (*Cohn et al., 2015*) for axon terminals, together with UAS-tdTomato (*Shaner et al., 2004*) for image registration (for genotypes see *Appendix 2—table 1.*). Flies were cold anesthetized and mounted on a custom fly holder, modified from *Weir et al., 2016*, with the head pitched forward so that its posterior surface was approximately horizontal (*Figure 1—figure supplement 1A*). Surfaces of the fly holder visible to the fly were covered in matte white paint (Citadel) and roughened to reduce confounding reflected polarized light cues (*Foster et al., 2018*). We fixed the fly to the holder using UV-curing glue (Fotoplast) around the posterior-dorsal cuticle of the head and at the base of the wings on either side of the thorax. To reduce movement of the brain, we fixed the legs, abdomen, and proboscis with beeswax. We used forceps to remove the cuticle and air-sacs above the optic lobe or central brain, depending on the recording site, and cut muscle 1 (*Demerec, 1950*) to reduce movement. Physiological saline (103 mM NaCl, 3 mM KCl, 1.5 mM $CaCl_2$, 4 mM $MgCl_2$, 26 mM $NaHCO_3$, 1 mM $NaH_2PO_4$, 10 mM trehalose, 10 mM glucose, 5 mM TES, 2 mM sucrose) was perfused continuously over the brain at 1.5 ml/min via a gravity drip system and the bath was maintained at 22°C for the

duration of experiments by an inline solution heater/cooler (SC-20, Warner Instruments) connected to a temperature controller (TC-324, Warner Instruments).

## Imaging setup

We used a two-photon excitation scanning microscope controlled by Slidebook (ver. 6, 3i) with a Ti: sapphire laser (Chameleon Vision, Coherent) at 920 nm and a 40× objective (0.8 numerical aperture, NIR Apo, Nikon). For each brain area imaged, we aimed to capture the full extent of the volume of labeled neurons, using a maximum step-size of 4 μm between imaging planes, and maintained a volume-rate of at least 1 Hz. Image resolution varied depending on the number of planes captured but was not less than 100 pixels in the longest dimension. We recorded frame capture markers and stimulus events on a DAQ (6259, NI) sampling at 10 kHz.

## Polarized light stimulus

We used a custom polarized light stimulus device comprising a UV LED (M340D3, Thorlabs), a 7.5 mm diameter aperture, a ground glass diffuser (DGUV10-1500, Thorlabs), a low-pass filter (FGUV11, Thorlabs), and a removable linear polarizer (BVO UV, Bolder Optic). The UV LED was controlled through MATLAB 2017a (Mathworks, MA) via a DAQ (6259, NI) and LED driver (LEDD1B, Thorlabs). The polarizer was rotated with a bipolar stepper motor (ROB-10551, SparkFun) and spur gears (1:1), and a motor driver (ROB-12779, SparkFun) controlled through MATLAB (2017a, Mathworks) via a DAQ (USB1208, MCC), with a minimum step-size of 7.5°. The motor was operated in open-loop and a Hall effect sensor (A1324, Allegro) was used to detect the proximity of a magnet which passed once per revolution, in order to verify correct operation. Angles of polarization and directions of rotation are expressed from an external viewpoint looking toward the fly (*Figure 1—figure supplement 1A*). 0°/180° corresponds to a vertical orientation in the transverse plane and an alignment with the fly's long-axis in the horizontal plane. We investigated the reproducibility of the polarizer's angular positions and measured <1° variation over multiple revolutions and <1° of position hysteresis (backlash) after reversing the direction of rotation. The surface of the polarizer was positioned frontally, 110 mm from the fly's head at an elevation of approximately 65° above the eye-equator (*Figure 1—figure supplement 1A*). The light subtended a solid angle of approximately 4° and the entirety of the fly, including the dorsal rim area of both eyes, was illuminated. We measured approximately 0.8 μW/cm$^2$ irradiance at the fly's head at the spectral peak of 342 nm (8.7 nm FWHM) with the polarizer attached (*Figure 1—figure supplement 1B*). We calibrated the LED power in order to maintain a similar irradiance value with the polarizer removed (*Figure 1—figure supplement 1B*). We measured a ± 5% modulation in light intensity over a full revolution of the device (*Figure 1—figure supplement 1B*), due to a slight off-axis tilt of the diffuser and polarizer. This intensity modulation was of similar magnitude both with the polarizer attached and removed, and was therefore unlikely to be an effect of polarization. We reasoned that if calcium activity in neurons was modulated by the rotation of the device with the polarizer attached, but not with the polarizer removed, then the varying angle of polarization throughout the revolution was its cause, rather than the varying light intensity. To quantify the difference in modulation between these two polarizer conditions, we report the change in polarization-selectivity index (ΔPSI) throughout (see *Polarization-selectivity index*).

We verified that the polarized light stimulus elicited an expected response in the dorsal rim photoreceptors by recording calcium signals in R7/R8 terminals in the medulla dorsal rim area (MEDRA) (*Figure 1—figure supplement 1C–E*). We observed preferential responses to different angles of polarized light across the MEDRA and approximately orthogonal preferred angles within R7/R8 pairs in individual columns (*Figure 1—figure supplement 1C–E*). Moving anterior to posterior across the right MEDRA, the preferred angle of polarization rotated counter-clockwise (*Figure 1—figure supplement 1E*), matching a previous characterization (*Weir et al., 2016*). We estimated that at least 80% of MEDRA columns were stimulated and conveyed polarization tunings that matched predictions based on the anatomy of photoreceptors at corresponding positions (*Weir et al., 2016*; *Figure 1—figure supplement 1E–G*), with weak responses or deviations observed only in the anterior-most columns (*Figure 1—figure supplement 1E,F*) likely due to their posterior receptive fields which faced away from the stimulus. With the polarizer removed, we observed no spatial

organization of tunings in photoreceptor terminals and PSI values close to zero (*Figure 1—figure supplement 1J*), indicating reduced modulation of activity by the stimulus.

## LED display

We used a 32 × 96 pixel display, composed of 8 × 8 panels of LEDs (470 nm, Adafruit) with controllers (*Reiser and Dickinson, 2008*), arranged in a half-cylinder spanning ±90° azimuth from visual midline and approximately ±30° elevation from the eye-equator (*Figure 1—figure supplement 1A*). Each LED pixel subtended a solid angle of approximately 1.5° at the eye-equator. At their maximum intensity, we measured approximately 0.11 μW/m$^2$ irradiance at the fly's head at the spectral peak of 460 nm (243 nm FWHM).

## Experimental protocols

Visual stimuli were presented in sets as described below. Between each stimulus set, 10 s of spontaneous activity was recorded in darkness with no visual stimulation. The polarizer could only be removed or attached between recordings, but could be done so while maintaining the same imaging parameters and field-of-view under both conditions.

### Angle of polarization tuning

To characterize responses to different angles of polarization, we rotated the polarizer discontinuously in 30° steps with the UV LED on throughout. Each of the 12 positions (six unique angles of polarization) was maintained for 4–4.5 s and we used 4 s of imaging data collected during this period in our analysis. With a volume-rate of at least 1 Hz, this ensured that at least four time-points were captured for every plane in the imaging volume. The polarizer was then rotated through 30° in 0.5 s. At least two complete revolutions of the polarizer were made. For recordings with the polarizer removed, the procedure was repeated and one revolution of the stimulus was made. The discontinuous rotation of the polarizer differs from most previous studies of polarization-sensitive neurons in insects, which have typically analyzed spike frequency modulated by continuously rotating stimuli, and as such the data obtained using this protocol may not be directly comparable.

### Polarized light flash

To characterize responses to individual wide-field flashes of polarized light, the polarizer was first rotated to 0° (vertical) in darkness. A series of three flashes of the UV LED were presented, 4 s on:4 s off. After 10 s, the same procedure was repeated with the polarizer at 90° (horizontal). The light was the same used in the angle of polarization tuning protocol. For recordings with the polarizer removed, the procedure was repeated with flashes at the 0° position.

### Unpolarized light flash

To characterize responses to individual wide-field flashes of unpolarized light, the entire LED display was illuminated following the same procedure as for polarized light flashes.

### Bars

To characterize retinotopic responses to unpolarized stimuli, a single bright, vertical bar was presented on the LED display (32 × 1 pixel) with all other LEDs off (0.78 Weber contrast). Bars initially remained stationary for 3 s, then jittered left and right (±1 pixel) for 3 s, followed by an inter-trial period of 4 s with all LEDs off. Bars were presented at five equally spaced azimuth positions spanning ±90°, presented sequentially from left to right around the fly. This procedure was repeated twice.

### Dot-field expansion

To characterize responses to unpolarized motion stimuli, a sparse random dot pattern was presented on the LED display that simulated forward translational optic-flow (thrust), with the frontal point of expansion approximately at the eye-equator. Approximately 1% of LEDs in the display were illuminated in each frame of the pattern, with all other LEDs off (0.83 Weber contrast). Windowed regions of this pattern were presented sequentially (lateral-left: −90°:−50° azimuth; frontal: −40°: +40° azimuth; lateral-right: +50°:+90° azimuth; each covering the full elevation extent of ±30°)

followed by the whole pattern (−90°:+90° azimuth). Motion was presented in each region for 4 s, with an inter-trial period of 4 s with all LEDs off. This procedure was repeated twice.

## Data analysis

### Data export

Recorded imaging data was exported as 16-bit tiff frames. We compiled all time-points for a single imaging plane and a maximum average intensity projection (MIP, detailed below) across all planes at each time-point.

### Image registration

We used a DFT-based registration algorithm (*Guizar-Sicairos et al., 2008*) to first correct for motion in the MIP of the activity-independent tdTomato channel across all time-points. We then applied the same registration displacements (*x,y*) to all individual planes of the activity-dependent GCaMP channel.

### Maximum intensity projection

We constructed a maximum intensity projection (MIP) based on each imaging plane's time-averaged fluorescence intensities during periods of inactivity, which avoided a bias towards including neurons that were bright throughout an experiment but did not necessarily show modulation (versus neurons which were inhibited for the majority of an experiment but were modulated nonetheless). The time-series of each pixel in the projection also originated from a fixed plane throughout the recording. In summary: for each imaging plane, we found an average intensity image sampling only frames captured during periods of inactivity between stimulus sets. We then found the imaging plane (*z*) with the highest average intensity at each position (*x,y*). The intensity time-series (*t*) from this location (*x, y,z*) was then inserted into a new array (*x,y,t*) to form the projection. Neighboring pixels in the projection could therefore contain signals from different imaging planes, but individual pixels contained signals from only one plane. All analyses were conducted on this projection unless otherwise stated.

### Angle of polarization tuning

For each pixel, we found the average fluorescence intensity across the frames captured during each angle presentation to obtain a polarization tuning curve. Since a polarization-tuned analyzer should respond identically to parallel angles of polarization (e.g. 0°/180°), we expected bimodal data with diametrically opposite modes. We therefore found the axial mean resultant vector, correcting for grouped data, and took its angle as the preferred angle of polarization, defined modulo 180° (*Batschelet, 1965*; *Berens, 2009*; *Zar, 1999*).

### Polarization-selectivity index

For each pixel, we found the average fluorescence intensity (ranging from 0 to $2^{16}$, arbitrary units) during the first two presentations of the angles closest to and diametrically opposite its preferred angle of polarization in the tuning experiment ($F_{pref}$). We then found the average intensity at orthogonal angles ($F_{ortho}$) and calculated the polarization-selectivity index (PSI) as the difference between $F_{pref}$ and $F_{ortho}$, divided by their sum, with possible values ranging from 0 to 1. Where average PSI values are reported for a driver line, we used a broad ROI drawn around all labeled neurons in the brain area recorded, which we refer to as the 'overall ROI'. To draw the overall ROI, we used an average intensity image from frames between stimulus sets as a guide. We also used this average intensity image to define additional regions: we defined regions of 'neurons' as the brightest 10% of pixels within the overall ROI, unless otherwise stated (e.g. *Figure 7B,C*), and 'background' as the dimmest 10% of pixels outside of the overall ROI. For the overall ROI, neurons and background regions, the distribution of PSI values within a recording tended to be non-normal; for average values we report the median value for an individual animal and the mean of the median values across animals. Where ΔPSI values are reported, we subtracted the mean PSI values within the same region across all tuning experiments recorded with the polarizer removed. Where we applied a PSI-threshold to filter polarization-selective pixels in a recording (e.g. tuning maps, polarotopy analysis), we used the mean + 1 SD of PSI values within its background. This typically resulted in a PSI threshold between 0.3 and 0.4. This threshold was modified for E-PG recordings in the protocerebral bridge

where PSI values of neurons tended to be lower than the background when averaged over multiple presentations; instead we used the mean + 1 SD of PSI values within neurons across all tuning experiments with the polarizer removed.

## Polarization tuning maps

To construct spatial maps of polarization tuning, we combined a color-coded representation of preferred angle of polarization and a grayscale representation of average intensity (*Figure 1—figure supplement 1J*). Pixels falling within the overall ROI which had an above-threshold PSI value (see *Polarization-selectivity index*) were assigned a color consistent with those used previously (*Weir et al., 2016*) to convey their preferred angle of polarization. All other pixels with below-threshold PSI value or falling outside of the overall ROI convey their average intensity during periods of inactivity with a normalized grayscale color-code (*Figure 1—figure supplement 1J*).

## Polarization tuning curves

For polarization tuning curves (*Appendix 1–figure 1A*), we used the responses of pixels in the region of 'neurons' in each animal (see *Polarization-selectivity index*). For each pixel, we calculated the average fluorescence intensity ($F_m$) at each angle of polarization presented and normalized values by calculating $\Delta F/F = F_m/F_0 - 1$, where $F_0$ was the average intensity during periods of inactivity between stimulus sets. We interpolated each pixel's resulting tuning curve in the Fourier domain, resampling to 360 data points, then shifted the curve according to the pixel's preferred angle of polarization (rounded to the closest degree) so that all tuning curves were aligned. We then found the average tuning curve for all pixels in each individual animal.

## Automatically generated ROIs

In addition to manually drawn ROIs, we generated ROIs based on polarization tuning maps (*Figure 1—figure supplement 2*). Briefly, we discretized tuning maps so that they contained only six preferred angles of polarization, corresponding to those presented in the tuning experiment ±15°, plus null values for excluded pixels. For each angle, we identified contiguous areas of 20 or more pixels with that tuning and retained the largest area as an ROI.

## Time-series

We found the mean fluorescence intensity of pixels within a given ROI in each frame to obtain its time-series ($F_t$), which we normalized to a baseline value as follows. For polarization tuning experiments, we calculated $\Delta F/F = F_t/F_0 - 1$, where $F_0$ was the root mean square value of the time-varying intensity across the entire experiment. For all other experiments, $F_0$ was the mean of $F_t$ during the 0.5 s preceding stimulus onset. To find the average time-series across multiple recordings with mismatched sampling times, we resampled values at a common rate using linear interpolation. This procedure produced no discernible alteration of the original data points.

## Polarotopy and scatter plots

For recordings in the medulla and AOTU, we included only the set of polarization-selective pixels, as described for the tuning maps (see *Polarization tuning map*). For recordings in the bulb and protocerebral bridge, we used ROIs drawn manually on individual glomeruli. We projected pixel or ROI positions (*x,y*) onto a single horizontal axis (anterior-posterior in the medulla, medial-lateral in the central brain) or vertical axis (ventral-dorsal throughout) and then normalized to give a linear position ranging from 0 to 1. The majority of recordings were performed in the right brain hemisphere; where left hemisphere recordings were included, we inverted their positions along both axes (i.e. in the medulla, anterior positions on the left were pooled with posterior positions on the right), since we expected the mirror-symmetric polarotopy found in the dorsal rim (*Figure 1—figure supplement 1G,H*) to be preserved downstream. We then pooled the normalized positions and corresponding preferred AoP across all recordings and created a scatter plot with a random subset of 1000 data points, displaying either the corresponding PSI value or preferred AoP as the color of each point in the plot.

We quantified circular-linear associations between preferred angle (multiplied by two to correct for axial data) and normalized position by finding the slope and phase offset of a regression line,

and then a correlation coefficient, according to *Kempter et al., 2012*. We found the correlation coefficient for the population by pooling all data points, then performed a permutation test on the pooled dataset with shuffled combinations of position and preferred AoP and recalculated the correlation coefficient 10,000 times. We report an upper-bound on the p-value as the proportion of shuffled datasets with a correlation coefficient exceeding that found for the experimental dataset plus one (*Phipson and Smyth, 2010*). We also found the correlation coefficients for individual recordings and an associated p-value (*Kempter et al., 2012*). Where indicated, the regression lines for the pooled dataset and for individual recordings with a sufficient number of pixels to give a meaningful correlation ($p < 0.05$) are shown on scatter plots.

We applied the Fisher z-transformation to correlation coefficients to find a mean correlation coefficient across flies. We used a hierarchical bootstrap method (*Saravanan et al., 2020*) to find 95% confidence intervals for the mean correlation coefficient found. We resampled with replacement from the population of flies, then resampled with replacement from all recordings made from those flies and recalculated the mean correlation coefficient after applying the Fisher z-transformation, repeated 10,000 times. From the bootstrapped population of mean correlation coefficients we found confidence intervals using the bias-corrected and accelerated method (*Efron, 1987*). In all cases, the correlation coefficient for the pooled dataset from all recordings was found to be close to the mean coefficient for individual flies and within the confidence interval calculated. For recordings in the bulb and protocerebral bridge, we also calculated the circular-circular correlation coefficient (*Berens, 2009*; *Zar, 1999*).

## Combined polarization and bar activity maps
For each pixel, we found the standard deviation of its fluorescence intensity across the frames captured during the angle of polarization tuning experiment. We then repeated this for the frames captured during the unpolarized bar experiment, resulting in maps of activity during each experiment. We normalized the values in each activity map individually, then subtracted the bar map from the polarization map (*Figure 4A*).

## Polar histograms
We found the normalized probability distribution of preferred angles of polarization with a bin width of 15˚. We then constructed polar histograms with each bin's probability depicted as the area of a wedge, rather than its radial length. We included in this analysis either all pixels within the overall ROI (*Figure 9*) (see *Polarization-selectivity index*) or the region of neurons only (*Figure 7*) (see *Polarization tuning maps*), in which case we excluded recordings with few above-threshold pixels (less than 10% of the overall ROI). The results were qualitatively similar in both cases.

## Population tuning vectors
For individual recordings, we found the direction and length of the population tuning in an individual animal by calculating the axial mean resultant vector of its preferred angles of polarization. For the pixel-based approach, we included all pixels within the overall ROI and weighted individual preferred angles by their PSI value (*Berens, 2009*), rather than applying a threshold. Since individual neurons with a larger area provided a greater contribution in this analysis we compared it with an ROI-based approach, using ROIs drawn manually on individual micro-glomeruli in the bulb. We excluded recordings with fewer than four ROIs, and weighted the individual preferred angle of an ROI by its mean PSI-value. The results were qualitatively similar for both approaches.

## Cross-correlation
For E-PG recordings in the protocerebral bridge, we manually drew ROIs on the 16 individual glomeruli visible in each recording (one additional column on either end of the PB does not contain E-PGs). We then paired each ROI on the left side with an ROI on the right side, using a pairing scheme which wrapped on either side independently (i.e. 1L/1R, 8L/2R, 7L/3R, see *Figure 10*; *Figure 10—figure supplement 1D*). For each pair, we obtained the time-series for the ROIs across all frames in the recording and found their normalized cross-correlation coefficient at zero lag, ranging from −1 to 1. We plot the coefficient values for each pair (*Figure 10—figure supplement 1E*) and the mean coefficient across all pairs from all recordings after applying the Fisher z-transformation.

We then shifted the pairing scheme by one position on the right side and repeated the procedure until all pairing schemes had been evaluated.

### Auto-correlation

For recordings in the bulb, we used ROIs manually drawn on individual micro-glomeruli. For E-PG recordings in the protocerebral bridge, we used ROIs drawn on pairs of left and right glomeruli (*Figure 10B-C*). For each ROI, we obtained the time-series across the first two cycles of the tuning experiment. We detrended the time-series and calculated its normalized auto-correlation function. We then found the time difference between the first peak in the function and the period of the stimulus presented during the tuning experiment. We plot the value of these time differences for each ROI, which we refer to as a 'peak shift' (*Figure 10—figure supplement 1C*), along with limits for the maximum expected peak shift for a phase-locked response to the stimulus (±2 s, half the duration of each angle presentation).

## Confocal imaging

### Immunostaining

Flies were raised at 25°C on a standard cornmeal/molasses diet in bottles or vials, under a 12:12 hr dark:light cycle, and we dissected 3–4 day old female flies (for genotypes see *Appendix 2—table 1.*). For *trans*-Tango analyses we dissected 17–18 day old female flies raised at 18°C (*Talay et al., 2017*).

Immunohistochemical staining was conducted as previously described (*Omoto et al., 2017*; *Omoto et al., 2018*). Briefly, brains were dissected in phosphate buffered saline (PBS) and fixed in ice–cold 4% EM–grade paraformaldehyde in PBS for 2.5 hr. They were subsequently washed for $4 \times 15$ min in ice–cold PBS followed by cold ethanol dehydration (5 min washes in 5, 10, 20, 50, 70, 100% EtOH). After incubation for approximately 12 hr in 100% EtOH at −20°C, brains were subjected to a rehydration procedure with EtOH in the reverse sequence. Brains were then washed for $4 \times 15$ min in ice–cold PBS and $4 \times 15$ min in ice-cold 0.3% PBT (PBS with 0.3% Triton X–100), followed by $4 \times 15$ min in room temperature (RT) 0.3% PBT. They were then incubated in blocking buffer (10% Normal Goat Serum in 0.3% PBT) for 30 min at RT. Following this, the brains were incubated in primary antibodies, diluted in blocking buffer at 4°C for approximately three days. They were subsequently washed $4 \times 15$ min in RT 0.3% PBT and placed in secondary antibodies diluted in blocking buffer at 4°C for approximately 3 days. They were finally washed $4 \times 15$ min in RT 0.3% PBT and placed in VectaShield (Vector Laboratories) at 4°C overnight before imaging. *trans*–Tango and GRASP analyses required separate staining of neuropil and respective fluorophores due to different incubation times.

### Confocal microscopy and image analysis

Processed brains were mounted on glass slides and imaged in either the anteroposterior (A–P) or dorsoventral (D–V) axis with a Zeiss LSM 700 Imager M2 using Zen 2009 (Carl Zeiss), with a 40× oil objective. Images were processed using Image J (FIJI) (*Schindelin et al., 2012*). Image stacks of the AOTU or EB were rotated slightly and interpolated to align the neuropil with the imaging plane. Background labeling was removed to improve visualization in some projections (*Figure 2B,C*, *Figure 5A*).

## Acknowledgements

We are grateful to Sam LoCascio for valuable technical assistance with the MCFO technique. We thank Holger Krapp, Kit Longden and Giovanni Frighetto for their advice and comments on the manuscript. We thank Stanley Heinze, Romain Franconville and two anonymous reviewers for their time and effort spent providing constructive feedback. We also thank Orkun Akin for providing reagents and code, Hannah Haberkern for insightful discussion of data, and Michael Reiser for helpful input at an early stage of the work. Tanya Wolff and Vivek Jayaraman kindly provided the split-Gal4 line SS00096. Stocks obtained from the Bloomington *Drosophila* Stock Center (NIH P40OD018537) were used in this study.

## Additional information

### Funding

| Funder | Grant reference number | Author |
| --- | --- | --- |
| National Institutes of Health | R01-NS096290 | Volker Hartenstein |
| National Institutes of Health | R01-EY026031 | Mark A Frye |

The funders had no role in study design, data collection and interpretation, or the decision to submit the work for publication.

### Author contributions

Ben J Hardcastle, Conceptualization, Data curation, Software, Formal analysis, Supervision, Validation, Investigation, Visualization, Methodology, Writing - original draft, Writing - review and editing; Jaison J Omoto, Conceptualization, Formal analysis, Supervision, Investigation, Visualization, Methodology, Writing - review and editing; Pratyush Kandimalla, Data curation, Formal analysis, Investigation, Visualization, Writing - review and editing; Bao-Chau M Nguyen, Data curation, Formal analysis, Investigation; Mehmet F Keleş, Investigation, Methodology; Natalie K Boyd, Investigation; Volker Hartenstein, Conceptualization, Resources, Supervision, Funding acquisition, Visualization, Project administration, Writing - review and editing; Mark A Frye, Conceptualization, Resources, Supervision, Funding acquisition, Project administration, Writing - review and editing

### Author ORCIDs

Ben J Hardcastle (iD) https://orcid.org/0000-0002-4819-5631
Jaison J Omoto (iD) https://orcid.org/0000-0003-2032-6350
Volker Hartenstein (iD) https://orcid.org/0000-0001-9676-7393
Mark A Frye (iD) https://orcid.org/0000-0003-3277-3094

### Decision letter and Author response

Decision letter https://doi.org/10.7554/eLife.63225.sa1
Author response https://doi.org/10.7554/eLife.63225.sa2

## Additional files

### Supplementary files

- Transparent reporting form

### Data availability

The data and analysis code generated during this study are available at the Open Science Framework. https://doi.org/10.17605/osf.io/3tsd6.

The following dataset was generated:

| Author(s) | Year | Dataset title | Dataset URL | Database and Identifier |
| --- | --- | --- | --- | --- |
| Hardcastle BJ | 2020 | Data and analysis code from: A visual pathway for skylight polarization processing in *Drosophila* | http://osf.io/3tsd6 | Open Science Framework, 10.17605/osf.io/3tsd6 |

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

# Appendix 1

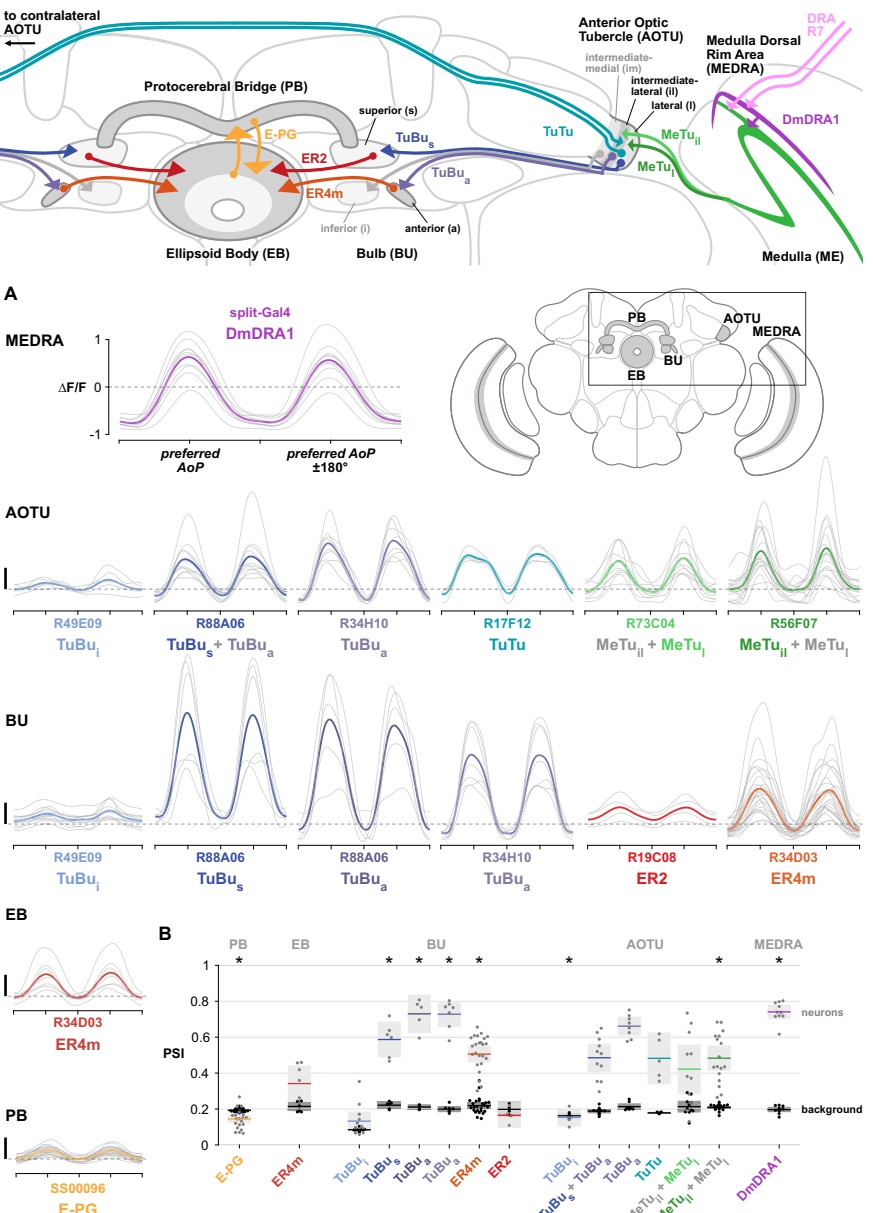

**Appendix 1—figure 1.** Summary of polarization pathway and response properties. (**A**) Polarization tuning curves for each neuron population recorded. Thin traces show tuning curves for individual animals, thick traces show their mean. All scale bars denote 1 ΔF/F. Dashed horizontal lines indicate 0 ΔF/F baseline. (**B**) Average polarization selectivity index (PSI) within each neuron population recorded (light dots) and within background regions (dark dots) in individual animals. Solid horizontal line indicates mean, shaded box denotes 95% confidence interval, asterisks denote statistically significant differences between neuron and background samples (p < 0.05, Bonferroni correction applied). See *Appendix 1–table 1* for corresponding data and p-values.

**Appendix 1—table 1.** Summary table of neurons in genetic drivers used, corresponding reconstructions in hemibrain connectome, and PSI data.

| Designation | Genetic driver / Neurons recorded (recording site) | Estimated population size (per recording) | Corresponding neurons | Instances traced (per brain hemisphere) | Average PSI† Background | Average PSI† Neurons | p-value (t-test, background vs neurons) | Average ΔPSI Background | p-value (t-test) | Average ΔPSI Neurons | p-value (t-test) |
|---|---|---|---|---|---|---|---|---|---|---|---|
| | | | Hemibrain connectome v1.2 (Scheffer et al., 2020) | | | | | (compared to control experiments with polarizing filter removed) | | | |
| DmDRA1-split | DmDRA1 (MEDRA) | 23 | n/a | n/a | 0.197 CI95 [0.18 0.21] N = 10 | 0.741 CI95 [0.70 0.78] N = 10 | 0.000000 | −0.002 CI95 [−0.02 0.01] N = 10 | 0.814673 | 0.586 CI95 [0.55 0.63] N = 10 | 0.000000 |
| R56F07 | MeTu$_{ll}$ + MeTu$_{ii}$ (AOTU) | ~50–100 | MC61* | 347 | 0.209 CI95 [0.20 0.22] N = 17 | 0.483 CI95 [0.41 0.55] N = 17 | 0.000001 | 0.001 CI95 [−0.01 0.01] N = 17 | 0.836089 | 0.280 CI95 [0.21 0.35] N = 17 | 0.000000 |
| R73C04 | MeTu$_{ll}$ + MeTu$_{iii}$ (AOTU) | ~100–150 | MC61* | 347 | 0.213 CI95 [0.18 0.25] N = 11 | 0.422 CI95 [0.29 0.56] N = 11 | 0.006841 | 0.000 CI95 [−0.03 0.03] N = 11 | 0.982761 | 0.242 CI95 [0.11 0.38] N = 11 | 0.002816 |
| R17F12 | TuTu (AOTU) | 2 | TuTuB | 2 | 0.178 CI95 [0.17 0.18] N = 5 | 0.482 CI95 [0.34 0.63] N = 5 | 0.003928 | −0.045 CI95 [−0.05 −0.04] N = 5 | 0.000027 | 0.373 CI95 [0.23 0.52] N = 5 | 0.002009 |
| R34H10 | TuBu$_a$ (AOTU) | 5–6** | | | 0.213 CI95 [0.20 0.23] N = 8 | 0.661 CI95 [0.61 0.71] N = 8 | 0.000000 | 0.033 CI95 [0.01 0.05] N = 8 | 0.003949 | 0.512 CI95 [0.46 0.57] N = 8 | 0.000000 |
| | TuBu$_a$ (BUa) | 5–6 | TuBu.01 | 5 | 0.200 CI95 [0.18 0.22] N = 7 | 0.728 CI95 [0.66 0.80] N = 7 | 0.000006 | 0.023 CI95 [0.01 0.04] N = 7 | 0.019318 | 0.512 CI95 [0.44 0.58] N = 7 | 0.000003 |
| R88A06 | TuBu$_i$ + TuBu$_a$ (AOTU) | 34 | | | 0.188 CI95 [0.18 0.20] N = 11 | 0.485 CI95 [0.41 0.56] N = 11 | 0.000013 | | | | |
| | TuBu$_s$ (BUs) | 28 | TuBu.01 TuBu.06 TuBu.08 TuBu.09 | 5 4 10 11 | 0.221 CI95 [0.20 0.24] N = 6 | 0.587 CI95 [0.49 0.69] N = 6 | 0.000454 | −0.028 CI95 [−0.05 −0.01] N = 6 | 0.015342 | 0.305 CI95 [0.21 0.41] N = 6 | 0.000538 |
| | TuBu$_a$ (BUa) | 5–6 | | | 0.211 CI95 [0.20 0.23] N = 5 | 0.730 CI95 [0.62 0.84] N = 5 | 0.000230 | −0.022 CI95 [−0.04 −0.01] N = 5 | 0.017749 | 0.620 CI95 [0.51 0.73] N = 5 | 0.000086 |
| R49E09 | TuBu$_i$ (AOTU) | 15–30 | TuBu.02 | 7 | 0.162 CI95 [0.15 0.18] N = 5 | 0.154 CI95 [0.10 0.21] N = 5 | 0.758387 | | | | |
| | TuBu$_i$ (BUi) | 15–30 | TuBu.03 TuBu.05 | 15 5 | 0.085 CI95 [0.08 0.09] N = 12 | 0.132 CI95 [0.08 0.19] N = 12 | 0.088816 | | | | |
| R19C08 | ER2 (BUs) | ~40*** | ER2_a ER2_b ER2_d ER2_c | 4 3 3 11 | 0.199 CI95 [0.16 0.24] N = 4 | 0.165 CI95 [0.09 0.24] N = 4 | 0.287547 | | | | |

*Appendix 1—table 1 continued on next page*

*Appendix 1—table 1 continued*

| Designation | Neurons recorded (recording site) | Estimated population size (per recording) | Corresponding neurons | Instances traced (per brain hemisphere) | Average PSI† | | | Average ΔPSI (compared to control experiments with polarizing filter removed) | | | |
| | | | Hemibrain connectome v1.2 (*Scheffer et al., 2020*) | | Background | Neurons | p-value (t-test, background vs neurons) | Background | p-value (t-test) | Neurons | p-value (t-test) |
|---|---|---|---|---|---|---|---|---|---|---|---|
| R34D03 | **ER4m** (BUa) | 5 | | | 0.219 CI95 [0.20 0.24] N = 25 | 0.506 CI95 [0.46 0.55] N = 25 | 0.000000 | −0.005 CI95 [−0.02 0.01] N = 25 | 0.579771 | 0.336 CI95 [0.29 0.38] N = 25 | 0.000000 |
| | **ER4m** (EB) | 10 | ER4m | 5 | 0.214 CI95 [0.19 0.24] N = 7 | 0.342 CI95 [0.24 0.44] N = 7 | 0.020818 | 0.028 CI95 [0.00 0.05] N = 7 | 0.039868 | 0.206 CI95 [0.11 0.31] N = 7 | 0.002423 |
| SS00096 | **E-PG** (PB) | 70 | EPG | 25 | 0.193 CI95 [0.19 0.20] N = 22 | 0.144 CI95 [0.12 0.17] N = 22 | 0.000133 | 0.008 CI95 [0.00 0.01] N = 22 | 0.000727 | 0.056 CI95 [0.03 0.08] N = 22 | 0.000062 |

†Data for *Appendix 1—figure 1B*

*Unsorted group of MeTu-like neurons (partially traced: excludes dendrites in medulla).

**Also contains other TuBu neurons with weak expression, not visible in calcium imaging experiments.

***Also contains ER3w neurons (10 instances in hemibrain) and ER3p neurons (nine instances total of ER3p_a plus ER3p_b). A second, previously characterized, genetic driver for ER2 (R78B06) (**Omoto et al., 2018**) contains 20–25 cell bodies. Polarization-sensitive neurons in R19C08 likely correspond to ER2_a, ER2_b and ER2_d.

## Appendix 2

**Appendix 2—table 1.** Experimental genotypes.

| Figure panel | Genotype |
|---|---|
| *Figure 1B* | *w / w; R13E04-p65.AD / +; VT059781-Gal4.DBD / 10xUAS-mCD8::GFP* (R13E04-p65.AD; VT059781-Gal4.DBD is abbreviated as *DmDRA1-split*) |
| *Figure 1B'* | *10xUAS-mCD8::RFP, 13xLexAop-mCD8::GFP / w; R13E04-p65.AD / R24F06-LexA; VT059781-Gal4.DBD / +* |
| *Figure 1B''* | *10xUAS-mCD8::RFP, 13xLexAop-mCD8::GFP / w; Rh3/Rh4-Gal4 / R13E04-LexA; + / +* |
| *Figure 1C–H* | *w / w; R13E04-p65.AD / UAS-sytGCaMP6s; VT059781-Gal4.DBD / UAS-tdTomato* |
| *Figure 1—figure supplement 1C-F* | *w / w; Rh3/Rh4-Gal4 / UAS-sytGCaMP6s; + / UAS-tdTomato* |
| *Figure 1—figure supplement 1I* | *w / w; R13E04-p65.AD / UAS-sytGCaMP6s; VT059781-Gal4.DBD / UAS-tdTomato* |
| *Figure 1—figure supplement 1J* | *w / w; Rh3/Rh4-Gal4 / UAS-sytGCaMP6s; + / UAS-tdTomato* |
| *Figure 1—figure supplement 2* | *w / w; R13E04-p65.AD / UAS-sytGCaMP6s; VT059781-Gal4.DBD / UAS-tdTomato* |
| *Figure 2A* | *w / w; + / +; R56F07-Gal4/10xUAS-mCD8::GFP* |
| *Figure 2B* | *UAS-myr::GFP, QUAS-mtdTomato-3×HA / w; R13E04-p65.AD / trans-Tango; VT059781-Gal4.DBD / +* |
| *Figure 2C* | *UAS-myr::GFP, QUAS-mtdTomato-3×HA / w; trans-Tango / +; Rh3/Rh4-Gal4 / +* |
| *Figure 2F,G* | Comparison between: *w / w; + / UAS-sytGCaMP6s; R56F07-Gal4 / UAS-tdTomato* *w / w; + / UAS-sytGCaMP6s; R73C04-Gal4 / UAS-tdTomato* |
| *Figure 2—figure supplement 1A* | *w / w; + / +; R56F07-Gal4 / 10xUAS-mCD8::GFP* |
| *Figure 2—figure supplement 1B-D* | *MCFO / w; MCFO / +; R56F07-Gal4 / MCFO* |
| *Figure 3A,C,E* | *w / w; + / UAS-sytGCaMP6s; R56F07-Gal4 / UAS-tdTomato* |
| *Figure 3B,D,F* | *w / w; + / UAS-sytGCaMP6s; R73C04-Gal4 / UAS-tdTomato* |
| *Figure 3—figure supplement 1A,C,E* | *w / w; + / UAS-sytGCaMP6s; R56F07-Gal4 / UAS-tdTomato* |
| *Figure 3—figure supplement 1B,D* | *w / w; + / UAS-sytGCaMP6s; R73C04-Gal4 / UAS-tdTomato* |
| *Figure 4A–F* | *w / w; + / UAS-sytGCaMP6s; R73C04-Gal4 / UAS-tdTomato* |
| *Figure 5A* | *UAS-myr::GFP, QUAS-mtdTomato-3×HA / w; trans-Tango / +; R73C04-Gal4 / +* |
| *Figure 5B* | *w / w; + / +; R17F12-Gal4 / 10xUAS-mCD8::GFP* |
| *Figure 5C–G* | *w / w; + / UAS-sytGCaMP6s; R17F12-Gal4 / UAS-tdTomato* |
| *Figure 6B,C* | Comparison between: *w / w; + / UAS-opGCaMP6s; R49E09-Gal4 / UAS-opGCaMP6s, UAS-CD4::tdTomato* *w / w; + / UAS-opGCaMP6s; R88A06-Gal4 / UAS-opGCaMP6s, UAS-CD4::tdTomato* *w / w; + / UAS-opGCaMP6s; R34H10-Gal4 / UAS-opGCaMP6s, UAS-CD4::tdTomato* |
| *Figure 6D,E* | *w / w; + / UAS-opGCaMP6s; R49E09-Gal4 / UAS-opGCaMP6s, UAS-CD4::tdTomato* |
| *Figure 6F,G* | *w / w; + / UAS-opGCaMP6s; R88A06-Gal4 / UAS-opGCaMP6s, UAS-CD4::tdTomato* |
| *Figure 6H,I* | *w / w; + / UAS-opGCaMP6s; R34H10-Gal4 / UAS-opGCaMP6s, UAS-CD4::tdTomato* |

*Continued on next page*

*Appendix 2—table 1 continued*

| Figure panel | Genotype |
|---|---|
| *Figure 7—figure supplement 1A,B* | Comparison between:<br>*w / w; + / UAS-opGCaMP6s; R49E09-Gal4 / UAS-opGCaMP6s, UAS-CD4::tdTomato*<br>*w / w; + / UAS-sytGCaMP6s; R49E09-Gal4 / UAS-tdTomato*<br>*w / w; + / UAS-opGCaMP6s; R88A06-Gal4 / UAS-opGCaMP6s, UAS-CD4::tdTomato*<br>*w / w; + / UAS-sytGCaMP6s; R88A06-Gal4 / UAS-tdTomato*<br>*w / w; + / UAS-opGCaMP6s; R34H10-Gal4 / UAS-opGCaMP6s, UAS-CD4::tdTomato*<br>*w / w; + / UAS-sytGCaMP6s; R34H10-Gal4 / UAS-tdTomato* |
| *Figure 7A* | *w / w; + / UAS-sytGCaMP6s; R88A06-Gal4 / UAS-tdTomato* |
| *Figure 7B,C* | Comparison between:<br>*w / w; + / UAS-sytGCaMP6s; R49E09-Gal4 / UAS-tdTomato*<br>*w / w; + / UAS-sytGCaMP6s; R88A06-Gal4 / UAS-tdTomato*<br>*w / w; + / UAS-sytGCaMP6s; R34H10-Gal4 / UAS-tdTomato* |
| *Figure 7D* | *w / w; + / UAS-sytGCaMP6s; R88A06-Gal4 / UAS-tdTomato* |
| *Figure 7E* | Comparison between:<br>*w / w; + / UAS-sytGCaMP6s; R88A06-Gal4 / UAS-tdTomato*<br>*w / w; + / UAS-sytGCaMP6s; R34H10-Gal4 / UAS-tdTomato* |
| *Figure 7F* | *w / w; + / UAS-sytGCaMP6s; R88A06-Gal4 / UAS-tdTomato* |
| *Figure 7G* | *w / w; + / UAS-sytGCaMP6s; R34H10-Gal4 / UAS-tdTomato* |
| *Figure 8B* | *10xUAS-mCD8::RFP, 13xLexAop-mCD8::GFP / w; R34D03-LexA / +; R34H10-Gal4 / +* |
| *Figure 8C* | *w / w; + / UAS-opGCaMP6s; R19C08-Gal4 / UAS-opGCaMP6s, UAS-CD4::tdTomato* |
| *Figure 8D* | Comparison between:<br>*w / w; + / UAS-opGCaMP6s; R19C08-Gal4 / UAS-opGCaMP6s, UAS-CD4::tdTomato*<br>*w / w; + / UAS-opGCaMP6s; R34D03-Gal4 / UAS-opGCaMP6s, UAS-CD4::tdTomato* |
| *Figure 8E* | *w / w; + / UAS-opGCaMP6s; R34D03-Gal4 / UAS-opGCaMP6s, UAS-CD4::tdTomato* |
| *Figure 8F* | *w / w; R34D03-LexA / UAS-nSyb-spGFP1-10, LexAop-CD4-spGFP11; R34H10-Gal4 / +* |
| *Figure 8G* | *w / w; + / UAS-opGCaMP6s; R34D03-Gal4 / UAS-opGCaMP6s, UAS-CD4::tdTomato* |
| *Figure 8H* | *w / w; + / UAS-sytGCaMP6s; R34H10-Gal4 / UAS-tdTomato* |
| *Figure 8—figure supplement 1A,C,E* | *w / w; + / UAS-sytGCaMP6s; R34H10-Gal4 / UAS-tdTomato* |
| *Figure 8—figure supplement 1B,D,F* | *w / w; + / UAS-opGCaMP6s; R34D03-Gal4 / UAS-opGCaMP6s, UAS-CD4::tdTomato* |
| *Figure 9A–E* | *w / w; + / UAS-sytGCaMP6s; R34D03-Gal4 / UAS-tdTomato* |
| *Figure 9F* | *w / w; + / UAS-opGCaMP6s; R34D03-Gal4 / UAS-opGCaMP6s, UAS-CD4::tdTomato* |
| *Figure 9G* | *w / w; + / UAS-sytGCaMP6s; R34H10-Gal4 / UAS-tdTomato* |
| *Figure 10B,C* | *w / w; R19G02-p65.AD / UAS-sytGCaMP6s; R70G12-Gal4.DBD / UAS-tdTomato*<br>*(R19G02-p65.AD; R70G12-Gal4.DBD is abbreviated as SS00096)* |
| *Figure 10D,E* | Comparison between:<br>*w / w; R19G02-p65.AD / UAS-sytGCaMP6s; R70G12-Gal4.DBD / UAS-tdTomato*<br>*w / w; + / UAS-sytGCaMP6s; R34D03-Gal4 / UAS-tdTomato* |
| *Figure 10F–I* | *w / w; R19G02-p65.AD / UAS-sytGCaMP6s; R70G12-Gal4.DBD / UAS-tdTomato* |
| *Figure 10—figure supplement 1A* | *w / w; R19G02-p65.AD / +; R70G12-Gal4.DBD / 10xUAS-mCD8::GFP* |
| *Figure 10—figure supplement 1B* | *w / w; R19G02-p65.AD / R34D03-LexA; R70G12-Gal4.DBD / UAS-nSyb-spGFP1-10, LexAop-CD4-spGFP11* |
| *Figure 10—figure supplement 1C* | Comparison between:<br>*w / w; + / UAS-sytGCaMP6s; R34H10-Gal4 / UAS-tdTomato*<br>*w / w; + / UAS-opGCaMP6s; R34D03-Gal4 / UAS-opGCaMP6s, UAS-CD4::tdTomato*<br>*w / w; R19G02-p65.AD / UAS-sytGCaMP6s; R70G12-Gal4.DBD / UAS-tdTomato* |
| *Figure 10—figure supplement 1E-G* | *w / w; R19G02-p65.AD / UAS-sytGCaMP6s; R70G12-Gal4.DBD / UAS-tdTomato* |

