## [Decision Letter]

**Acceptance summary:**

We find the work to be of high significance for a pathway that plays an important role in flying insects but has not yet received the attention it deserves. Your work is very thorough and impressive, reconstructing the pathway(s) from the eye to the central brain.

**Decision letter after peer review:**

Thank you for submitting your article "A visual pathway for skylight polarization processing in *Drosophila*" for consideration by *eLife*. Your article has been reviewed by four peer reviewers, and the evaluation has been overseen by a Reviewing Editor and Ronald Calabrese as the Senior Editor. The following individuals involved in review of your submission have agreed to reveal their identity: Stanley Heinze (Reviewer #1); Romain Franconville (Reviewer #2).

The reviewers have discussed at length the reviews with one another and the Reviewing Editor has drafted this decision to help you prepare a revised submission.

The four reviewers found the work to be of high significance for a pathway that plays an important role in flying insects but has not yet received the attention it deserves. Your work is very thorough and impressive, reconstructing the pathway(s) from the eye to the central brain, and *eLife* would be happy to publish it after you address the comments of the reviewers.

Besides a long series of useful suggestions from the four reviewers that have to do with the presentation of the data and the clarity of the figures, there are three major points that we would like you to address:

– One is to take advantage of the hemibrain reconstruction and to connect your neurons and connections to these recent data.

– It is also suggested that, while mapping ring neurons onto EPG cells, you integrate the recent work on plasticity by Fisher et al. and Kim et al. at a parallel synapse (and attempt to interpret the EPG response data from that angle).

– The last point has to do with the PSI, which appears to have confused the reviewers as to the way you define it: Do you compute the PSI using the average fluorescence intensities rather than the DF/F? The reviewers suggest that you show more tuning curves to provide a better intuition of what this PSI means.

Reviewer #1:

The manuscript by Hardcastle et al. is a rather monumental work that moves the understanding of the neural circuits for polarization vision in *Drosophila* from non-described to almost fully described in one go. I feel that this could easily have been three separate papers and reading it was a bit overwhelming at times, but it is a truly remarkable manuscript. It starts with describing polarization sensitive circuits in the optic lobe, then moves on to the processing stages along the anterior optic pathway and eventually focuses its attention to the central complex. While the first two parts are comparatively straight forward, the central complex part is by far the most interesting (to me). This is mostly because there is a wealth of surprising findings that go far beyond what is known in other species and additionally provide many fascinating access points for future studies, both in *Drosophila* and in insects that have been classical subjects of polarized light vision research.

The work follows a nice narrative, covers a wide range of anatomical and functional methods, is meticulously done, the methods are sound and the data are highly informative and relevant. While I cannot comment on the fly genetics part of the work (not my expertise), all other aspects of the work are excellent in my view.

Regarding the presentation of the data, I found the figures too small and overwhelming, with not all panels being referred to in the text. Maybe less information would make the message clearer at times, but in any case, I recommend making the figures much larger.

I much liked the Discussion, and it was refreshing to see that the authors are keenly aware of the work in non-fly species and integrate this work in their interpretation whenever needed.

I cannot wait to see follow up work, especially the interaction between the heading direction bump in the EPG neurons and the E-vector responses.

1) The authors only follow up one of the two parallel pathways into the CX, given that the polarization signal is less clear in the superior bulb. Yet parallel processing in the AOT is a seemingly highly conserved feature and it would be fantastic to know at least anatomically, if these neurons (R4m and R2) maintain this separation on the level of the CX or if they converge on the same E-PG cells. This would not require physiology, but a short look at the hemibrain data. It would generate strong predictions about the functional significance of these parallel pathways.

2) The authors use the classic division of the PB into eight glomeruli, yet data from *Drosophila* shows that there are nine glomeruli in the PB, including for the EPG neurons imaged in this study. This should be reflected in the analysis in this study. If the ninth neuron pair is not stained in the used driver line (as appeared in Wolff et al., 2015), this needs to be at least discussed. Judging by the projection patterns of column 1 and 9, the physiology of both should be identical, so no effect on the distribution of the data is likely. However, the reported discrepancy in columnar pairing across the midline between locusts and flies will have to be revised accordingly. It should be noted that the innermost column (near the midline) is the one that exists additionally in flies), as apparent by the lack of PEN and PFN neurons in this region.

3) The focus of activity in one quadrant of the EB when stimulating the ring neurons is highly intriguing! This means that there are local calcium dynamics at work that are restricted to parts of the arborization trees of the ring neurons. This to me is very unexpected and exciting. Also, this would not be reflected in the electrophysiology of these cells, when recording the spiking rates in response to the stimuli and might therefore be a general mechanism relevant also for locusts etc. I suggest to emphasize this a bit more, as it literally changed my way of thinking about the entire system! Also, it is intuitively very much in line with the recent papers on plasticity between ring neurons and EPG cells, where coincident activity in both cell types leads to synaptic depression and hence disinhibition of EPG cells and the generation of a visually driven EPG activity bump. Integrating this recent work would benefit the paper in general, but seems crucial for interpreting data on this critical part of the pathway.

4) along the lines of point three above; the recent work on plasticity of the synapse between ring neurons and EPG cells demonstrates that the stimulation history of these cells is critically important for their activity profiles. What role could this effect have on the tuning distribution in EPG cells?

5) Finally, data on other species nearly always used continuously rotating E-vectors as stimuli rather than stationary E-vectors. As the rotation direction strongly affects the tuning angle (see for example Stone et al., 2017, Figure 1), the data might not be entirely comparable. Especially when dynamic processes like in the transformation of the ring neuron code to the EPG neuron code are important, these difference might have large effects on the results. Could the authors comment on this differences in the Discussion?

Reviewer #2:

This is an important and impressive study filling a critical gap in the literature. The authors trace polarization vision pathways in *Drosophila* from the periphery all the way to the central complex – a structure involved in orientation behaviors in insects. This will allow to synthesize two streams of research on the subject: that polarization is an essential orientation cue used by many insects, mapped onto central complex neurons in some species on one side, and that the central complex circuits contain representations of the animal heading that is used to guide flexible behaviors, the dissection of which has been rendered possible by the *Drosophila* toolkit in recent years on the other side. The authors mix careful anatomical work with circuit tracing and in vivo imaging to delineate the polarization pathway while characterizing the transformations taking place at each stage. The manuscript is clear, systematic, and does not shy away from the complexity of the data. This is undoubtedly going to be an important resource for researchers in the field. I also do appreciate that code and data are made easily accessible.

1) Figure 7 would be more convincing to me if the Bulb data, rather than the EB data, was shown. As the author point out, EB imaging of densely packed ring neurons axonal processes is conducive to more artefacts than the Bulb cleanly separated glomeruli (besides area, the signal in the EB could be dominated by neurons randomly expressing higher concentration of the indicator, etc)

2) E-PG responses: the authors are upfront with the fact that the data are not entirely conclusive, but I think they could discuss a number of other possibilities of why that might be so. First, ER4m ring neuron synapses onto E-PG are likely inhibitory. Given that in that immobilized preparation the EB attractor network is not "engaged" and therefore the activity level possibly low, this might make ER4m driven responses in the EPG challenging to detect. Second, if one imagine that a mechanism similar to the one described in Fisher et al., 2020, and Kim et al., 2020, is at play at the ER4m to EPG synapses, then it's possible that the mapping between ER4m activity profiles and EPG bump position has not taken place as the animals were reared in conditions where polarized light plays little role, and that moreover, once immobilized while a polarizer is rotated, this mapping would deteriorate even further. I think the paper would gain from discussing such possibilities, as well as qualifying the findings made on EPG responses in that light.

3) Throughout the manuscript, it might be helpful to show some tuning curves (meaning response amplitude vs. polarization angle) as for now it is pretty hard to compare responses of the various types to each other.

4) Figure 1A could be improved by more hand holding for the reader unfamiliar with the DRA. It is currently hard to relate the inset to the rest of the panel.

5) Even more discussion/speculation about the various transformations happening could make the manuscript more appealing to a wider readership. For example:

– What is dmDRA1 for? why are its responses "cleaner" than the ones of the MeTu neurons?

– Where is the difference between the responses of MeTu and TuBu neurons coming from (looks like the range of MeTu is more limited. Do TuBu sample from several MeTu?)?

6) Figure 5 and arguments about the distribution of preferred angles (Figure 5F, G). This might need to be qualified: first, the possible functional role of such a distribution can only be inferred if one knows how the post-synaptic partners sample it. Is there an argument for the sampling being homogeneous? Second, this is not impossible for some of this to be imputable to the imaging angle/projection used, where some part of the volume of the BU are less accessible/blurred by others.

Reviewer #3:

Although the importance of polarized light for animal navigation has been recognized for years , the functional organization of the underlying sensory map in the central brain is largely unknown. Using a combination of trans-synaptic labeling and imaging techniques for defined neuronal populations, the manuscript by Frye and colleagues is describing a detailed analysis of the neural pathway for polarized light in the *Drosophila* visual system. The series of high-quality analyses not only uncovered a novel type of circuit design, polarotopy, at the peripheral optic lobes but also revealed how polarization identity is represented and transformed across multiple neuropils to the central brain. Especially the segregation of retinotopic visual features and polarization identities in distinct conserved AOTU domains, the innervation of the bulb region by parallel pathways with different visual features and the transformation in a more flexible CX representation is a major advance to the current morphological descriptions in *Drosophila* and the electrophysiological studies in other insects. Furthermore, with the identification of novel experimental tools for cell type-specific manipulations the new study by Hardcastle et al., will also support the dissection of non-linear circuit design for multimodal integration within the CX.

Reviewer #4:

Hardcastle and colleagues present work tracing polarized light signals through *Drosophila*'s visual system, from the dorsal rim into the central complex. This work is important because it identifies pathways in *Drosophila* using synaptic tracing and employing tools that target specific neural populations. It ties its work in nicely to the extensive work on polarization vision in other insects. It is clearly written and its experiments are well conceived and explained.

I noted three areas where I think the paper could be strengthened:

1) The tracing and anatomical work is well done and convincing. However, since trans-tango's specificity to functional synapses remains a little uncertain, I think it would be very helpful to use the hemibrain dataset published earlier this year to confirm synapses between the neuron classes identified here. As a second part of this, and perhaps more important, it would be great to identify the correspondence between the TuBu type they reported and what is labeled in the hemibrain dataset. As of now on the hemibrain, there are 10 TuBu types (from TuBu01 to TuBu10) and TuBu01 seems to be specifically connected to R4m. This would be very important and useful information for readers who would want to further investigate these neurons.

2) The PSI is defined as (R_para_ – R_ortho_)/(R_para_ + R_ortho_). This maxes out at \pm 1 when all the responses are positive, but if any are negative, this formulation for an index doesn't work as well (for instance, allowing a divide by 0). I'm a bit confused about how this index was computed throughout, given that, for instance, the DmDRA1 neurons are inhibited by R7 via ort at anti-preferred angles, so there are nominally negative responses. Moreover, in the Materials and methods, it says F0 was computed as RMS F(t), to computed dF/F, but I don't see how this wouldn't lead to negative response values. I think the methodology for computing DSI needs to be clarified. In particular, how are the authors computing the responses – deviations from a baseline (computed how?) or are they setting the minimum measured dF/F to be a new 0 response?

3) It was quite surprising to me that a small source of polarized light could activate so much of the dorsal rim, as the authors also note. Were the flies w-/w- with the white gene only showing up as miniwhite on insertion constructs? If they had only orange or pink eyes, rather that wildtype red eyes, then that could explain it. As part of addressing this, the Materials and methods should include full genotypes for all experimental flies, not just stock references, to be able to evaluate this sort question directly.

---

## [Author Response]

The four reviewers found the work to be of high significance for a pathway that plays an important role in flying insects but has not yet received the attention it deserves. Your work is very thorough and impressive, reconstructing the pathway(s) from the eye to the central brain, and eLife would be happy to publish it after you address the comments of the reviewers.Besides a long series of useful suggestions from the four reviewers that have to do with the presentation of the data and the clarity of the figures, there are three major points that we would like you to address:– One is to take advantage of the hemibrain reconstruction and to connect your neurons and connections to these recent data.

In discussing our own results we have made reference to the connectivity that can be found within the hemibrain dataset.

For easy reference we have added a table in Appendix 1 which links the neurons we recorded in each genetic driver line with what we believe to be the corresponding neurons in the hemibrain dataset.

We also refer to a recently published manuscript (Hulse et al., 2020) which contains a detailed analysis of the hemibrain dataset for the areas and neurons we investigated here

– It is also suggested that, while mapping ring neurons onto EPG cells, you integrate the recent work on plasticity by Fisher et al. and Kim et al. at a parallel synapse (and attempt to interpret the EPG response data from that angle).

We have expanded the discussion of our results in light of this recent work and provided some additional speculation (detailed below).

– The last point has to do with the PSI, which appears to have confused the reviewers as to the way you define it: Do you compute the PSI using the average fluorescence intensities rather than the DF/F? The reviewers suggest that you show more tuning curves to provide a better intuition of what this PSI means.

We have clarified this definition where first introduced. We have also added a new figure which contains tuning curves for each population of neurons recorded (Appendix 1—figure 1), although they are derived differently and in our opinion the PSI values enable a better comparison across the many different neuron populations studied:

tuning curves are normalized to baseline fluorescence levels in the neurons, which can influence the magnitude of the values calculated and can result in negative values; PSI values use absolute fluorescence levels (ranging from 0 to 2^16) and are normalized not to baseline activity but the sum of the activity levels during stimulus presentation, which results in only positive values ranging from 0 to 1.

Reviewer #1:The manuscript by Hardcastle et al. is a rather monumental work that moves the understanding of the neural circuits for polarization vision in *Drosophila* from non-described to almost fully described in one go. I feel that this could easily have been three separate papers and reading it was a bit overwhelming at times, but it is a truly remarkable manuscript.

In an attempt to make the narrative less overwhelming we have added subheadings within the Results section to better guide the reader. We have also split the following into two separate figures each: Figure 2, Figure 2—figure supplement 1, and Figure 3.

It starts with describing polarization sensitive circuits in the optic lobe, then moves on to the processing stages along the anterior optic pathway and eventually focuses its attention to the central complex. While the first two parts are comparatively straight forward, the central complex part is by far the most interesting (to me). This is mostly because there is a wealth of surprising findings that go far beyond what is known in other species and additionally provide many fascinating access points for future studies, both in *Drosophila* and in insects that have been classical subjects of polarized light vision research.The work follows a nice narrative, covers a wide range of anatomical and functional methods, is meticulously done, the methods are sound and the data are highly informative and relevant. While I cannot comment on the fly genetics part of the work (not my expertise), all other aspects of the work are excellent in my view.Regarding the presentation of the data, I found the figures too small and overwhelming, with not all panels being referred to in the text. Maybe less information would make the message clearer at times, but in any case, I recommend making the figures much larger.

We have resized and/or rearranged all figures to improve the presentation

I much liked the Discussion, and it was refreshing to see that the authors are keenly aware of the work in non-fly species and integrate this work in their interpretation whenever needed.I cannot wait to see follow up work, especially the interaction between the heading direction bump in the EPG neurons and the E-vector responses.

We’re very grateful for this encouragement.

1) The authors only follow up one of the two parallel pathways into the CX, given that the polarization signal is less clear in the superior bulb. Yet parallel processing in the AOT is a seemingly highly conserved feature and it would be fantastic to know at least anatomically, if these neurons (R4m and R2) maintain this separation on the level of the CX or if they converge on the same E-PG cells. This would not require physiology, but a short look at the hemibrain data. It would generate strong predictions about the functional significance of these parallel pathways.

Yes, both populations converge on the same E-PG neurons.

2) The authors use the classic division of the PB into eight glomeruli, yet data from *Drosophila* shows that there are nine glomeruli in the PB, including for the EPG neurons imaged in this study. This should be reflected in the analysis in this study. If the ninth neuron pair is not stained in the used driver line (as appeared in Wolff et al., 2015), this needs to be at least discussed. Judging by the projection patterns of column 1 and 9, the physiology of both should be identical, so no effect on the distribution of the data is likely. However, the reported discrepancy in columnar pairing across the midline between locusts and flies will have to be revised accordingly. It should be noted that the innermost column (near the midline) is the one that exists additionally in flies), as apparent by the lack of PEN and PFN neurons in this region.

We have added discussion of these points where the columnar arrangement of the E-PG neurons is first introduced, under the new subheading “E-PG organization and pairings in the PB”.

We agree that the activity profiles of columns 1 (E-PG) and 9 (E-PGt, not included in the driver line used here) would likely be the same.

We have maintained our analysis which refers only to columns 1-8 on each side and disregards the E-PGt neurons, since we assume that they do not represent a unique position/response in the array: E-PG neurons in columns 1 and 8 are therefore considered neighbors in the PB, as they are in the EB.

3) The focus of activity in one quadrant of the EB when stimulating the ring neurons is highly intriguing! This means that there are local calcium dynamics at work that are restricted to parts of the arborization trees of the ring neurons. This to me is very unexpected and exciting. Also, this would not be reflected in the electrophysiology of these cells, when recording the spiking rates in response to the stimuli and might therefore be a general mechanism relevant also for locusts etc. I suggest to emphasize this a bit more, as it literally changed my way of thinking about the entire system! Also, it is intuitively very much in line with the recent papers on plasticity between ring neurons and EPG cells, where coincident activity in both cell types leads to synaptic depression and hence disinhibition of EPG cells and the generation of a visually driven EPG activity bump. Integrating this recent work would benefit the paper in general, but seems crucial for interpreting data on this critical part of the pathway.

We also found this result to be exciting and unexpected. We have added to the Discussion some interpretation of these points and speculation as to how the system may operate:

“We noted that in some recordings, above-threshold PSI values were spatially localized to approximately one quadrant of the EB (Figure 9A, B, top, arrowhead). […] Increased activity within a section of the EB, as we observed here, may facilitate this process.”

4) along the lines of point three above; the recent work on plasticity of the synapse between ring neurons and EPG cells demonstrates that the stimulation history of these cells is critically important for their activity profiles. What role could this effect have on the tuning distribution in EPG cells?

We completely agree with this idea and have speculated about the consequence of this effect:

“The differences between ER4m and E-PG responses, in their periodicity and their PSI values, certainly imply that a transformation or gating of signals occurs at their synapses. […] The plasticity of these synapses, together with the reciprocal synapses from E-PG neurons back onto ER4m neurons, also invite the question of whether the observed preference of ER4m populations for particular AoPs in the EB may also have a complex dependence on internal state or previous visual experience.”

5) Finally, data on other species nearly always used continuously rotating E-vectors as stimuli rather than stationary E-vectors. As the rotation direction strongly affects the tuning angle (see for example Stone et al. 2017, Figure 1), the data might not be entirely comparable. Especially when dynamic processes like in the transformation of the ring neuron code to the EPG neuron code are important, these differences might have large effects on the results. Could the authors comment on these differences in the Discussion?

To clarify, our procedure did contain rotations with the light on, between the static presentations – although these sections were not analyzed, the direction of rotation could still have affected the responses. We performed additional experiments with the rotation direction reversed (not explicitly mentioned in the manuscript) and found no observable differences in responses or polarotopic organization upstream of the CX. In E-PG neurons, the responses were variable even for the same rotation direction, hence interpreting the effect of reversing the rotation alone was tricky, however we intend to follow up this line of investigation in future work. We agree that the data may not be comparable with those obtained from a continuous rotation and we have added a note on this difference:

“The discontinuous rotation of the polarizer differs from most previous studies of polarization-sensitive neurons in insects, which have typically analyzed spike frequency modulated by continuously rotating stimuli, and as such the data obtained using this protocol may not be directly comparable.”

Reviewer #2:This is an important and impressive study filling a critical gap in the literature. The authors trace polarization vision pathways in *Drosophila* from the periphery all the way to the central complex -- a structure involved in orientation behaviors in insects. This will allow to synthesize two streams of research on the subject: that polarization is an essential orientation cue used by many insects, mapped onto central complex neurons in some species on one side, and that the central complex circuits contain representations of the animal heading that is used to guide flexible behaviors, the dissection of which has been rendered possible by the *Drosophila* toolkit in recent years on the other side. The authors mix careful anatomical work with circuit tracing and in vivo imaging to delineate the polarization pathway while characterizing the transformations taking place at each stage. The manuscript is clear, systematic, and does not shy away from the complexity of the data. This is undoubtedly going to be an important resource for researchers in the field. I also do appreciate that code and data are made easily accessible.1) Figure 7 would be more convincing to me if the Bulb data, rather than the EB data, was shown. As the author point out, EB imaging of densely packed ring neurons axonal processes is conducive to more artefacts than the Bulb cleanly separated glomeruli (besides area, the signal in the EB could be dominated by neurons randomly expressing higher concentration of the indicator, etc)

We present the data for both the EB and the bulb in panels E and F, whereby the bulb data represent responses obtained from individual neurons. The effect of an overall population tuning was less pronounced in the bulb than in the EB, but still larger than that found for the population of TuBu neurons in the anterior bulb. However, as the reviewer notes in point 6, the imaging angle/projection used may make some parts of the bulb volume obscured/less accessible, and so neither approach is ideal. We also considered it an advantage to be able to demonstrate the responses of ER4m neurons deriving from both hemispheres together by showing example recordings from the EB.

2) E-PG responses: the authors are upfront with the fact that the data are not entirely conclusive, but I think they could discuss a number of other possibilities of why that might be so. First, ER4m ring neuron synapses onto E-PG are likely inhibitory. Given that in that immobilized preparation the EB attractor network is not "engaged" and therefore the activity level possibly low, this might make ER4m driven responses in the EPG challenging to detect. Second, if one imagine that a mechanism similar to the one described in Fisher et al., 2020, and Kim et al., 2020, is at play at the ER4m to EPG synapses, then it's possible that the mapping between ER4m activity profiles and EPG bump position has not taken place as the animals were reared in conditions where polarized light plays little role, and that moreover, once immobilized while a polarizer is rotated, this mapping would deteriorate even further. I think the paper would gain from discussing such possibilities, as well as qualifying the findings made on EPG responses in that light.

As noted above (Editorial, reviewer #1 section), we have expanded the discussion and interpretation of these results. We have incorporated the interpretation proposed by the reviewer, which we also consider as an entirely likely scenario.

3) Throughout the manuscript, it might be helpful to show some tuning curves (meaning response amplitude vs. polarization angle) as for now it is pretty hard to compare responses of the various types to each other.

As noted above (Editorial section), we have included a new figure (Appendix 1—figure 1) which presents tuning curves for each neuron population together.

4) Figure 1A could be improved by more hand holding for the reader unfamiliar with the DRA. It is currently hard to relate the inset to the rest of the panel.

We agree with this assessment and have completely revised the panel.

5) Figure 5 and arguments about the distribution of preferred angles (Figure 5F, G). This might need to be qualified: first, the possible functional role of such a distribution can only be inferred if one knows how the post-synaptic partners sample it. Is there an argument for the sampling being homogeneous? Second, this is not impossible for some of this to be imputable to the imaging angle/projection used, where some part of the volume of the BU are less accessible/blurred by others.

We thank the reviewer for raising these good points and have added the following section to address them:

“Without further investigation of the downstream circuitry in the central complex that processes these polarized light inputs from the bulb, our interpretations of possible functional roles are somewhat speculative. […] However, due to the physical shape of the bulb and the organization of micro-glomeruli within it, combined with the relatively low resolution of our functional imaging experiments, we could not ascertain whether all TuBu neurons were individually separable and their spatial locations accurately captured within the imaging volume, and these issues could have affected the uniformity of the distributions that we observed.”

Reviewer #4:Hardcastle and colleagues present work tracing polarized light signals through *Drosophila's* visual system, from the dorsal rim into the central complex. This work is important because it identifies pathways in *Drosophila* using synaptic tracing and employing tools that target specific neural populations. It ties its work in nicely to the extensive work on polarization vision in other insects. It is clearly written and its experiments are well conceived and explained.I noted three areas where I think the paper could be strengthened:1) The tracing and anatomical work is well done and convincing. However, since trans-tango's specificity to functional synapses remains a little uncertain, I think it would be very helpful to use the hemibrain dataset published earlier this year to confirm synapses between the neuron classes identified here. As a second part of this, and perhaps more important, it would be great to identify the correspondence between the TuBu type they reported and what is labeled in the hemibrain dataset. As of now on the hemibrain, there are 10 TuBu types (from TuBu01 to TuBu10) and TuBu01 seems to be specifically connected to R4m. This would be very important and useful information for readers who would want to further investigate these neurons.

As noted above (Editorial section), we have made numerous revisions to weave in the hemibrain dataset. In particular, we have added a table (Appendix 1—table 1) where we specify the driver lines used, match them to specific neuron types as annotated in the hemibrain dataset, and include data on the number of individual neurons assigned to each neuron type in the hemibrain/driver line.

2) The PSI is defined as (R_para_ – R_ortho_)/(R_para_ + R_ortho_). This maxes out at \pm 1 when all the responses are positive, but if any are negative, this formulation for an index doesn't work as well (for instance, allowing a divide by 0). I'm a bit confused about how this index was computed throughout, given that, for instance, the DmDRA1 neurons are inhibited by R7 via ort at anti-preferred angles, so there are nominally negative responses. Moreover, in the Materials and methods, it says F0 was computed as RMS F(t), to computed dF/F, but I don't see how this wouldn't lead to negative response values. I think the methodology for computing DSI needs to be clarified. In particular, how are the authors computing the responses – deviations from a baseline (computed how?) or are they setting the minimum measured dF/F to be a new 0 response?

As noted above (Editorial section), we have clarified these details in the text where first introduced.

3) It was quite surprising to me that a small source of polarized light could activate so much of the dorsal rim, as the authors also note. Were the flies w-/w- with the white gene only showing up as miniwhite on insertion constructs? If they had only orange or pink eyes, rather that wildtype red eyes, then that could explain it. As part of addressing this, the Materials and methods should include full genotypes for all experimental flies, not just stock references, to be able to evaluate this sort question directly.

That’s correct, and an interesting explanation. We cannot rule out the possibility that light hitting the rhabdomeres side-on or obliquely, as opposed to axially after entering via the lens, could have been responsible for some portion of the responses observed. However in that case we would expect that R7 and R8 responses would be similar for a given AoP, and neither would show a strong selectivity for a particular AoP. We therefore don’t believe that this effect alone could be responsible for the responses we observed.